# Host genetic regulation of human gut microbial structural variation

Daria V. Zhernakova[1,18], Daoming Wang[1,2,18], Lei Liu[3,18], Sergio Andreu-Sánchez[1,2], Yue Zhang[1,2], Angel J. Ruiz-Moreno[1,2], Haoran Peng[1], Niels Plomp[3,4], Ángela Del Castillo-Izquierdo[1,3], Ranko Gacesa[1,4], Esteban A. Lopera-Maya[1], Godfrey S. Temba[5,6,7], Vesla I. Kullaya[6,8], Sander S. van Leeuwen[9], Lifelines Cohort Study*, Ramnik J. Xavier[10,11], Quirijn de Mast[5,7], Leo A. B. Joosten[5,12], Niels P. Riksen[5], Joost H. W. Rutten[5], Mihai G. Netea[5,7,13,14], Serena Sanna[1,15], Cisca Wijmenga[1], Rinse K. Weersma[4], Alexandra Zhernakova[1], Hermie J. M. Harmsen[3,19 ✉] & Jingyuan Fu[1,2,19 ✉]

Although the impact of host genetics on gut microbial diversity and the abundance of specific taxa is well established[1–6], little is known about how host genetics regulates the genetic diversity of gut microorganisms. Here we conducted a meta-analysis of associations between human genetic variation and gut microbial structural variation in 9,015 individuals from four Dutch cohorts. Strikingly, the presence rate of a structural variation segment in *Faecalibacterium prausnitzii* that harbours an *N*-acetylgalactosamine (GalNAc) utilization gene cluster is higher in individuals who secrete the type A oligosaccharide antigen terminating in GalNAc, a feature that is jointly determined by human *ABO* and *FUT2* genotypes, and we could replicate this association in a Tanzanian cohort. In vitro experiments demonstrated that GalNAc can be used as the sole carbohydrate source for *F. prausnitzii* strains that carry the GalNAc-metabolizing pathway. Further in silico and in vitro studies demonstrated that other *ABO*-associated species can also utilize GalNAc, particularly *Collinsella aerofaciens*. The GalNAc utilization genes are also associated with the host's cardiometabolic health, particularly in individuals with mucosal A-antigen. Together, the findings of our study demonstrate that genetic associations across the human genome and bacterial metagenome can provide functional insights into the reciprocal host–microbiome relationship.

Gut microorganisms and humans have evolved in a symbiotic relationship. Humans provide an intestinal environment with resources for microorganisms to live, and gut microbes can provide bioactive molecules that affect human physiology and mediate the impact of dietary and environmental exposures on humans[7–11]. Gut microorganisms can also protect their host against other pathogenic microorganisms, train the immune system and play other important roles in human health[12,13]. Although there are some data showing host–microorganism symbiotic relationships[14–16], genetics-based evidence remains limited. So far, several human genomic loci have been associated with the abundance of several taxa, including well-replicated associations with the *LCT* and *ABO* genes[1]. However, little is known about genetic interaction between the human genome and the gut microbiome, a fact supported

by the discovery of population-specific strains[17]. This led us to reason that associations between genetic variants in the human genome and those in the human metagenome can provide functional insights into the host–microorganism symbiotic relationship. To our knowledge, such analyses have not yet been carried out at the whole-genome scale.

Bacterial genomes are known to evolve rapidly. Genomic variation leads to bacterial strains that can differ in fitness, carbohydrate utilization, metabolizing capacity, pathogenicity and other biological properties[18]. Bacterial structural variations (SVs) are highly variable genomic segments, of variable lengths, that can exert pronounced effects on microbial functionality, increasing bacterial genome plasticity and enabling rapid adaptation to environments[19]. SVs are common in human gut microbial genomes and there is a large inter-individual

[1]University of Groningen, University Medical Center Groningen, Department of Genetics, Groningen, The Netherlands. [2]University of Groningen, University Medical Center Groningen, Department of Pediatrics, Groningen, The Netherlands. [3]University of Groningen, University Medical Center Groningen, Department of Medical Microbiology and Infection Prevention, Groningen, The Netherlands. [4]University of Groningen, University Medical Center Groningen, Department of Gastroenterology and Hepatology, Groningen, The Netherlands. [5]Department of Internal Medicine, Radboud University Medical Center, Nijmegen, The Netherlands. [6]Department of Medical Biochemistry and Molecular Biology, Kilimanjaro Christian Medical University College, Moshi, Tanzania. [7]Radboud Center for Infectious Diseases, Radboud University Medical Center, Nijmegen, The Netherlands. [8]Kilimanjaro Clinical Research Institute, Kilimanjaro Christian Medical Center, Moshi, Tanzania. [9]University of Groningen, University Medical Center Groningen, Department of Laboratory Medicine, Groningen, The Netherlands. [10]Broad Institute of MIT and Harvard, Cambridge, MA, USA. [11]Center for Computational and Integrative Biology, Department of Molecular Biology, Massachusetts General Hospital, Boston, MA, USA. [12]Department of Medical Genetics, Iuliu Hațieganu University of Medicine and Pharmacy, Cluj-Napoca, Romania. [13]Department of Immunology and Metabolism, Life and Medical Sciences Institute, University of Bonn, Bonn, Germany. [14]Human Genomics Laboratory, Craiova University of Medicine and Pharmacy, Craiova, Romania. [15]Institute for Genetic and Biomedical Research, National Research Council, Cagliari, Italy. [18]These authors contributed equally: Daria V. Zhernakova, Daoming Wang, Lei Liu. [19]These authors jointly supervised this work: Hermie J. M. Harmsen, Jingyuan Fu. *A list of authors and their affiliations appears at the end of the paper. ✉e-mail: h.j.m.harmsen@umcg.nl; j.fu@umcg.nl

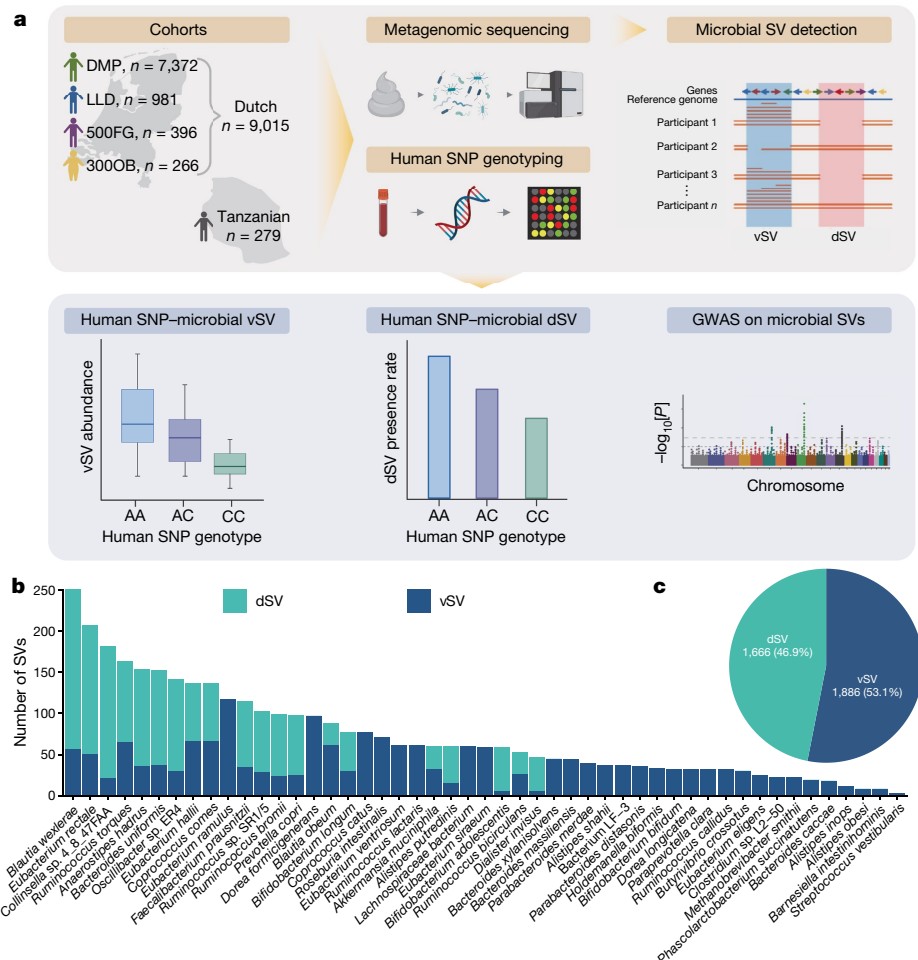

**Fig. 1 | An overview of the workflow and microbial SV data. a**, The workflow of this GWAS on gut microbial SVs. In this study we integrated data for 9,015 individuals from whom both gut microbial metagenomic and host genetic data are available from four cohorts. Data from a Tanzanian cohort of 279 individuals were included as a replication cohort. We generated gut microbial SV profiles based on metagenomic sequencing, with the dSVs and vSVs then subjected to association analysis with the genotypes of more than 6 million common SNPs in the human genome. The genetic associations are presented in box plots for vSVs, bar plots for dSVs and Manhattan plots for the whole-genome level. Created with BioRender.com. **b**, The number of common microbial SVs detected in 49 species for GWAS. Each bar represents a species. The *y* axis refers to the number of common SVs detected in that species. dSVs and vSVs are coloured in green and blue, respectively. **c**, A pie chart of the 3,552 common SVs, including 1,666 dSVs and 1,886 vSVs, involved in the GWAS.

difference in microbial SVs between humans[20–22]. Identification of deletion SVs (dSVs; genomic regions that are either detectable or absent in the metagenomic sample) or variable SVs (vSVs; genomic regions whose abundances are highly variable across samples) using metagenomic sequencing has revealed that gut microbial SVs are related to human health[20–22]. Longitudinal analysis has demonstrated that gut microbial SVs show species-specific temporal stability[22]. This suggests a potential adaptation of gut bacteria to the individual-specific intestinal environment. However, little is known about how human genetics shapes the individual's intestinal environment and exert selective pressure on the genetic landscape of the gut microbiome. The limited studies carried out thus far usually focused on bacterial or viral isolates[23,24]. Genetic association between human genetic variants and microbial SVs may thus help us understand the mechanisms underlying the symbiotic relationship between gut microorganisms and their human host.

In the present study, we carried out a large-scale meta-analysis of genetic associations between human genotypes and microbial SVs in the gut microbiome, involving 9,015 individuals from four Dutch cohorts. Associations significant at the Bonferroni-corrected *P* < 0.05 level were then replicated in a Tanzanian cohort (*n* = 279). Follow-up bioinformatics and experimental validation pinpointed causal genes involved in host–microbiome interaction and improved our functional

understanding of human genetic regulation of gut microbial genetic diversity.

## Heritability of gut microbial SVs

This study involved 9,015 Dutch individuals for whom both metagenomic and host genetic data were available (Fig. 1a). These individuals came from four Dutch cohorts: the Dutch Microbiome Project[8] (DMP; *n* = 7,372), Lifelines-DEEP[25] (LLD; *n* = 981), the 500 Functional Genomics Project[26] (500FG; *n* = 396) and 300-Obesity[21] (300OB; *n* = 266). To replicate associations in individuals with a different genetic background and lifestyle, we involved the 300-Tanzanian cohort (300TZFG; *n* = 279) as a replication cohort. The analysis workflow is presented in Fig. 1a.

We used SGV-Finder[20] to generate SV profiles. In brief, this method mapped sequencing reads to reference genomes, resolved possible ambiguous read alignments and then split the microbial genomes into bins. The metagenomic coverage of these bins was compared across samples (Methods). SGV-Finder identifies bins with coverage close to 0 in 25–75% of samples as dSVs and bins that show variable coverage as vSVs. SV identification is possible only for gut microbial species with sufficient metagenomic sequencing coverage (Methods). In total, we detected 14,196 SVs in 108 gut microbial species, including 10,265 dSVs

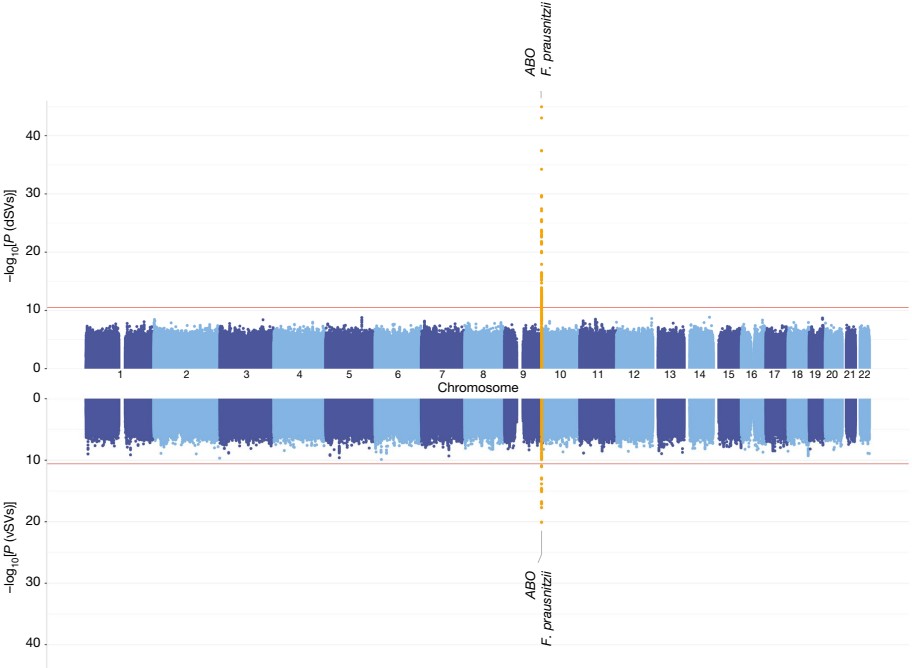

**Fig. 2 | A Manhattan plot of genome-wide associations of human SNPs and gut microbial SVs.** GWAS results for dSVs (top) and vSVs (bottom). The $x$ axis shows the genomic position on the human chromosomes (chromosomes 1–22) for both the top and bottom panels. The $y$ axes in both panels show statistical significance as $-\log_{10}[P]$ estimated using a linear mixed model by fastGWA. The plotted $P$ values are not adjusted for multiple testing. The red horizontal lines indicate the study-wide significance cutoffs determined using the Bonferroni method: $3.00 \times 10^{-11}$ for dSV and $2.65 \times 10^{-11}$ for vSV associations. Significantly associated loci are highlighted in yellow and labelled with the nearby human gene and the corresponding species name.

and 3,931 vSVs, with 3–379 SVs per species (Extended Data Fig. 1a,b and Supplementary Table 1). The species with the largest number of SVs were *Dorea formicigenerans*, *Dorea longicatena* and *Blautia wexlerae* (Extended Data Fig. 1c and Supplementary Table 1). The number of samples with sufficient coverage to detect SVs ranged from 11 to 7,716 for different species. The abundance of these species collectively accounted for an average of 80.8% of faecal microbiome composition (range 17.8% to 97.1%; Extended Data Fig. 1d). To ensure statistical power for association with host genetics, we selected those vSVs that were detected in at least 10% of samples and dSVs with deletion rates between 5% and 95% (Methods). This resulted in 3,552 SVs including 1,666 dSVs and 1,886 vSVs from 49 bacterial taxa (Fig. 1b,c and Supplementary Tables 1–3).

To assess the extent to which the gut microbial SVs can be determined by host genetics, we first estimated the heritability of 1,339 out of 3,552 SVs that were present in 1,092 first- or second-degree relative pairs from the DMP cohort. After correcting for species abundance, family-based heritability estimation ($h^2$) revealed one heritable dSV at a false discovery rate of <0.05 (Supplementary Table 4): a 2-kilobase (kb) dSV of *F. prausnitzii* (577–579 bp) with an estimated $h^2$ of 0.38. In addition, 26 dSVs and 51 vSVs showed nominally significant heritability ($P < 0.05$), with an average $h^2$ of 0.28 and 0.41, respectively (Supplementary Tables 4 and 5). Next, we compared SV heritability with species abundance heritability and observed an additional effect of host genetics on microbial SV level (Extended Data Fig. 2 and Supplementary Note 1). However, this study still lacks sufficient power for heritability calculation and comparison. Accurate heritability estimations of species abundance and microbial genetic variation would require a much larger sample size and careful experimental design (for example, twin studies).

### *ABO* locus and *F. prausnitzii* SVs

Next we associated the 3,552 SVs with more than 6 million human single nucleotide polymorphisms (SNPs) per cohort, followed by a meta-analysis. The genetic associations significant at the Bonferroni-corrected $P < 0.05$ level were all associations between the *ABO* locus and SVs of *F. prausnitzii*, including four dSVs and one vSV (Fig. 2, Extended Data Fig. 3a and Supplementary Tables 6 and 7). The strongest association was found between rs635634 and a 2-kb dSV region (577–579 kb) of *F. prausnitzii* ($b_{meta} = 0.88$, $P_{meta} = 1.21 \times 10^{-45}$). The SNP rs635634 is located in the *ABO* gene, which encodes a glycosyltransferase that modifies oligosaccharides on the cell surface and determines the ABO blood group. The *ABO* locus is one of the few loci that have repeatedly been associated with the abundances of several gut bacteria, including *Collinsella* species, *Bifidobacterium* and *Faecalibacterium*[2,6,27].

We further replicated identified associations in the 300TZFG[28] cohort, which had distinct genetic background, lifestyle and environmental exposures (Supplementary Note 2). SVs of *F. prausnitzii* were detected in 201 individuals from 300TZFG, either at similar or different frequencies compared to those observed in the Dutch cohorts (Extended Data Fig. 3b). We detected 156 associations with the *ABO* locus at a nominally significant level ($P < 0.05$; Supplementary Table 8). Two *F. prausnitzii* dSVs, 575–577 and 577–579, showed association with *ABO* (Extended Data Fig. 3c,d), encompassing both shared signals and population-specific signals.

In addition to the *ABO* association, our study also yielded 210 independent suggestive associations (clumping linkage disequilibrium $r^2 < 0.1$) at the genome-wide significance $P < 5 \times 10^{-8}$ level: 58 associations with dSVs involving 17 species and 152 associations with vSVs involving 33 species (Supplementary Tables 6 and 7).

### *ABO* association is dependent on *FUT2*

ABO genotype determines host blood type, and we further analysed whether the association with *ABO* SNPs represented the association with *ABO*-coded blood groups in Dutch samples. Blood groups were imputed using SNP genotype data (Methods). Indeed, all five *ABO*-associated *F. prausnitzii* SVs were associated with the host's ABO

blood group (Extended Data Fig. 4 and Supplementary Table 9). The *F. prausnitzii* 577–579-kb dSV region was more frequent in individuals with blood group A or AB than in individuals with blood group B or O ($P_{meta} = 1.24 \times 10^{-44}$, $P_{DMP} = 1.03 \times 10^{-32}$). The association was also dependent on *FUT2* secretor status, which determines whether fucosyl precursors of A- or B-antigens are secreted into body fluids and intestinal mucus. The secretor-determining SNP rs679574 itself was suggestively associated with the presence of this dSV ($P_{meta} = 2.92 \times 10^{-9}$), and A-antigen presence was associated with the *F. prausnitzii* 577–579 dSV only in FUT2 secretors ($P_{meta} = 4.85 \times 10^{-51}$, $P_{DMP\_secretors} = 9.39 \times 10^{-37}$, $P_{DMP\_nonsecretors} = 0.88$; Fig. 3a). After correcting for the population genetic structure of *F. prausnitzii* (Extended Data Fig. 5a and Supplementary Table 10), *F. prausnitzii* associations with the *ABO* locus remained significant ($P_{DMP} = 2.24 \times 10^{-32}$; Supplementary Table 11).

The *ABO* locus was previously associated with the abundance of *Faecalibacterium* species in a German cohort, with a rather modest effect size ($\beta = -0.14$, $P = 4.33 \times 10^{-9}$)[6]. This association was not replicated in two other studies[2,27] or in our cohorts ($P_{DMP\_secretors} = 0.08$; Extended Data Fig. 5b). Notably, we did observe a significant interaction between the blood group and dSV 577–579 ($P_{DMP\_secretors} = 1.47 \times 10^{-3}$) on the abundance of *F. prausnitzii*, suggesting that the ABO association with *F. prausnitzii* abundance may depend on the presence of the dSV region.

## GalNAc pathway in the SV region

A-antigen is an oligosaccharide that can be secreted into intestinal mucus and degraded by carbohydrate-active enzymes of gut bacteria[29–31]. Therefore, we reasoned that the associated SV regions may give *F. prausnitzii* the capacity to utilize saccharides released from A-antigen as a carbohydrate source. All five ABO-associated *F. prausnitzii* SVs were modestly correlated with each other (Spearman correlation $R > 0.13$, $P < 0.05$; Supplementary Table 12). After adjusting for other associated SVs, the strength of associations decreased, and the association of two dSVs (577–579 and 1154–1155) out of the five SVs remained significant after Bonferroni-correction, suggesting that other SVs partially tag the same signal as the top 577–579 dSV (Supplementary Table 13). However, most of the dSVs still showed significant associations, especially the top ABO-associated 577–579-kb dSV region. This means that the 577–579-kb dSV captured most of the signal, but not all. To fine-map the microbial genomic region that captures the causal genes, we isolated *F. prausnitzii* from human faeces, carried out whole-genome sequencing and selected 12 distinct *F. prausnitzii* strains. Seven strains showed a deletion that overlaps with the top *ABO*-associated 577–579 segment (Supplementary Fig. 1), expanding this 2-kb dSV region to a 23-kb region. We then used the *F. prausnitzii* HTF-238 strain with this complete region (2,640–2,663 kb) as the reference for gene characterization.

In this expanded region, we identified 27 genes (Supplementary Table 14), including those involved in carbohydrate metabolism, particularly the pathway involved in GalNAc metabolism, including a cluster of genes responsible for the uptake and metabolism of D-galactosamine and GalNAc (Fig. 3b,c and Supplementary Table 14). GalNAc sugar is part of the A-antigen encoded by *ABO*, and it might be used as an energy source for bacteria when it is secreted to mucus[32]. Specifically, the region contains one gene, *GH109*, that encodes a glycoside hydrolase that can cleave GalNAc from A-antigen, as well as nine genes involved in five key metabolic steps of downstream GalNAc utilization (Fig. 3b and Supplementary Note 3). Moreover, the region also contains two genes involved in the galactose degradation pathway (the Leloir and tagatose 6-phosphate (T6P) pathways). Other genes and genetic elements in this region, including transcriptional regulators, transposons and several uncharacterized genes, were not likely to be directly involved in carbohydrate metabolism.

Furthermore, we found that this SV region is likely to be a mobile element. By investigating SV sharedness between cohousing individuals,

we found evidence to support the transmission of GalNAc-containing strains between people. Moreover, a 4-year follow-up analysis in 119 individuals shows a higher frequency of gain than of loss of GalNAc-containing strains over time (Extended Data Fig. 6a–e, Supplementary Fig. 2 and Supplementary Note 4).

## Bacteria can use GalNAc as a carbon source

As multiple genes involved in carbohydrate metabolism were identified in the SV region of *F. prausnitzii*, we next investigated whether the genes in this region are crucial for bacterial utilization of the specific monosaccharide substrates, including GalNAc, galactose, glucose, lactose, mannose, *N*-acetylglucosamine, fructose, *N*-acetylneuraminic acid and 2′-fucosyllactose. All 12 selected *F. prausnitzii* strains were subjected to growth rate experiments in yeast casitone fatty acids (YCFA) medium with the monosaccharides above as the sole carbohydrate source, and YCFA without a carbohydrate source was used as a negative control.

The GalNAc utilization pathway turned out to be crucial for bacterial growth in the GalNAc medium. Strains lacking the GalNAc pathway could not grow (Fig. 3d), whereas six out of seven strains (except ATCC 27768) with the GalNAc pathway could grow, although HTF-383 exhibited slightly slower growth and reached a similar cell density level at a later time (Extended Data Fig. 7a). In contrast to the findings for GalNAc utilization, all strains were able to grow on galactose, but those with the region containing the Leloir and T6P pathways showed a higher growth rate than those without (Fig. 3e), indicating that these pathways, although not essential, can improve galactose utilization efficiency. The presence or absence of pathways in this region did not show a notable influence on bacterial utilization of other monosaccharides (Extended Data Fig. 7b).

## Inversion affects GalNAc gene expression

ATCC 27768 was the only strain that harbours the GalNAc pathway that did not grow in the GalNAc medium. However, the GalNAc region is reversed in ATCC 27768 (Fig. 3c), and this genomic inversion may result in dysfunction of this pathway. Thus, we carried out a GalNAc induction experiment to investigate the transcription of GalNAc genes and potential regulators (*ptsH*, *rhaR* and *immR*) in this region. ATCC 27768 was first pre-cultured in a glucose medium, and the resulting bacterial culture was split and transferred to either glucose or GalNAc medium (Methods). We then compared the expression fold change in GalNAc medium to that in glucose medium. The positive control was the close relative strain HTF-495, which can grow in GalNAc medium. The negative control was HTF-441, which lacks the GalNAc utilization gene cluster (Extended Data Fig. 8).

Gene expression of GalNAc genes was not detected in HTF-441, confirming their absence (data not shown). Notably, following GalNAc induction, the expression of three GalNAc uptake genes, *agaC*, *agaD* and *agaV*, was only marginally increased in ATCC 27768, whereas these genes showed a marked increase in HTF-495. For instance, GalNAc induction resulted in a 63.5-fold increase in *agaC* expression in HTF-495 compared to glucose induction, but in only a threefold change in ATCC 27768 (Fig. 3f). However, the expression of other GalNAc genes showed similar fold changes in ATCC 27768 and HTF-495 (Extended Data Fig. 8). This suggests that genomic inversion of ATCC 27768 affects the expression of only GalNAc uptake genes and not GalNAc metabolism genes.

## GalNAc pathway in other taxa

So far, the *ABO* locus has been associated with the abundances of nine bacterial taxa[2,6,27] (Supplementary Table 15), including those of three species: *C. aerofaciens*, *Faecalicatena lactaris* and *Bifidobacterium bifidum*. However, except for those of the genus *Collinsella*, none of these associations have been replicated in multiple studies.

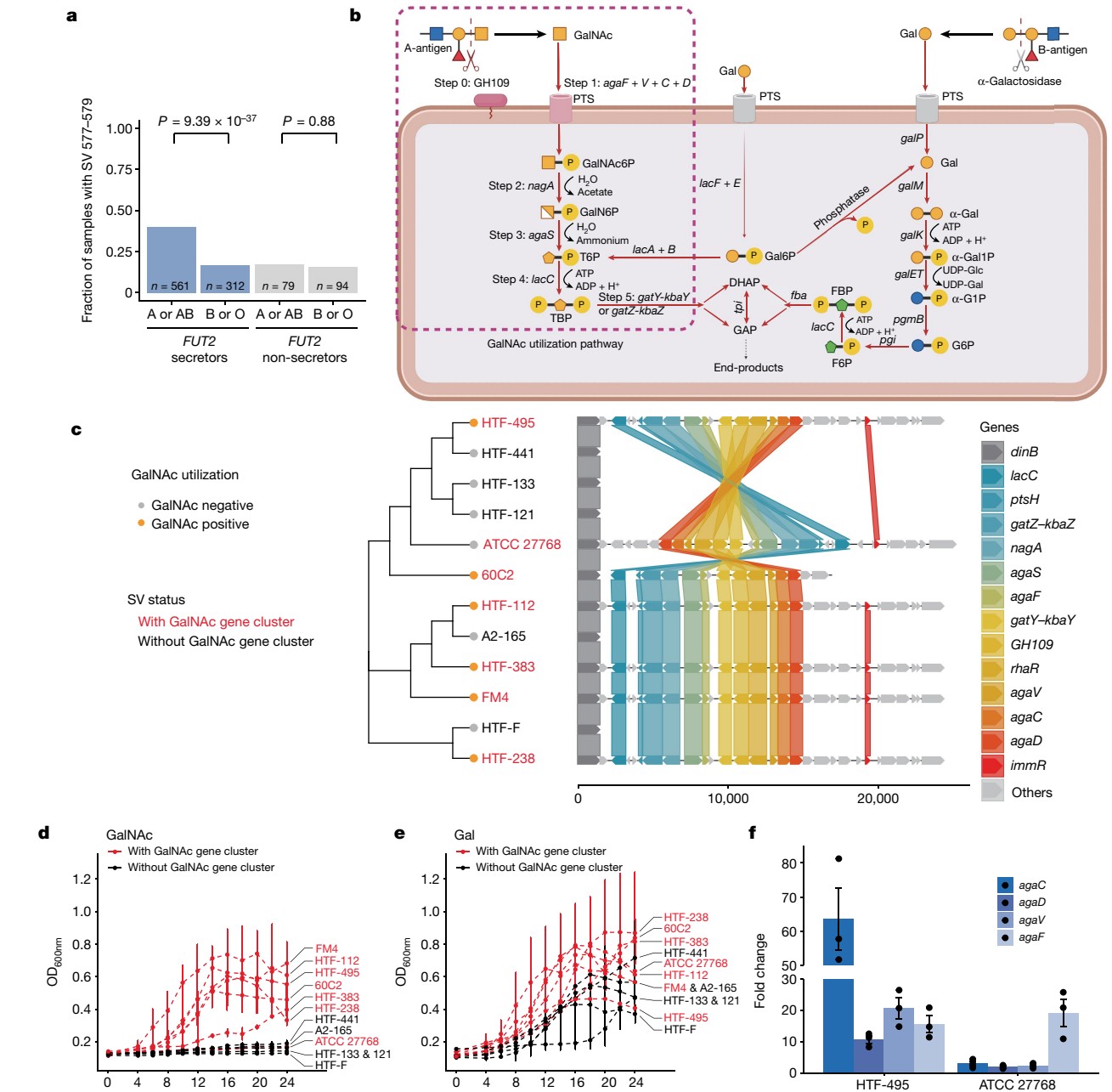

**Fig. 3 | GalNAc utilization underlies the *ABO* association with *F. prausnitzii* SVs. a**, Association between ABO blood type and the presence or absence of genomic segment 577–579 kb (SV 577–579) of *F. prausnitzii* is dependent on FUT2 secretor status. Individuals were grouped into different groups on the basis of blood types (A versus B or O blood types) and FUT2 secretor status (secretors versus non-secretors). The *y* axis refers to the fraction of individuals with the 577–579-kb region in the DMP dataset. Unadjusted association *P* values are reported, based on linear mixed models. **b**, A scheme of the GalNAc pathway identified in the associated SV region, which is divided into the initial cleavage from A-antigen (step 0) and five key steps of GalNAc utilization (steps 1–5; Supplementary Note 3). *agaF + V + C + D* genes encode four subunits of the GalNAc PTS system II complex protein. DHAP, dihydroxyacetone phosphate; FBP, fructose-1,6-bisphosphate; F6P, fructose 6-phosphate; GAP, glyceraldehyde 3-phosphate; PTS, phosphotransferase system; TBP, tagatose 1,6-bisphosphate. Created with BioRender.com. **c**, A phylogenetic tree of the strains used for the

culture experiments and organization of genes involved in GalNAc utilization in the SV region. The *x* axis indicates the base pair position starting from the flanking gene *dinB*. The different coloured lines indicate the location of the same genes in different strains. The names of strains with and without the SV region are in red and black, respectively. **d**,**e**, Growth curves of *F. prausnitzii* strains with or without SV 577–579 on medium with GalNAc (**d**) and galactose (**e**). The *x* axis refers to time points in hours. The *y* axis refers to cell density as measured as the optical density at 600 nm (OD$_{600\text{nm}}$). Points with bars on the growth curves represent means ± s.d. of three replicates. **f**, Fold change of gene expression following GalNAc induction compared to glucose induction. The *y* axis shows the fold change of the expression of the *agaC*, *agaD*, *agaV* and *agaF* genes in the strains HTF-495 and ATCC 27768 following GalNAc induction relative to glucose induction. Each dot indicates one replicate. Bars and error bars indicate means ± s.e. of three replicates.

We wondered whether the presence of the GalNAc pathway may explain the *ABO* association with the abundance of those taxa. We therefore extracted 10,487 assembled genomes of *ABO*-associated species from

the Unified Human Gastrointestinal Genome collection[33], including 1,103 assemblies of *C. aerofaciens*, 484 of *F. lactaris*, 1,109 of *B. bifidum* and 7,791 of *F. prausnitzii* (Supplementary Table 16). We then carried

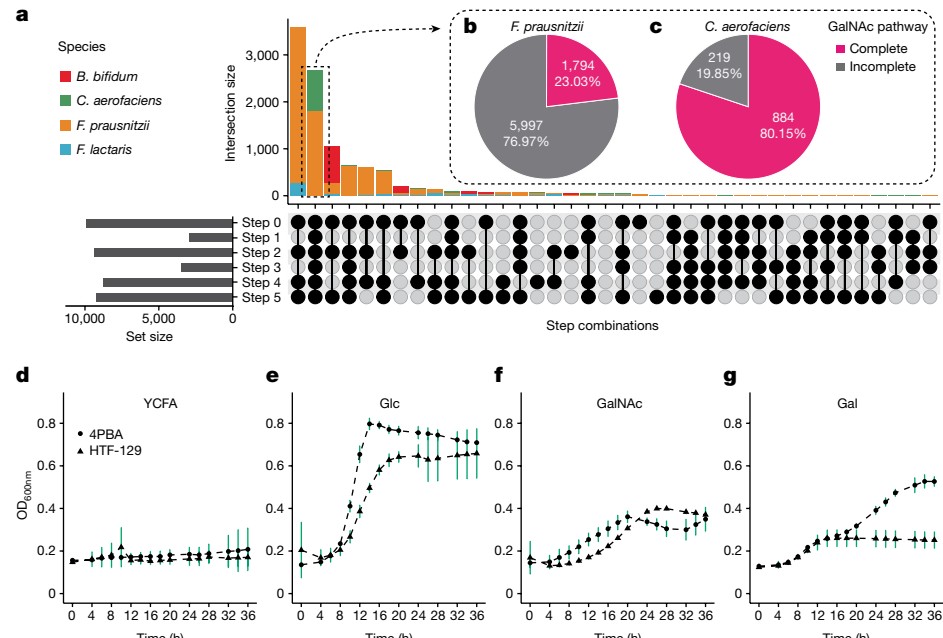

**Fig. 4 | GalNAc utilization capacity of strains of *F. prausnitzii* and other ABO-associated species. a**, Completeness of the GalNAc pathway in four gut microbial species associated with human ABO blood type: *F. prausnitzii*, *C. aerofaciens*, *F. lactaris* and *B. bifidum*. The upper bar plot shows the number of strains (intersection size), and the combination of black dots in each vertical column underneath represents the presence of the genes in the corresponding step. The bar plot on the left represents the number of strains (set size) containing the genes of each corresponding GalNAc metabolism step. **b**,**c**, Proportion of strains with a complete GalNAc pathway in *F. prausnitzii* (**b**) and *C. aerofaciens* (**c**).

**d**–**g**, Growth curves of *C. aerofaciens* strains containing GalNAc pathway genes on medium supplemented with different sugars. The *x* axis indicates the hours after culturing in the medium. The *y* axis indicates cell density measured as the $OD_{600nm}$ value. Points on the growth curves represent means ± s.d. of three replicates. YCFA medium and YCFA–glucose (Glc) medium were used as negative and positive controls, respectively. GalNAc and galactose (Gal) were supplied in YCFA medium to test whether *C. aerofaciens* can grow with the monosaccharides released from A-antigen and B-antigen as their sole carbohydrate sources.

out an orthologue search for the GalNAc pathway genes. We found that GalNAc genes were present in 28–95% of assemblies (Fig. 4a and Supplementary Table 16). However, the complete pathway was found in only 2,678 assemblies (26%), including 1,794 *F. prausnitzii* strains (23%) and 884 *C. aerofaciens* strains (80%) (Fig. 4b,c and Supplementary Table 16). The high fraction of GalNAc-pathway-containing strains of *C. aerofaciens* supports the association between *Collinsella* abundance and *ABO*. In accordance with these results, we also confirmed GalNAc utilization capacity for two *C. aerofaciens* strains (Fig. 4d–g). However, we did not detect the complete GalNAc pathway in *B. bifidum* genomes, suggesting a potentially different underlying mechanism for *B. bifidum* associations with human blood type.

## GalNAc utilization supports human health

We further estimated the total abundance of GalNAc genes in the whole microbial community. These GalNAc genes showed a strong intercorrelation, indicating that they are probably present as a gene cluster and function collaboratively. Similarly, the abundance levels of GalNAc genes were associated with the ABO blood type in FUT2 secretors (Extended Data Fig. 9 and Supplementary Table 17). The significance observed at the gene level was much stronger than the association with the *F. prausnitzii* SV region, with the lowest *P* value of $4.19 \times 10^{-223}$ observed for *lacC* (Extended Data Fig. 9 and Supplementary Table 17).

We further reasoned that the abundance of GalNAc genes might be more relevant for human health in individuals with mucosal A-antigens than for those without. To check this, we characterized individuals in our cohorts as having either genetically determined presence or absence of A-antigen in intestinal mucus, based on their *ABO* and *FUT2* genotypes. FUT2 secretors with A-antigens (A or AB blood type) were identified as individuals with mucosal A-antigen, and all others were

considered individuals without mucosal A-antigen. In line with our previous findings, the abundance of GalNAc genes showed remarkable differences between individuals with and without mucosal A-antigen. The top associations were found for the *lacC* gene involved in catalytic step 4 from T6P to tagatose 1,6-bisphosphate ($P = 1.30 \times 10^{-280}$) and the *gatY–kbaY* gene involved in catalytic step 5 from tagatose 1,6-bisphosphate to dihydroxyacetone phosphate or glyceraldehyde 3-phosphate ($P = 2.60 \times 10^{-259}$; Fig. 5a,b and Supplementary Table 17). As many gut microorganisms can have the GalNAc pathway, we further reasoned that the presence of mucosal A-antigen can provide an extra energy source to promote the growth of GalNAc utilizers. In agreement with this, our findings showed that the abundances of GalNAc genes were positively associated with microbial richness and diversity and that these associations were stronger in individuals with mucosal A-antigen ($P_{heterogeneity} < 0.05$, $I^2 > 0.7$; Fig. 5c, Extended Data Fig. 10a and Supplementary Table 18). For instance, the correlation between the abundance of the *agaF* gene and microbial richness was 0.26 (Spearman correlation, $P = 1.79 \times 10^{-29}$) in individuals with mucosal A-antigen but only 0.13 (Spearman correlation, $P = 1.13 \times 10^{-16}$) in individuals without mucosal A-antigen (Supplementary Table 18). We observed similar results after correcting for the presence of the 577–579 dSV and *F. prausnitzii* and *C. aerofaciens* abundances.

Similarly, we associated the abundances of microbial GalNAc genes with 240 environmental exposure and health-related parameters in individuals with and without mucosal A-antigen. At the Bonferroni-corrected *P* < 0.05 level, we detected 50 significant associations in the A-antigen presence group and 17 associations in the A-antigen absence group. Notably, microbial GalNAc gene abundances were significantly associated with blood glucose, Bristol stool type and general health only in individuals with mucosal A-antigen (linear regression, Bonferroni-corrected *P* < 0.05, $P_{heterogeneity} < 0.05$; Fig. 5d,e,

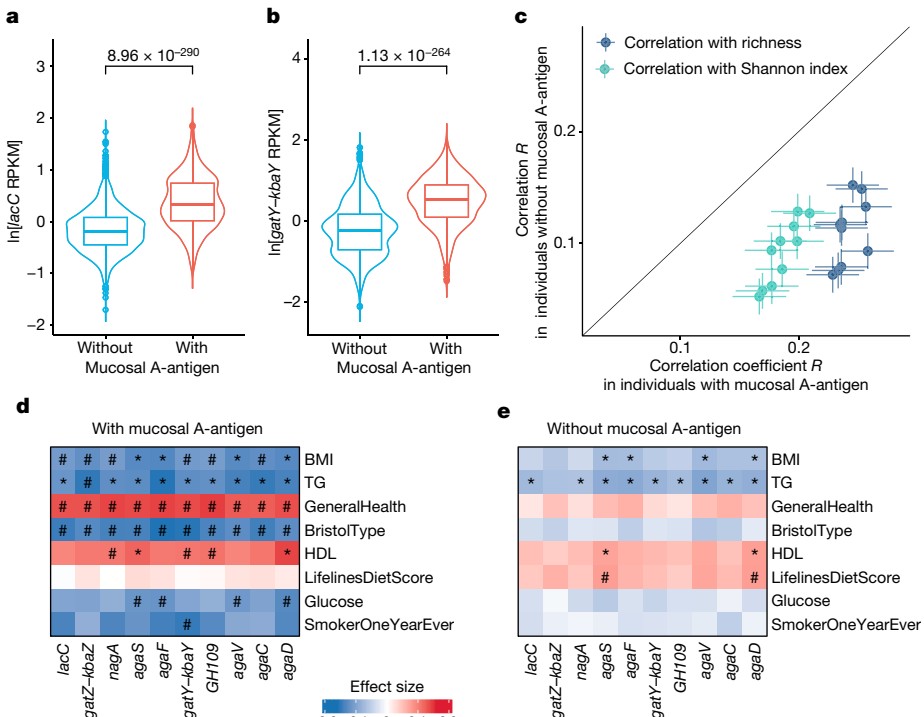

**Fig. 5 | Comparison of GalNAc associations between two groups of individuals with or without mucosal A-antigen. a,b**, Comparison of the abundance of GalNAc genes, *lacC* (**a**) and *gatY–kbaY* (**b**), between two groups of individuals ($n_{without}$ = 3,866 and $n_{with}$ = 1,868). The violin plots show ln-transformed gene abundance with units of reads per kilobase million (RPKM). The inner box plots represent summary statistics: the centre line represents the median, the box hinges represent the lower and upper quartiles of the distribution, whiskers extend no further than 1.5× interquartile range from the hinges, and data beyond the end of the whiskers are outliers plotted as individual points. Unadjusted *P* values are reported, based on linear mixed models. **c**, Comparison of the correlations of GalNAc metabolism gene abundance with gut microbiome α-diversity and richness between two groups: individuals with mucosal A-antigen (*x* axis; *n* = 1,868) and those without (*y* axis, *n* = 3,866). Each dot represents a Spearman correlation coefficient between a GalNAc gene with the Shannon index (green) or richness (blue). The error bars indicate the confidence interval of the correlation *R* estimation based on s.e. **d,e**, The association between GalNAc metabolism gene abundance and host phenotypes in individuals with (**d**; *n* = 1,868) and without (**e**; *n* = 3,866) mucosal A-antigens (unadjusted *P* values are estimated from linear regression). Hash symbols indicate group-specific significant associations. Asterisks indicate significant associations shared by the two groups (Bonferroni-corrected *P* < 0.05). Exact *P* values are listed in Supplementary Table 19. Positive associations are in red. Negative associations are in blue. The colour gradients reflect the effect size. BMI, body mass index; BristolType, Bristol stool type; GeneralHealth, general health score; HDL, high-density lipoprotein cholesterol; TG, triglyceride levels.

Extended Data Fig. 10b, and Supplementary Table 19). Although we observed 11 significant associations between GalNAc genes and blood triglycerides and high-density lipoprotein in both groups, the effect sizes in the individuals with mucosal A-antigen are higher than in those without ($P_{heterogeneity}$ < 0.05; Extended Data Fig. 10b).

## Discussion

We carried out a genome-wide association study (GWAS) between host genetics and gut microbial SVs in 9,015 individuals from four Dutch cohorts. We found that the human *ABO*-encoded A blood group is strongly associated with a genomic fragment in *F. prausnitzii* harbouring a GalNAc metabolism gene cluster. This association was replicated in a Tanzanian cohort. Strain culture experiments showed that the GalNAc pathway is essential for utilization of GalNAc as a carbohydrate source, which explains the previously observed associations between the *ABO* locus and the relative abundances of *F. prausnitzii* and *C. aerofaciens*.

Several studies have been carried out linking microbial abundance with host genetics in small- or medium-sized cohorts of up to several thousand samples, and genetic effects on microbial abundance were generally found to be small[2–6,27,34–38]. Although several attempts have been made to extend this to microbial functionality level, these analyses were based on the annotations of metabolic pathways, which are far from complete. Our study demonstrates that associations of host genetics with bacterial SVs can help pinpoint putative causal genes and close the gap from species abundance to functionality. Notably, our study included taxonomic abundance as a covariate in the association analyses to identify associations with specific SV regions that are independent of taxa abundance. Our study highlights the importance of moving from taxonomic abundance measurements to bacterial pathways and gene levels for developing a better understanding of the effect of host genetics on the gut microbiome. We have demonstrated this for the *ABO* locus, where the A or AB blood type coded by the *ABO* genotype in FUT2 secretors was associated with bacterial GalNAc gene abundances (lowest *P* = 4.19 × 10⁻²²³) and with an SV region containing the GalNAc pathway in *F. prausnitzii* (*P* = 4.85 × 10⁻⁵¹), whereas no *ABO* association was observed with the abundance of *F. prausnitzii* (*P* = 0.08) in our cohorts.

In addition to ABO, our analysis also yielded 210 suggestive associations at the genome-wide significance level (*P* < 5 × 10⁻⁸), including genetic variants associated with diabetic neuropathy (rs10773589, located close to the *TMEM132D* gene) that affected the presence of an *Anaerostipes hadrus* dSV and variants affecting expression of the *FBLN5* gene (encoding fibulin 5, an extracellular matrix protein that may have a role in bacterial adhesion) that were associated with dSVs of *Collinsella* species.

The association between ABO and the GalNAc pathway was previously observed in a mosaic pig population[39]. In pigs, the GalNAc pathway was identified in *Erysipelotrichaceae* species. However, the abundance of *Erysipelotrichaceae* species in our human cohorts is relatively low,

accounting for only 0.05% of the total community on average. We did not detect any associations between *ABO* and *Erysipelotrichaceae* or their SVs in our human cohorts. Instead, *F. prausnitzii* and *C. aerofaciens* were likely to be the major GalNAc users in the human gut, with 23.1% of *F. prausnitzii* and 81.1% of *C. aerofaciens* assemblies containing the complete GalNAc pathway. Moreover, in contrast to the findings of the study in pigs, in which the association between ABO and the GalNAc pathway was independent of the *FUT2* genotype, the association we observed in humans was strongly dependent on *FUT2* secretor status. Our data also suggest that the presence of GalNAc genes in individuals who are genetically predisposed to have secreted mucosal A-antigen may benefit human health. In addition, we found indications that the GalNAc genes can be made dysfunctional through genomic inversion and that they can be transmitted among bacteria and shared between humans.

The ABO blood group has been associated with various complex diseases and traits in humans, such as venous thromboembolism, lipid levels and other cardiometabolic phenotypes, as well as susceptibility to and severity of many infectious diseases including dengue, malaria and severe acute respiratory syndrome coronavirus 2 infection[40–42]. For example, ABO A blood group has been found to increase the risk of early childhood asthma and *Streptococcus pneumoniae* infection[43]; affect the serum level of ICAM-1, a cell-surface glycoprotein typically expressed on endothelial cells and immune cells[44]; and increase the risk of coronary artery disease[45] and affect circulating levels of cardiovascular-disease-related proteins[46]. The widespread relevance of the *ABO* locus in human health highlights the importance of our human-based microbiome association study. The strong association between *ABO* and bacterial GalNAc-metabolizing genes, and the link of the latter to microbial diversity and richness, support a new hypothesis that *ABO* may affect human health through its effect on the gut microbiome, in addition to already known mechanisms. Given this information, it might be beneficial to increase GalNAc-utilizing strains such as *F. prausnitzii* and *C. aerofaciens* to increase microbial diversity, which could have a beneficial impact on the general health of individuals with mucosal A-antigen. In line with this, our data also showed that bacterial GalNAc gene abundance is positively associated with human health, depending on the presence of mucosal A-antigen.

Our study represents a framework of investigating the crosstalk between our human 'first genome' and microbial 'second genome'. We acknowledge several limitations in our study. First, we focused on the common dSVs and vSVs in gut microbial genomes, assessed on the basis of the abundance and distribution of short reads mapped along bacterial genomes. Our study did not capture other types of SV, such as inversions and translocations, whose comprehensive identification will require whole-genome resequencing and de novo assembly of short or, ideally, long reads. Nonetheless, we could show that genomic inversion could result in dysfunction of the GalNAc pathway. Second, our study did not include other types of genetic variation, such as single nucleotide variants (SNVs), which have great potential impact on bacterial functionality and host–microorganisms interaction. However, analysing genetic associations across the millions of SNVs in the human genome and the hundreds of millions of SNVs in the metagenome would require a much larger sample size. Moreover, functional annotation of SNVs is still challenging. The third limitation of the current study is related to the use of faecal microbiota data to represent the gut microbiome. It is important to note that the microbiome is not entirely the same across the different intestinal compartments, and further investigation into the microbiome of different gastrointestinal tract segments and mucosal layers would provide a more comprehensive landscape of host–microorganisms genetic crosstalk[47]. Fourth, our primary analyses involved only Dutch cohorts, which are very geographically and genetically homogeneous, although we were able to include a Tanzanian replication cohort with a different genetic background, diet and environmental exposure profile. Future work is needed to assess host genetic and microbial genetic associations in more diverse populations to build a better understanding of host–microbiome co-adaptation and co-divergence, as well as to aid in fine-mapping of causal genes.

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

**Lifelines Cohort Study**

**Raul Aguirre-Gamboa[1], Patrick Deelen[1], Lude Franke[1], Jan A. Kuivenhoven[2], Esteban A. Lopera-Maya[1], Ilja M. Nolte[16], Serena Sanna[1], Harold Snieder[16], Morris A. Swertz[1], Peter M. Visscher[16,17], Judith M. Vonk[16] & Cisca Wijmenga[1]**

[16]University of Groningen, University Medical Center Groningen, Department of Epidemiology, Groningen, The Netherlands. [17]Institute for Molecular Bioscience, The University of Queensland, Brisbane, Queensland, Australia.

## Methods

### Cohort description

**DMP.** The DMP consists of 8,719 individuals and is part of the Lifelines study, a multidisciplinary prospective population-based cohort study that utilizes a unique three-generation design to examine health and health-related behaviours in 167,729 people living in the northern Netherlands. Lifelines uses a broad range of investigative procedures to assess the biomedical, socio-demographic, behavioural, physical and psychological factors that contribute to health and disease, with a special focus on multi-morbidity and complex genetics[48].

Microbiome data generation for the DMP was described elsewhere[8]. In brief, fresh-frozen faecal samples were collected from participants of the DMP study. Microbial DNA was isolated using the QIAamp Fast DNA Stool Mini Kit (Qiagen) by the QIAcube automated sample preparation system (Qiagen). Metagenomic sequencing was carried out at Novogene, China using the Illumina HiSeq 2000 sequencer. After filtering, 8,534 DMP samples were used for SV calling.

DMP genotype data generation was described previously[2]. In brief, genotyping was carried out using the Infinium Global Screening Array MultiEthnic Diseases version. Missing genotypes were imputed using Haplotype Reference Consortium (HRC) panel v.1.1 (ref. 49). Only bi-allelic SNPs with imputation quality >0.4, minor allele frequency (MAF) > 0.05, call rate >0.95 and Hardy–Weinberg equilibrium $P$-value > $10^{-6}$ were retained. A total of 7,738 samples had both metagenomic and genotype data after quality control (QC)[2]. We further removed 349 samples overlapping with the LLD cohort. This resulted in phenotype, metagenomic and genotype data being available for 7,389 DMP samples.

**LLD.** The LLD cohort is another part of the Lifelines cohort consisting of 1,539 individuals. Microbiome data generation for LLD was described elsewhere[25]. Fresh-frozen faecal samples were collected, and DNA was isolated with the AllPrep DNA/RNA Mini Kit (Qiagen, catalogue number 80204). Sequencing was carried out using the Illumina HiSeq platform at the Broad Institute, Boston. A total of 1,135 metagenomic samples passed QC.

Genotyping was carried out using the CytoSNP and ImmunoChip assays, as previously described[50], and missing genotypes were imputed using the HRC v.1.1 reference panel[49]. A total of 984 samples had phenotype, metagenomic and genotype data.

**500FG.** The 500FG cohort is part of the Dutch Human Functional Genomics Project (DHFGP) and consists of 534 individuals. The metagenomic data generation was described previously[26,51]. Briefly, DNA was isolated from faecal samples with the AllPrep DNA/RNA Mini Kit, and libraries were sequenced on the Illumina HiSeq 2000 platform. A total of 450 metagenomic samples passed QC and were included in SV calling.

500FG genotype data generation was described previously[52]. Briefly, genotyping was carried out using the Illumina HumanOmniExpressExome-8 v.1.0 SNP chip. Missing genotypes were imputed using the Genome of the Netherlands as a reference panel[53]. After QC, 396 samples had phenotype, metagenomic and genotype data.

**300OB.** 300OB is also part of the DHFGP and consists of 302 individuals with body mass index > 27 kg m$^{-2}$. Metagenomic data generation was described previously[26,54] and was carried out using a similar protocol and analysis pipeline to those of LLD. A total of 302 samples had metagenomic data available for SV calling.

300OB genotype data generation was described previously[55]. In brief, samples were genotyped on the Illumina HumanCoreExome-24 BeadChip Kit or the Illumina Infinium Omni-express chip. Imputation was carried out using the HRC v.1.1 reference panel[49]. After genotype QC, 274 samples had phenotype, genotype and metagenomic data available.

**300TZFG.** For replication in non-European individuals, we included 300TZFG, a population cohort of 323 individuals from both rural and urban areas of Tanzania. This study is part of the DHFGP. Metagenomic data generation has been described previously[28]. Briefly, bacterial DNA was isolated using the AllPrep 96 PowerFecal DNA/RNA kit (Qiagen), and libraries were sequenced on the Illumina NovaSeq 6000 platform. A total of 320 samples passed QC and were available for SV calling.

Host genotype data generation was described previously[56]. In brief, samples were genotyped on the Global Screening Array SNP chip, and genotype imputation was carried out using Minimac4 with the HRC v.1.1 reference panel. After genotype QC, phenotype, genotype and metagenomic data were available for 279 samples.

### QC of metagenomic sequencing data

We removed host-genome-contaminated reads and low-quality reads from the raw metagenomic sequencing data using KneadData (v.0.7.4), Bowtie2 (v.2.3.4.3)[57] and Trimmomatic (v.0.39)[58]. In brief, the data-cleaning procedure included two main steps: raw reads mapped to the human reference genome GRCh37 (hg19) were filtered out; and adapter sequences and low-quality reads were filtered out using Trimmomatic with default settings (SLIDINGWINDOW:4:20 MINLEN:50).

### Taxonomic abundance

We estimated the relative abundance of gut microbial species from the cleaned metagenomic reads using Kraken2 (v.2.1.2)[59] in conjunction with Bracken (v.2.6.2)[60] based on the same reference genomes included in the database of SGV-Finder, and MetaPhlAn 3 (ref. 61) based on the MetaPhlAn database of clade-specific marker genes (mpa_v30). The first of these was used in the GWAS analysis to remove the confounding effect of species abundance, and the last of these was used for the gut microbiome diversity and richness calculation.

### Metagenomic SV detection

SVs are highly variable genomic segments within bacterial genomes that can be absent from the metagenomes of some individuals and present with variable abundance in other individuals. On the basis of the cleaned metagenomic reads, we detected microbial SVs using SGV-Finder with default parameters. SGV-Finder (v.1) was developed and described previously[20] and can detect two types of SV—vSVs and dSVs.

In brief, the SV-calling procedure includes two main steps: resolving ambiguous reads with multiple alignments according to the mapping quality and genomic coverage using the iterative-coverage-based read assignment algorithm and reassigning ambiguous reads to the most likely reference with high accuracy; and splitting the reference genomes of each microbial species into genomic bins and examining the coverage of genomic bins across all samples. For the determination of dSVs within each species, the genomic bins are classified as deleted (coverage close to 0) or retained (coverage close to median coverage of the genome) bins in each sample, and those that are deleted in 25–75% of samples are kept in the analysis as raw dSVs. The raw dSVs that are highly correlated in co-occurrence are further merged into larger SV regions to produce the final dSV profile. For the determination of vSVs within each species, the coverage of genomic bins within each sample is standardized using the $Z$-score approach. Each bin is then assessed across all samples, and those that are highly variable on the basis of a $\beta'$ distribution are kept as raw vSVs. The raw vSVs that are highly correlated in standardized coverage are further merged into large SV regions to produce the final vSV profile.

To define the genes that belong to the SV region, we expanded the genomic coordinates of SVs 1 kb upstream and downstream, with the genes that overlap with the expanded genomic region considered genes that belong to the corresponding SV.

To identify highly variable genomic segments and detect SVs, we used the reference database provided by SGV-Finder, which is based on the proGenomes database (http://progenomes1.embl.de/)[62].

We called SVs using default parameters in a larger panel of 13,195 samples from 10 datasets: 7 population cohorts (HMP1 (ref. 63), HMP2 (refs. 64,65), DMP[8], LLD baseline[25,48], LLD follow-up[22], 500FG[66] and 300TZFG[28]) and 3 disease cohorts (300OB[67], IBD[68] and HIV[69]). This resulted in 10,265 dSVs and 3,931 vSVs. All bacterial species with SV calling were present in at least 75 samples. For the current study, we focused on the four Dutch cohorts for which host genetic data were also available: DMP, LLD baseline, 500FG and 300OB. We removed samples with <5% of SVs called. After sample removal, SV and genotype data were available for 9,015 samples from the four cohorts: DMP ($n$ = 7,372), LLD baseline ($n$ = 981), 500FG ($n$ = 396) and 300OB ($n$ = 266).

### SV filtering and normalization
First, we carried out filtering per cohort. Only SVs that were called in >10% of samples were used in the analyses. In addition, we removed dSVs with a MAF (frequency of either deletion or its absence) <5% and with both reference and alternative allele count ≤80 (this number was determined on the basis of the recommendation that the number of cases and controls is >10× the number of predictors in the generalized linear model association test[70]; see below). Next, we kept only SVs that were present in at least two cohorts. vSV data were normalized using inverse normal rank transformation for the heritability and association analyses.

### Heritability estimation
We estimated SV heritability using the GREML software from the GCTA toolbox (v.1.94.1). We applied the family-based approach[71] implemented in GREML on the SV data from the DMP cohort because this cohort has the largest sample size and contains relatives. A total of 7,389 samples with genotype and microbiome data were used for the analysis. To estimate heritability, we used default settings correcting for age, sex, total metagenomic sequencing read number and species abundance. Heritability estimates for species abundance and the corresponding confidence intervals were obtained from ref. 8, which estimated heritability on the basis of family relations in the same DMP cohort.

### GWAS and meta-analysis
The manipulation of human genotype datasets was conducted using PLINK (version alpha 2.1). Association analysis was carried out using fastGWA from the GCTA toolbox (v.1.94.1)[72], per cohort per SV. For dSVs, we used the generalized linear mixed model-based version of the tool (--fastGWA-mlm-binary)[73]. In the association analyses, we used a sparse genetic relationship matrix (GRM) created from the full GRM built on genotyped (non-imputed) SNPs with MAF > 5% using GCTA with default options (--make-grm and --make-bK-sparse 0.05). The following covariates were added to the model: age, sex, total metagenomic sequencing read number and centred log ratio (CLR)-transformed species abundance. The total read count was standardized to have a mean of zero and a variance of one. Meta-analysis was carried out using the Metal software (version 2020-05-05)[74] with default options (weighting cohort-based $P$ values according to sample size). To control for multiple testing, we applied the Bonferroni-corrected genome-wide significance threshold ($5 \times 10^{-8}$/SV number) and considered association results with $P$ values below this threshold as statistically significant. For dSVs, the $P$-value threshold was $5 \times 10^{-8}$/1,666 = $3.00 \times 10^{-11}$. For vSVs, it was $5 \times 10^{-8}$/1,886 = $2.65 \times 10^{-11}$.

### Association with ABO blood group
We used two approaches to determine the ABO blood group. In the DMP cohort, we determined the blood group on the basis of three variants (rs8176719, rs41302905 and rs8176746), as described previously[2]. For LLD and 500FG, in which some of these variants were not genotyped, we used a less sensitive approach based on two SNPs, rs8176693 (T allele determines blood group B) and rs505922 (T allele determines blood group O), as reported in previously published papers[75,76]. Association

of blood groups with *F. prausnitzii* SVs was carried out in R (v.4.1.0) using (generalized) linear mixed models using the R package lme4qtl (v.0.2.2). This package allows a kinship matrix to be included as a random effect to account for sample relatedness. For each cohort, we created a kinship matrix based on a GRM built by GCTA using the function kinship from the R package kinship2 (v.1.9.6). We corrected for the same covariates as in the GWAS as described above. Meta-analysis was carried out using Metal[74].

### Population genetic structure of *F. prausnitzii*
We calculated an SV-based between-sample microbial genetic dissimilarity based on Canberra distance for each microbial species separately using the vegdist() function of the R package vegan (v.2.6-2) to generate species-specific genetic distance matrices ($M_{SV}$). We then carried out a principal coordinate analysis based on $M_{SV}$ using the pcoa() function of the R package ape (v.5.6-2), with the negative eigenvalues corrected with Cailliez's method[53].

### Phylogenetic tree construction
For the *F. prausnitzii* strains with SVs containing the GalNAc utilization gene cluster, we first constructed a phylogenetic tree using the RAxML approach based on 81 accurately selected single-copy marker genes[77]. We then constructed another phylogenetic tree using RAxML (v.8) based on the GalNAc utilization genes located in the SV region[78]. The phylogenetic trees were converted to between-strain cophenetic distances using the cophenetic() function from the R package stats (v.4.3.0).

The phylogenetic tree shown in Fig. 3c was constructed using CSI Phylogeny 1.4 on the basis of SNPs of whole-genome sequences of the 12 isolates[79] and was visualized using the R packages ggtree (v.3.2.1) and gggenomes (v.0.9.9.9000)[80].

### Cohousing and SV sharing
Cohousing information at the time of faecal sampling is known for 8,880 individuals from the DMP cohort. For this cohort, we removed individuals not cohousing with any other participant and those with no microbial or genetic information. For 2,631 participants, we assessed whether any individual cohousing with them at the time of sampling had *F. prausnitzii* 577–579. We then used a logistic regression using the presence or absence of 577–579 as a dependent variable and the secretion of A-antigens and the presence of household SV as independent variables to estimate the effect of the presence of SV in the household on SV presence in an individual. We also assessed the possible gain or loss of *F. prausnitzii* in 338 individuals whose gut microbiome was profiled again after 4 years[22]. For 119 individuals, *F. prausnitzii* SV profiles were generated at both time points.

### Genomic island prediction
Genomic islands were predicted by SIGI-HMM[81] and IslandPath-DIMOB[82] as integrated into IslandViewer 4, a computational tool that integrates multiple genomic island prediction methods[83]. Both SIGI-HMM and IslandPath-DIMOB have been shown to have high overall accuracy, with IslandPath-DIMOB having a slightly higher recall and SIGI-HMM having a slightly higher precision.

### Microbial gene annotation
The genes of *F. prausnitzii* strains and reference genomes used for gut microbial SV calling were annotated using MicrobeAnnotator (v.2.0.5)[84] and Bakta (v.1.8.1)[85]. For the annotation of genes encoding glycoside hydrolase family 109 (GH109) in *F. prausnitzii* and *C. aerofaciens* strains, we first obtained 2,113 GH109 protein sequences from CAZy (http://www.cazy.org/GH109_characterized.html)[86] and then conducted a homologue search of GH109 genes in the genomes of *F. prausnitzii* and *C. aerofaciens* strains using tblastn (v.2.5.0+)[87] with the following parameters: -outfmt 7 -evalue 1e-10.

## Homologue search in genes involved in the GalNAc pathway

We downloaded 10,487 assembled genomes of *ABO*-associated species from the Unified Human Gastrointestinal Genome collection[33], including 1,103 assemblies of *C. aerofaciens*, 484 of *F. lactaris*, 1,109 of *B. bifidum* and 7,791 of *F. prausnitzii*. We then used the sequences of genes located in SV 577–579 as queries and carried out a homologue search in the assemblies using tblastn (v.2.5.0+)[87] with the following parameters: -outfmt 7 -evalue 1e-10.

## Protein family search and profiling with shortBRED

We searched the metagenomes for 27 bacterial proteins identified in the SV segment of *F. prausnitzii* (excluding *dinB* and *HTF-238_O2530*, which were used as SV region markers and are not located within the SV), including the genes known to be involved in GalNAc metabolism, using the shortBRED toolkit (v.0.9.5)[88]. We extracted the genes located in the SV and converted the gene sequences to protein sequences, as required by shortBRED. We used the shortBRED tool shortbred_identify.py (v.0.9.5) to identify unique markers for the query genes, using the UniRef90 database (downloaded on 1 November 2021) as a negative control.

Next, the shortbred_quantify.py tool (v.0.9.5) was used to quantify these markers in metagenomes. First, we assessed the association of these gene abundances with the ABO blood group. We log-transformed the RPKM values provided by shortBRED and carried out a linear mixed model analysis using shortBRED gene abundances as outcomes and ABO A or AB blood group as a predictor accounting for sample relatedness using random effects in the lme4qtl package. We also included other covariates as predictors, including age, sex, total metagenomic sequencing read number and CLR-transformed *F. prausnitzii* abundance, together with four *F. prausnitzii* dSVs and one vSV found to be associated with *ABO* in the primary GWAS analysis.

Next, we estimated the association of gene abundance with the α-diversity (Shannon index and richness) of the gut microbiome in DMP using linear regression using the following formula:

α-diversity = SV 577–579 + *F. prausnitzii* taxonomic abundance + *C. aerofaciens* taxonomic abundance + gene abundance.

## Bacterial strains and growth

The *Faecalibacterium* and *Collinsella* strains used in this study were from culture collections (ATCC and DSMZ) and our local strain collection (Department of Medical Microbiology, University Medical Center Groningen, Groningen, the Netherlands). On the basis of the presence or absence of SVs, the following *Faecalibacterium* strains were selected: *F. prausnitzii* A2-165 (DSM 17677), *F. prausnitzii* ATCC 27768, *F. prausnitzii* HTF-F (DSM 26943), *F. prausnitzii* HTF-112, *F. prausnitzii* HTF-495, *F. prausnitzii* HTF-238, *F. prausnitzii* HTF-383, *F. prausnitzii* 60C2, *F. prausnitzii* HTF-121, *F. prausnitzii* HTF-133, *F. prausnitzii* HTF−441 and *F. prausnitzii* FM4. Two strains of *C. aerofaciens* were selected on the basis of the presence of the GalNAc genes: *C. aerofaciens* 4PBA and *C. aerofaciens* HTF-129.

Strains were cultured in a modified YCFA medium supplemented with different carbohydrates (glucose, galactose, GalNAc, mannose, lactose, fructose, *N*-acetylglucosamine, 2-fucosyllactose and *N*-acetylneuraminic acid). YCFA medium was prepared as for YCFA–glucose (YCFAG) medium described before[89] without the addition of glucose. YCFA medium was composed of (g l$^{-1}$) 10 casitone, 2.5 yeast extract, 4 sodium bicarbonate, 0.45 dipotassium hydrogen phosphate, 0.45 potassium dihydrogen phosphate, 0.9 sodium chloride, 0.09 magnesium (II) sulfate heptahydrate, 0.12 calcium chloride dihydrate, 2.7 sodium acetate, 1 cysteine, 5 ml 0.02% resazurin and 0.2% haemin, 1 ml pink vitamin mixture and yellow vitamin mixture, and the liquid medium. The pink vitamin mixture (per 100 ml) contains 1 mg biotin, 1 mg cobalamin, 3 mg *p*-aminobenzoic acid, 5 mg folic acid and 15 mg pyridoxamine. The yellow vitamin mixture (per 100 ml)

contains 5 mg thiamine and 5 mg riboflavin. The liquid medium includes 600 µl l$^{-1}$ propionate (≥99% purity, Sigma-Aldrich), 100 µl l$^{-1}$ isobutyrate (≥99% purity, Sigma-Aldrich), 100 µl l$^{-1}$ isovalerate (≥99% purity, Sigma-Aldrich) and 100 µl l$^{-1}$ valerate (≥99% purity, Sigma-Aldrich). The medium is adjusted to a final pH of 6.5.

Growth experiments were carried out in a Bactron 600 anaerobic incubator (Kentron Microbiome BV) using a 24-well flat-bottom-plate with total volume of 1 ml per well YCFA medium supplemented with 4.5 g l$^{-1}$ of the desired carbohydrate source. Cultures were started at an initial OD$_{600nm}$ range of 0.10–0.15 by the addition of an overnight glucose-grown pre-culture, and growth was monitored anaerobically at 600 nm over 24 h at 37 °C. Readings were taken every 2 h, after 10 s shaking, using Epoch 2 (Agilent BioTek Instruments), and growth curves were generated using Gen5 software. Each growth condition was carried out in triplicate using three independent pre-cultures. Data of growth curves are reported as means ± s.d.

## Gene expression analysis of GalNAc induction

**Sample collection.** The *F. prausnitzii* strains HTF-495, HTF-441 and ATCC 27768 were selected to test the mRNA expression level of genes on the basis of the shortest distance within the phylogenetic tree. The *F. prausnitzii* strains were pre-cultured individually in YCFAG medium overnight anaerobically at 37 °C in triplicate. To get enough biomass, these pre-cultures were used to inoculate fresh triplicates of each strain in a ratio of 1:20 (20 ml) and incubated for 24 h anaerobically at 37 °C in YCFAG medium. Each culture was then split into two tubes (10 ml per tube) and centrifuged at 3,000 r.p.m. for 10 min. The supernatants were removed and resuspended with 10 ml YCFAG or YCFA-GalNAc, separately for each culture, in a total of 18 samples. After 6 h of incubation, a 1:1 ratio (10 ml) of ice-cold killing buffer (20 mM Tris-HCl pH 7.5, 5 mM MgCl$_2$, 20 mM NaN$_3$) was added to the cultures. Samples were centrifuged at 3,000 r.p.m. for 10 min at 4 °C, and the supernatants were removed. The pellets were resuspended in 1 ml TRIzol (Invitrogen) and stored at −80 °C until further RNA isolation.

**RNA isolation and cDNA synthesis.** For RNA isolation, 200 µl of RNAse-free chloroform was added to each sample and incubated at room temperature for 5 min. After incubation, the samples were centrifuged at 12,000*g* at 4 °C, and the aqueous phase was recovered into a new tube. To precipitate RNA, 500 µl of RNAse-free isopropanol was added to each sample and mixed briefly. Samples were incubated for 10 min at room temperature and centrifuged for 10 min at 12,000*g* and 4 °C. The supernatant was removed, and the pellets were washed in 1 ml of 75% RNAse-free ethanol, vortexed briefly and centrifuged for 5 min at 7,500*g* at 4 °C. The supernatant was removed, and the pellets were air-dried at room temperature for 10 min. Afterward, the samples were resuspended with RNAse-free water.

Finally, DNA contamination was removed from 10 µg of the sample using TURBO DNA-free Kit (Invitrogen). cDNA was generated using the TaqMan Reverse Transcription Reagents (Invitrogen) with random hexamers.

**Quantitative PCR.** Samples were diluted to working concentration and used as a template for quantitative PCR (qPCR) amplification of the target genes (for primers, see Supplementary Table 20). Each reaction contained 10 µl of GoTaq qPCR Master Mix (Promega), 9 µl of DNA template (10 ng) and two times 0.5 µl primer solution (20 µM) in a total reaction volume of 20 µl. The amplification was carried out in a 7500 Real-Time PCR System (Applied Biosystems). The amplification program comprised two stages: an initial denaturation step at 95 °C for 2 min, followed by 40 two-step cycles at 95 °C for 15 s and at 60 °C for 1 min. At the end of the run, a melting curve analysis was carried out. The cycle threshold ($C_t$) value was first determined using the 7500 Real-Time PCR System detection system and then adjusted manually to set the threshold within the exponential phase of the curves. All qPCR

reactions were carried out in triplicate. The $\Delta C_t$ values of the genes of interest were obtained by correction for the $C_t$ value of *rpoA* as the housekeeping gene. Afterward, the different $2^{-\Delta C_t}$ values of each strain were calculated per condition. These values were used to determine the relative fold change expression of the genes after GalNAc induction compared to growth in glucose.

**Ethical approval.** The Lifelines study was approved by the ethics committee of the University Medical Center Groningen (METc2007/152). All participants signed an informed consent form before enrolment. Additional written consents were signed by the DMP participants or legal representatives for children aged under 18 years. The LLD study was approved by the Institutional Ethics Review Board of the University Medical Center Groningen (ref. M12.113965), the Netherlands. The 300OB study was approved by the IRB CMO Regio Arnhem-Nijmegen (number 46846.091.13). The 500FG study was approved by the Ethical Committee of Radboud University Nijmegen (NL42561.091.12, 2012/550). The inclusion of volunteers and experiments was conducted according to the principles expressed in the Declaration of Helsinki. All volunteers gave written informed consent before any material was taken. The 300FGTZ study was approved by the Ethical Committees of the Kilimanjaro Christian Medical University College (CRERC; number 936) and the National Institute for Medical Research (NIMR/HQ/R.8a/Vol. IX/2290) in Tanzania. The Tanzanian cohort provided consent for the use of their data for the purposes of this analysis.

### Reporting summary

Further information on research design is available in the Nature Portfolio Reporting Summary linked to this article.

## Data availability

The profile of SVs of all samples and the full summary statistics of genetic associations with bacterial dSVs and vSVs are available at https://doi.org/10.25452/figshare.plus.c.6877849. The assembled bacterial genomes from the growth experiment are available at the National Center for Biotechnology Information (NCBI) with accession number PRJNA1024432. The raw metagenomic sequencing data of all four cohorts are publicly available. The data for three are deposited at the European Genome–Phenome Archive: DMP (accession number EGAS00001005027), LLD (accession number EGAD00001001991) and 300OB (accession number EGAD00001005083). The 500FG data are available at the NCBI Sequence Read Archive under accession number PRJNA319574. The metagenomic data of 300TZFG are available in the NCBI BioProject database under accession number PRJNA686265. To protect participant's privacy and respect the research agreements in the informed consent, genotyping data and participant metadata are not publicly available and cannot be deposited in public repositories. The DMP and LLD data can be accessed by all bona fide researchers with a scientific proposal by contacting the Lifelines Biobank (instructions at https://www.lifelines.nl/researcher/how-to-apply). Researchers will need to fill in an application form, which will be reviewed within 2 working weeks. If the proposed research complies with Lifelines regulations (for example, noncommercial use and guarantee of participants' privacy), researchers will then receive a financial offer and a data and material transfer agreement to sign. In general, data will be released within 2 weeks after signing the offer and data and material transfer agreement. The data will be released in a remote system (the Lifelines workspace) running on a high-performance computer cluster to ensure data quality and security. As Lifelines is a non-profit organization dependent on (governmental) subsidies, a fee is required to cover the costs of controlled data access and supporting infrastructure. The fee for data access on the high-performance computer is €3,500 for 1 year and the fee for the Lifelines Workspace environment is €4,500 for 1 year, or less for shorter periods of time. There are no restrictions on the downstream re-use of aggregated, non-identifiable results (as approved by Lifelines), nor are there authorship requirements, but Lifelines does request that it is acknowledged in publications using these data. The data access policy, data access fees and an example Data and Material Transfer Agreement (which includes details on how to acknowledge the use of Lifelines data in publications) are described in detail at https://www.lifelines.nl/researcher/how-to-apply. Note that data access for replication can be arranged through Lifelines. Lifelines will not charge an access fee for controlled access to the full dataset used in the manuscript (including phenotype and sequencing data), for the specific purpose of replication of the results presented in this Article or for further assessment by the reviewers, for a period of three months. Researchers interested in such a replication study or review assessment can contact Lifelines at research@lifelines.nl. The genotype and metadata of the 500FG, 300OB and 300TZFG cohorts can be requested through the Human Functional Genomics Data Access Committee (Martin.Jaeger@radboudumc.nl). There are no conditions associated with their use, with the exception of those associated with data that may lead to compromising participant confidentiality, such as raw genomics data. The data are freely available, and no agreement or costs are required. The applicants would receive a response within 4 weeks from application. Gut microbial SV calling was conducted on the basis of reference microbial genomes from the proGenomes database (http://progenomes1.embl.de/). ShortBRED analysis was carried out on the basis of the UniRef90 database (https://ftp.uniprot.org/pub/databases/uniprot/uniref/uniref90/). Source data are provided with this paper.

## Code availability

The code for statistical analysis and visualization is available through https://doi.org/10.5281/zenodo.10018199.

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

**Acknowledgements** We thank all of the volunteers in the Lifelines cohort (https://www.lifelines.nl/) and the Human Functional Genomics Project (http://www.humanfunctionalgenomics.org/) for their participation and the project staff for their help and management; K. McIntyre for critical reading and editing; J.M. van Dijl for valuable discussions on the GalNAc induction experiment; and the Genomics Coordination Center for providing data infrastructure and access to high-performance computing clusters. The generation and management of GWAS data for the Lifelines Cohort Study is supported by the University Medical Center Groningen Genetics Lifelines Initiative. This study is supported by Netherlands Organization for Scientific Research (NWO)-VICI grant VI.C.202.022 (J.F.), NWO-VIDI grant 016.178.056 (A.Z.), NWO-VENI grant 194.006 (D.V.Z.), NWO-VENI grant 222.016 (D.W.), European Research Council (ERC)-Consolidator grant 101001678 (J.F.), ERC Starting Grant 715772 (A.Z.) and Dutch Heart Foundation grant IN-CONTROL (CVON2018-27 to J.F., A.Z., M.G.N., L.A.B.J., J.H.W.R. and N.P.R.). In addition, C.W. and J.F. are supported by the Netherlands Organ-on-Chip Initiative, an NWO Gravitation project (024.003.001) funded by the Ministry of Education, Culture and Science of the government of the Netherlands. J.F. is supported by the AMMODO Science Award 2023 for Biomedical Sciences from Stichting Ammodo. A.Z. is further supported by the NWO Gravitation grant Exposome-NL (024.004.017), and EU Horizon Europe Program grant INITIALISE (101094099). S.S. is supported by Next Generation EU grant Project Age-It (DM MUR 1557 11.10.2022), ERC Starting Grant 2022 (101075624) and NutrAGE grant (DM MUR 844 16.07.2021). R.K.W. is supported by the Seerave Foundation and NWO. L.L. is supported by a joint fellowship from the University Medical Center Groningen and China Scholarship Council (CSC) with grant number CSC201908320432. Y.Z. is supported by a joint fellowship from the University Medical Center Groningen and CSC with grant number CSC202006170040. N.P. is supported by a grant of the Graduate School of Medical Sciences of the University of Groningen, the Netherlands. H.P. is supported by a joint fellowship from the University Medical Center Groningen and CSC with grant number CSC202208060107. The 300TZ cohort received financial support from the Joint Programming Initiative, A Healthy Diet for a Healthy Life (JPI-HDHL; project 529051018, TransMic) and ZonMw (the Netherlands Organization for Health Research and Development). Figures 1a and 3b were created with BioRender.com, with publication licences XX263AR9Z1 and BQ263ARFK2, respectively.

**Author contributions** J.F. and H.J.M.H. conceptualized the study. D.V.Z., D.W., S.A.-S., Y.Z. and H.P. carried out data analysis: D.V.Z. for genetic association; D.W. for SV profiling, annotation and microbiome analysis; S.A.-S. for homologue search; Y.Z. for gene abundance profiling; and H.P. for the genomic island searching. R.G. processed metagenomic sequencing data in the DMP. E.A.L.-M. and S.S. processed human genetics data in the DMP. D.W., L.L. and A.J.R.-M. annotated bacterial genes. L.L. carried out the strain culture and growth assay experiments. L.L., N.P. and Á.D.C.-I. carried out the gene expression analysis. S.S.v.L. aided in interpreting the results. C.W., J.F. and A.Z. set up the LLD cohort. C.W., J.F., A.Z. and R.K.W. set up the DMP. L.A.B.J., N.P.R., J.H.W.R. and M.G.N. set up the 500FG and 300OB cohorts. G.S.T., V.I.K., R.J.X. and Q.d.M. set up the 300TZFG cohort. D.V.Z., D.W., L.L., H.J.M.H. and J.F. drafted the manuscript. All authors reviewed and edited the manuscript.

**Competing interests** H.J.M.H. in the past received a research grant from Chr. Hansen A.G., Denmark. R.K.W. acted as a consultant for Takeda, received unrestricted research grants from Takeda, Johnson & Johnson, Tramedico and Ferring, and received speaker fees from MSD, Abbvie and Janssen Pharmaceuticals. All other authors declare no competing interests.

**Additional information**
**Correspondence and requests for materials** should be addressed to Hermie J. M. Harmsen or Jingyuan Fu.

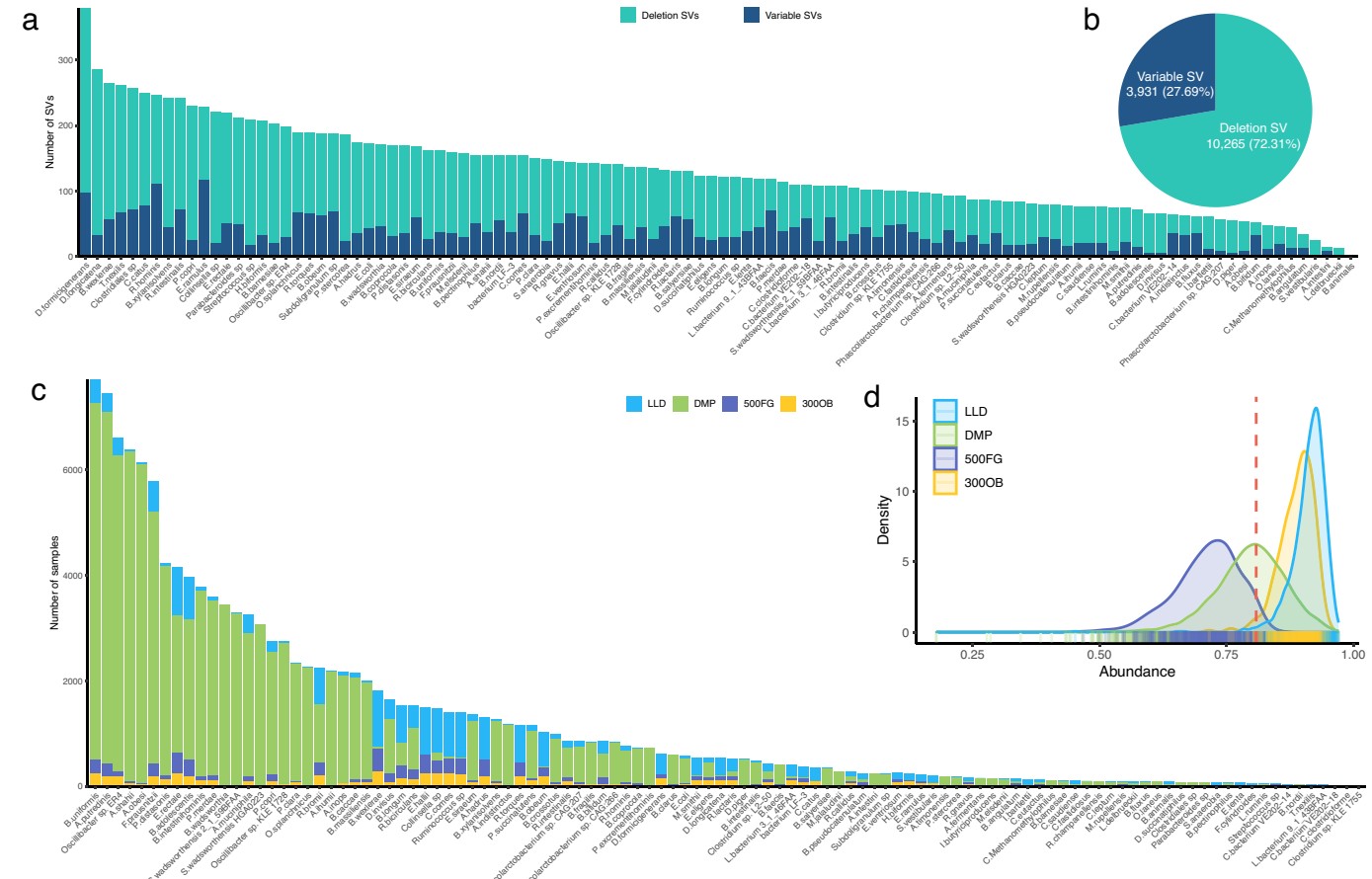

**Extended Data Fig. 1 | Overview of gut microbial SVs detected in the Dutch cohorts before filtering. a**, Number of dSVs and vSVs detected per species. Each bar represents a species, and the y-axis shows the number of SVs detected per species. The number of dSVs and vSVs are colored in green and blue, respectively. **b**, Percentages of the overall number of dSVs and vSVs detected. Pie chart shows the number of dSVs and vSVs and the corresponding proportions. **c**, Number of samples with detected SVs per species, colored by cohort. X-axis indicates the different species. Y-axis indicates the number of samples with detectable SVs in that species, colored by different cohorts. **d**, Distribution of the total relative abundance of species detected with SVs per cohort. X-axis is the total abundance in terms of proportion of a microbial community. Y-axis is density. The red dashed vertical line indicates the average total relative abundance of species with sufficient coverage to detect SVs in all Dutch samples.

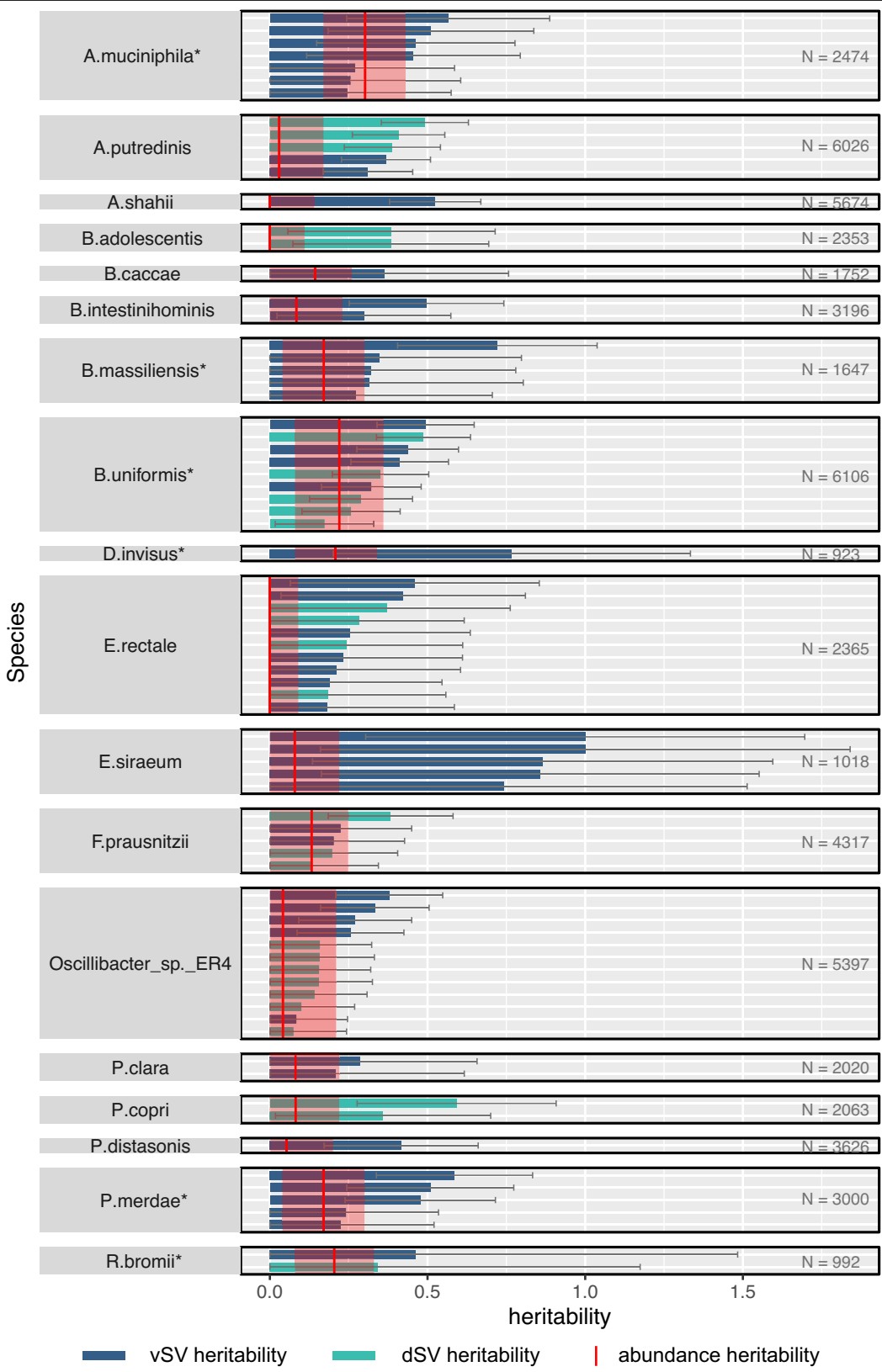

**Extended Data Fig. 2 | Heritability of gut microbial SVs and corresponding species abundance.** Bar height indicates heritability $h^2$ values for vSVs (blue) and dSVs (cyan) that are nominally significant ($P < 0.05$), estimated using GCTA. Error bars in gray are the 95% confidence intervals (1.96 × standard error) of the estimated heritability. Solid red vertical lines indicate the heritability of the corresponding species abundance previously reported in Gacesa et al. *Nature* (2020), and the red opaque rectangles indicate the corresponding 95% confidence interval. Species names marked with * indicate those with significant abundance heritability at $P < 0.05$ level. The number of samples used for SV heritability estimation is given on the right.

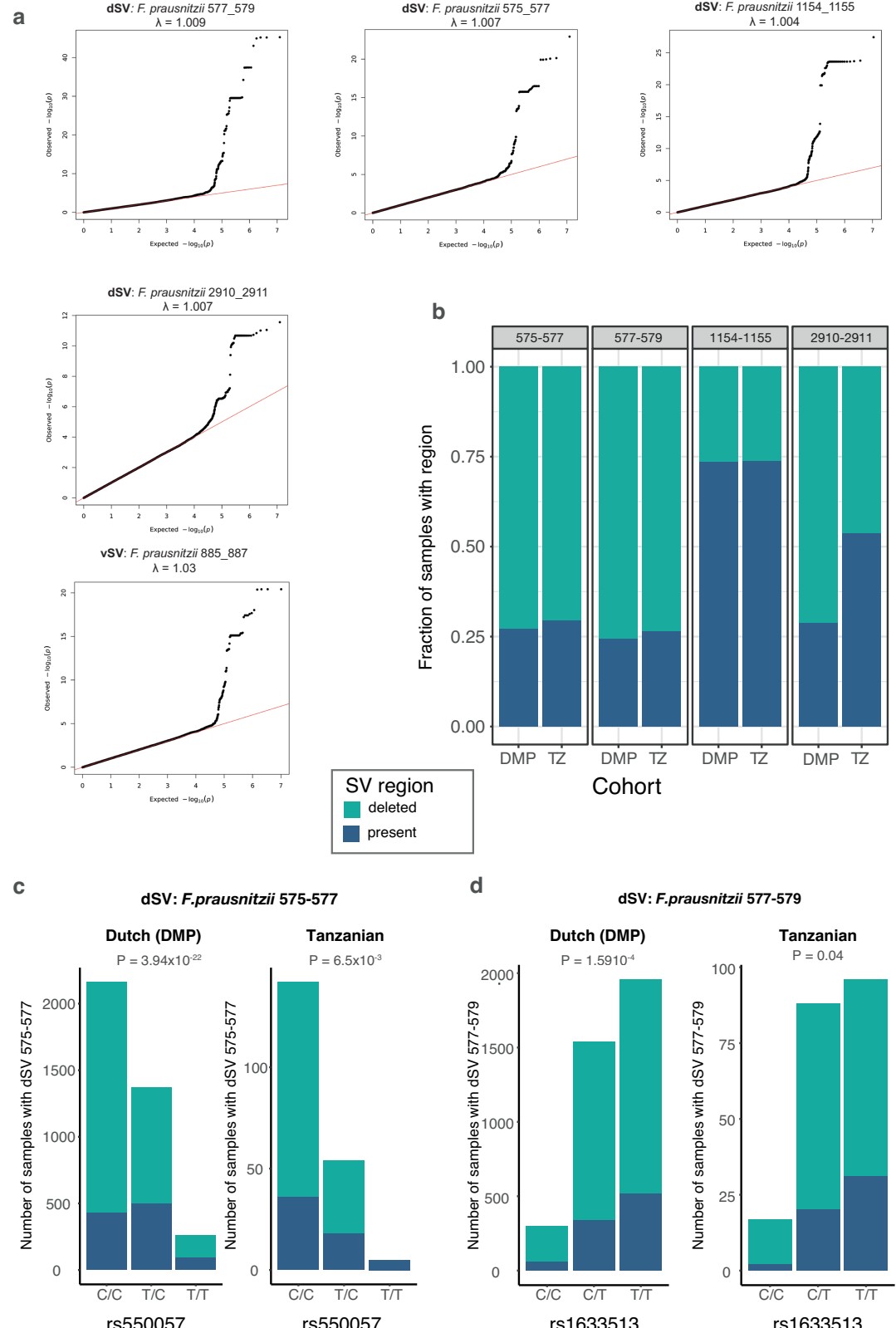

**Extended Data Fig. 3** | See next page for caption.

**Extended Data Fig. 3 | Comparison of *F. prausnitzii* dSV associations between the Dutch and Tanzanian cohorts. a**, Q-Q plots of the associations between human SNPs and *F. prausnitzii* SVs in the Dutch cohorts. X-axis is the expected log-transformed P-value. Y-axis is the observed log-transformed unadjusted P-value derived from the linear mixed model. Each dot represents an association P-value. The diagonal is represented by the solid red line. The lambda ($\lambda$) value represents the genomic inflation factor. **b**, Comparison of the presence/absence ratio of dSVs between the DMP cohort (N = 5044) and the Tanzanian cohort (N = 201). Bar plots show the fraction of samples with (dark blue) or without (cyan) the dSV region. **c**, Association of *F. prausnitzii* dSV 575–577 with rs550057 in both the Dutch and Tanzanian cohorts. Bar plots show the number of samples with (dark blue) and without (cyan) the dSV region per genotype. The *P*-value indicates the unadjusted association significance assessed by fastGWA using linear mixed models. **d**, Association of *F. prausnitzii* 577–579 with rs1633513 in both cohorts. Bar plots show the number of samples with (dark blue) and without (cyan) the dSV region per genotype. The *P*-values indicate the unadjusted association significance assessed by fastGWA based using linear mixed models.

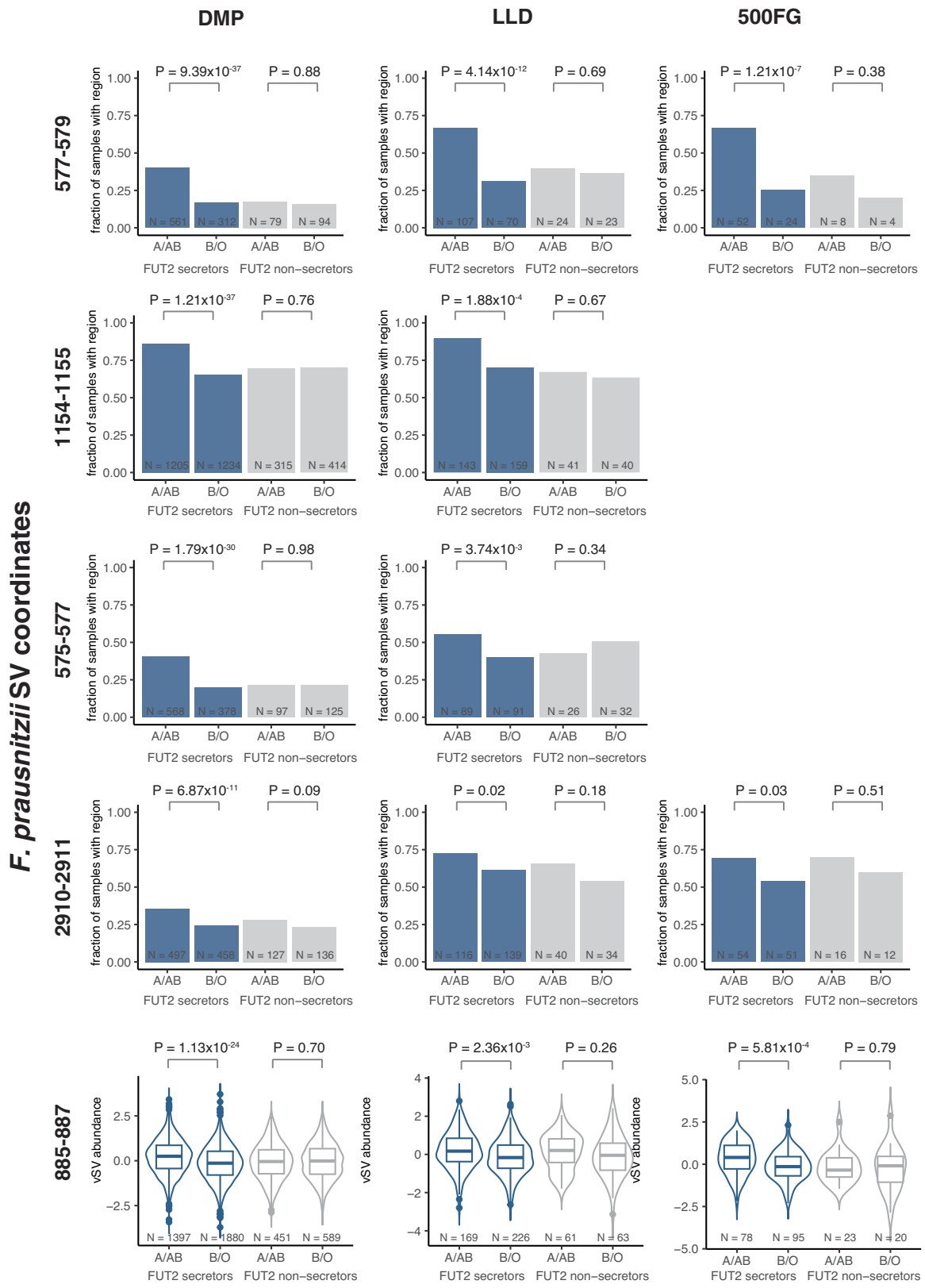

**Extended Data Fig. 4 | The associations of *F. prausnitzii* SVs with ABO blood groups depend on FUT secretor status.** Each row of panels represents a different SV region. Each column of panels represents a different cohort. The associations of dSVs are visualized using bar plots where the y-axis shows the fraction of samples with the dSV region. Number of samples for each group is given at the bottom of each bar. The associations of vSVs are visualized using violin plots where the y-axis shows the standardized coverage of vSVs. Violin plots show density distribution, whereas the inner boxplots represent summary statistics: the center line is the median, the box hinges are the lower and upper quartiles of the distribution, the whiskers extend no further than 1.5× interquartile range from the hinges, and data beyond the end of the whiskers are outliers plotted as individual points. The x-axis of all plots shows the ABO blood group (A/AB or B/O) split by *FUT2* status. Unadjusted *P*-values show the significance of the association estimated using linear mixed models.

**a**

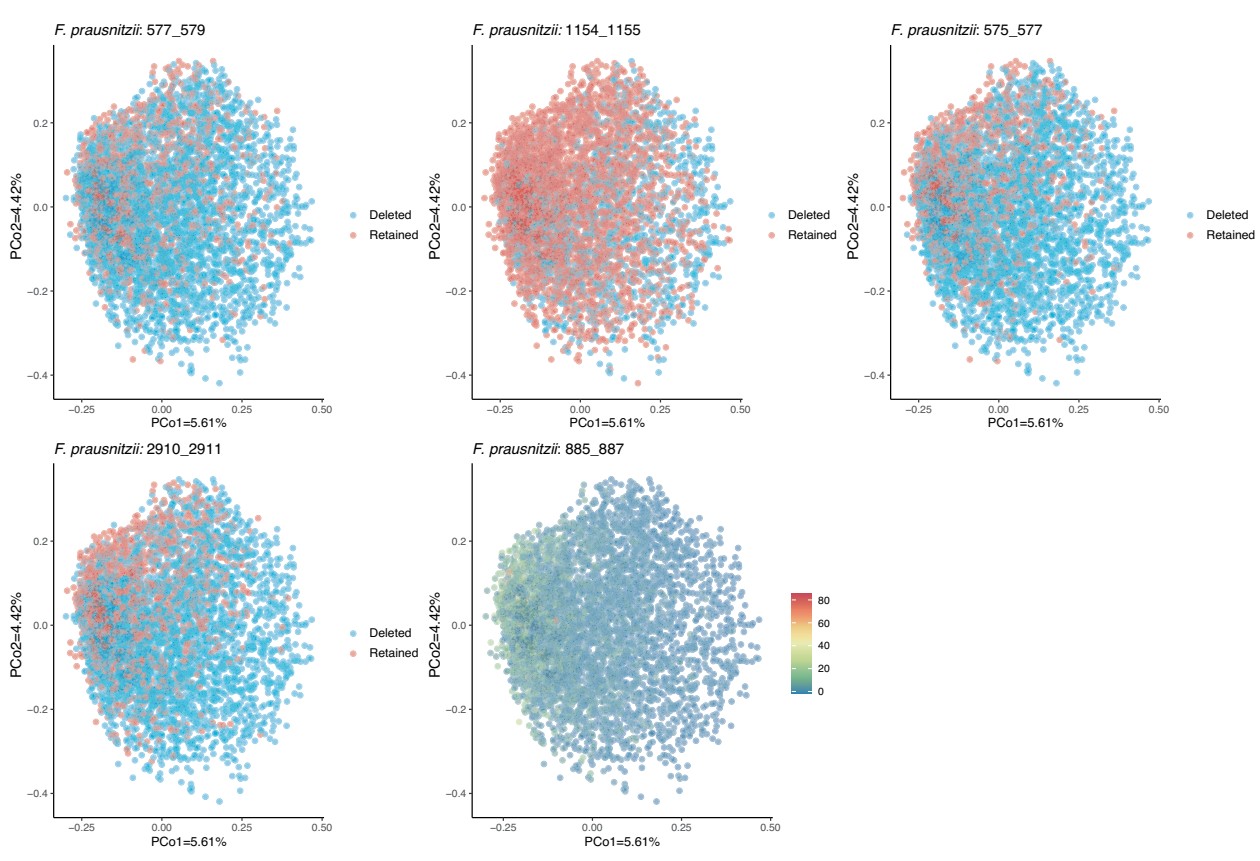

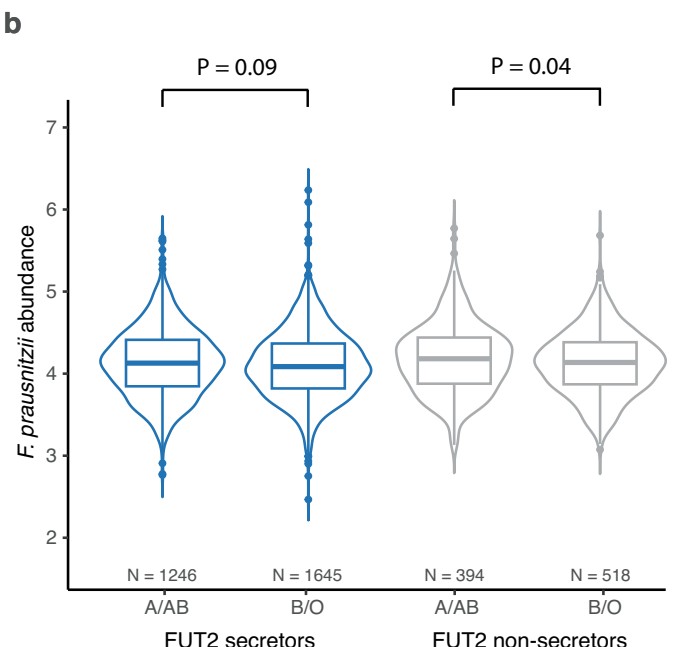

**b**

**Extended Data Fig. 5** | See next page for caption.

**Extended Data Fig. 5 | Population genetic structure of *F. prausnitzii* and its association with ABO-associated SVs. a**, The x-axis and y-axis show the top two PCs. Each dot represents a sample. In each panel figure, dots are colored differently based on the presence/absence status of dSVs or the standardized coverage of vSVs following the color key shown in the figure. The significance of associations between population genetic structure of *F. prausnitzii* and SVs was estimated by permutational multivariate analysis of variance (PERMANOVA), and all associations for the five ABO-associated SVs are significant (unadjusted $P < 0.05$). **b**, Association of *F. prausnitzii* abundance with ABO blood groups in the DMP cohort. Species abundance (CLR-transformed) of *F. prausnitzii* (y-axis) plotted against ABO-encoded blood groups for all DMP individuals (x-axis). Violin plots show density distribution, whereas the inner boxplots represent summary statistics: the center line is the median, the box hinges are the lower and upper quartiles of the distribution, the whiskers extend no further than 1.5× interquartile range from the hinges, and data beyond the end of the whiskers are outliers plotted as individual points. Unadjusted P-value reflects the significance of association of ABO blood groups (A/AB vs B/O) with *F. prausnitzii* abundance after adjusting for covariates assessed using linear mixed models.

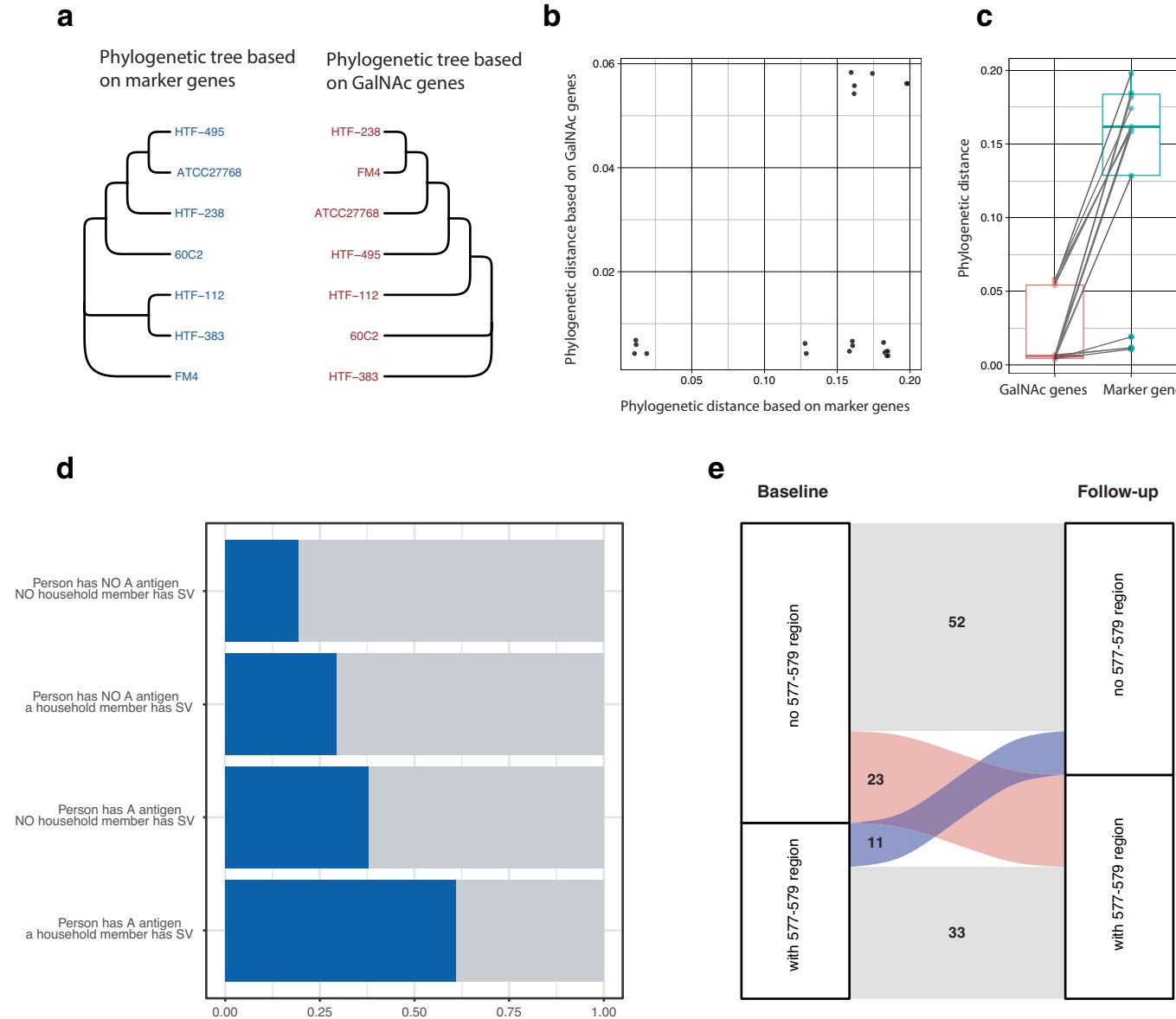

**Extended Data Fig. 6 | Transmission of *F. prausnitzii* SVs. a**, Different phylogenetic distances between *F. prausnitzii* strains based on marker genes and GalNAc genes. Phylogenetic trees based on the phylogenetic distance of marker genes (left) and GalNAc genes (right). **b**, Scatter plot of between-strain genetic distance based on marker genes (x-axis) and GalNAc genes (y-axis). Each dot represents pairwise phylogenetic distance between two strains. **c**, Comparison of between-strain genetic distance based on marker genes and GalNAc genes. Each box plot represents the distribution of pairwise phylogenetic distances of strains based either on GalNAc genes or on species marker genes. The center line of the box is the median, the box hinges are the lower and upper quartiles of the distribution, and each dot represents a pair of two strains

(npair = 21). **d**, Fraction of samples with or without SV 577–579 dependent on household members. Each blue bar represents the fraction of samples with the GalNAc-containing 577–579 dSV region in individuals with/without A-antigen and with/without household members with the dSV. **e**, Gain or loss of the *F. prausnitzii* dSV 577–579 region after 4 years. Alluvial plot shows the number of individuals with and without the 577–579 region in *F. prausnitzii* at two time-points 4-years apart. The number of individuals with the same presence or absence status for this region after 4 years is indicated in gray. The number of individuals who gained the region is indicated in red. The number of those who lost the region is indicated in blue.

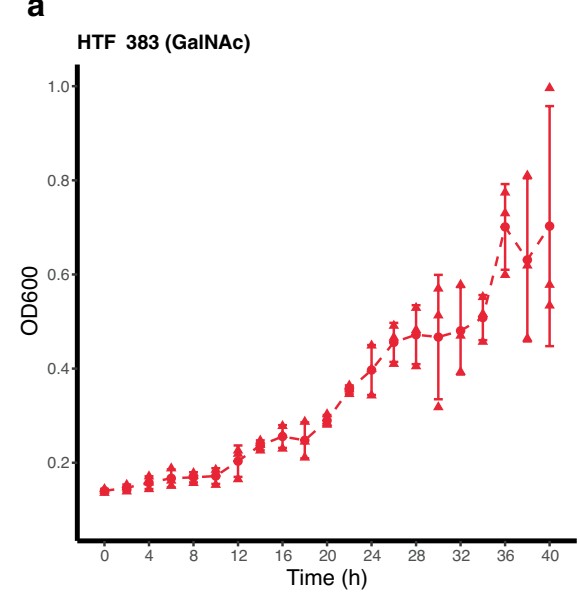

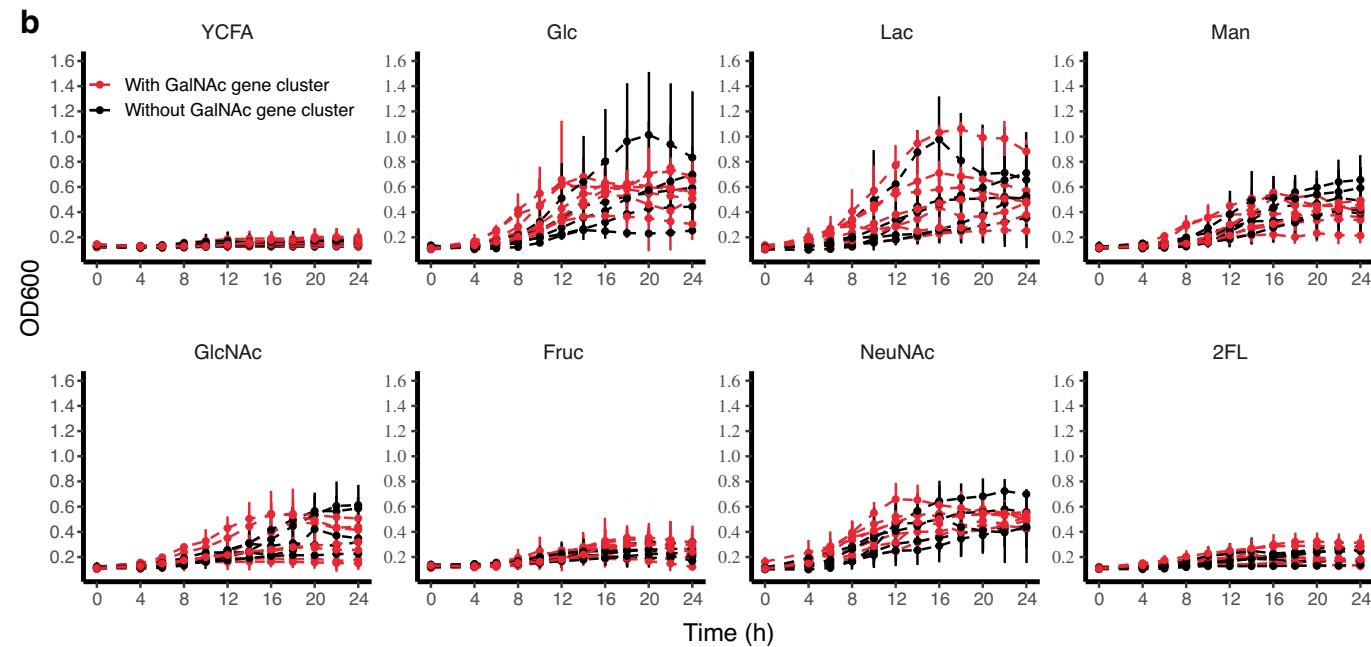

**Extended Data Fig. 7 | Growth curves for *F. prausnitzii* strains in media supplemented with different sugars. a**, Growth curve of *F. prausnitzii* strain HTF−383 in GalNAc medium over a longer growth time (40 h). X-axis refers to culturing time (in hours). Y-axis refers to the cell density measured as $OD_{600}$ value. The red points with bars on the growth curve represent the means ± standard deviation of three replicates. **b**, Red lines indicate the growth curves of strains with SV 577−579. Black lines indicate the growth curve of strains without this SV. YCFA is Yeast Casitone Fatty Acids (YCFA) basic medium without supplemental sugars. The other graphs are for the YCFA medium supplemented with the sugar indicated: glucose (Glc), lactose (Lac), mannose (Man), N-acetylglucosamine (GlcNAc), fructose (Fruc), N-acetylneuraminic acid (NeuNAc) and 2-fucosyllactose (2FL). X-axis is the culturing time (in hours). Y-axis is the cell density measured as $OD_{600}$ value. The red and black points with bars on the growth curves represent the means ± standard deviation of three replicates.

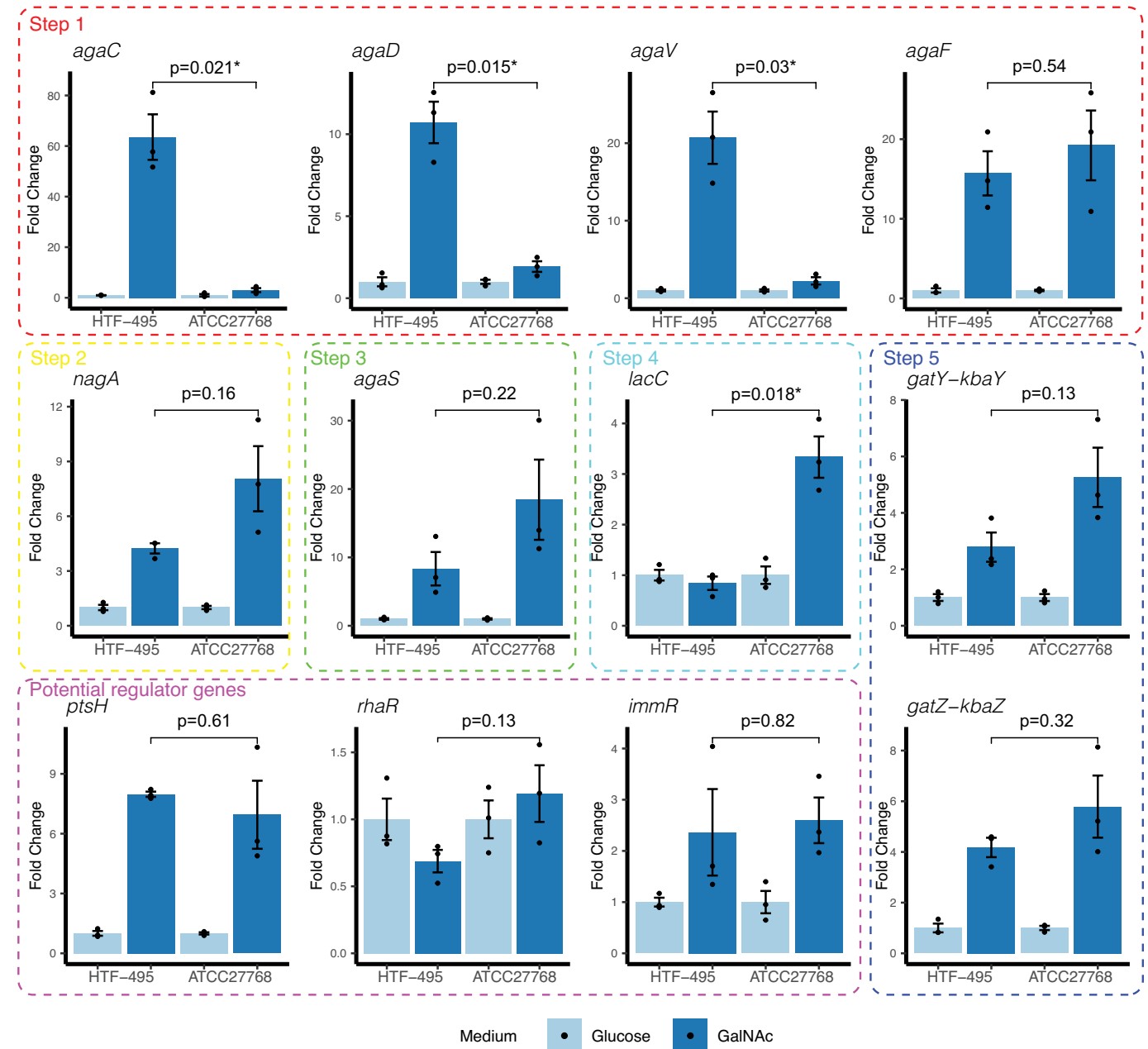

**Extended Data Fig. 8 | Comparison of GalNAc-induced expression fold changes in GalNAc genes between HTF-495 and ATCC27768.** Bar plots represent the average fold changes of gene expression induced by GalNAc compared to glucose induction. Each dot represents a sample, and the bar indicates the standard error of three replicates. The light blue and dark blue bars represent the expression fold change of glucose induction and GalNAc induction in relation to glucose induction, respectively. Thus, the fold change in glucose was set to 1. Unadjusted *P*-values estimated using the two-sided t-test are given for each gene. * Indicates a difference significant at unadjusted *P* < 0.05 level.

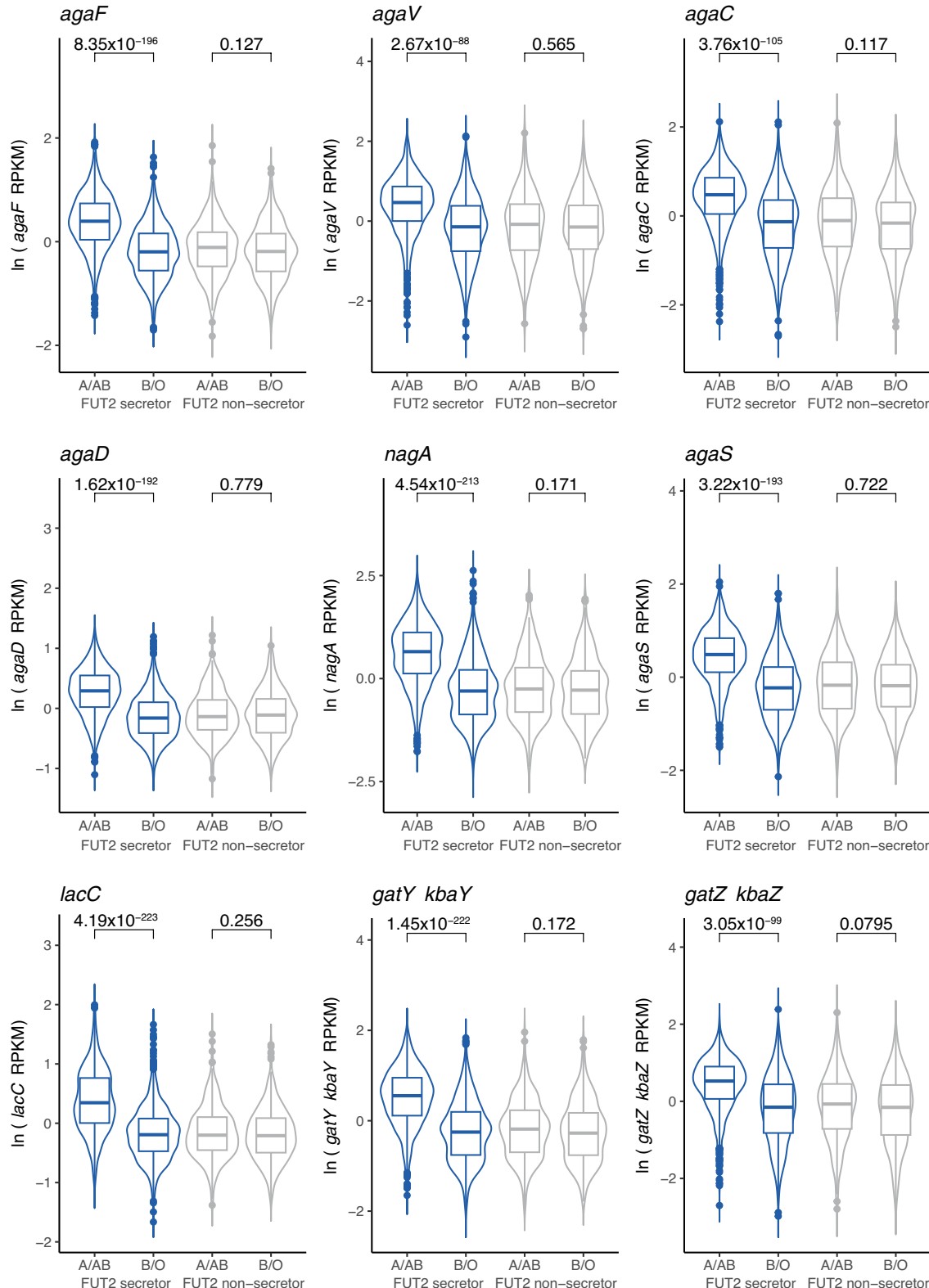

**Extended Data Fig. 9** | See next page for caption.

**Extended Data Fig. 9 | Association of nine key GalNAc pathway genes with A-antigen presence in FUT2 secretors and non-secretors.** Y-axis of the violin plots is the gene abundance in the DMP cohort ($n$ = 1868, 2496, 610 and 760 for FUT2 secretor A/AB, FUT2 secretor B/O, FUT2 non-secretor A/AB and FUT2 non-secretor B/O, respectively). The RPKM value of gene abundance (number of reads per kilobase of transcript per million reads mapped) was profiled using ShortBRED. The y-axis unit *ln(RPKM)* stands for normalized gene abundance after ln(RPKM + 1) transformation, adjusted for covariates (age, sex, total read number, species abundance and the five *F. prausnitzii* SVs associated with *ABO*). The violin plots for FUT2 secretors are colored in blue. Plots for FUT2 non-secretors are in gray. Violin plots show density distribution, while the inner boxplots represent summary statistics: the center line is the median, the box hinges are the lower and upper quartiles of the distribution, whiskers extend no further than 1.5× interquartile range from the hinges, and data beyond the end of the whiskers are outliers plotted as individual points. Unadjusted *P*-values show the significance of the association between the ABO blood group and gene abundance while adjusting for covariates estimated using linear mixed models.

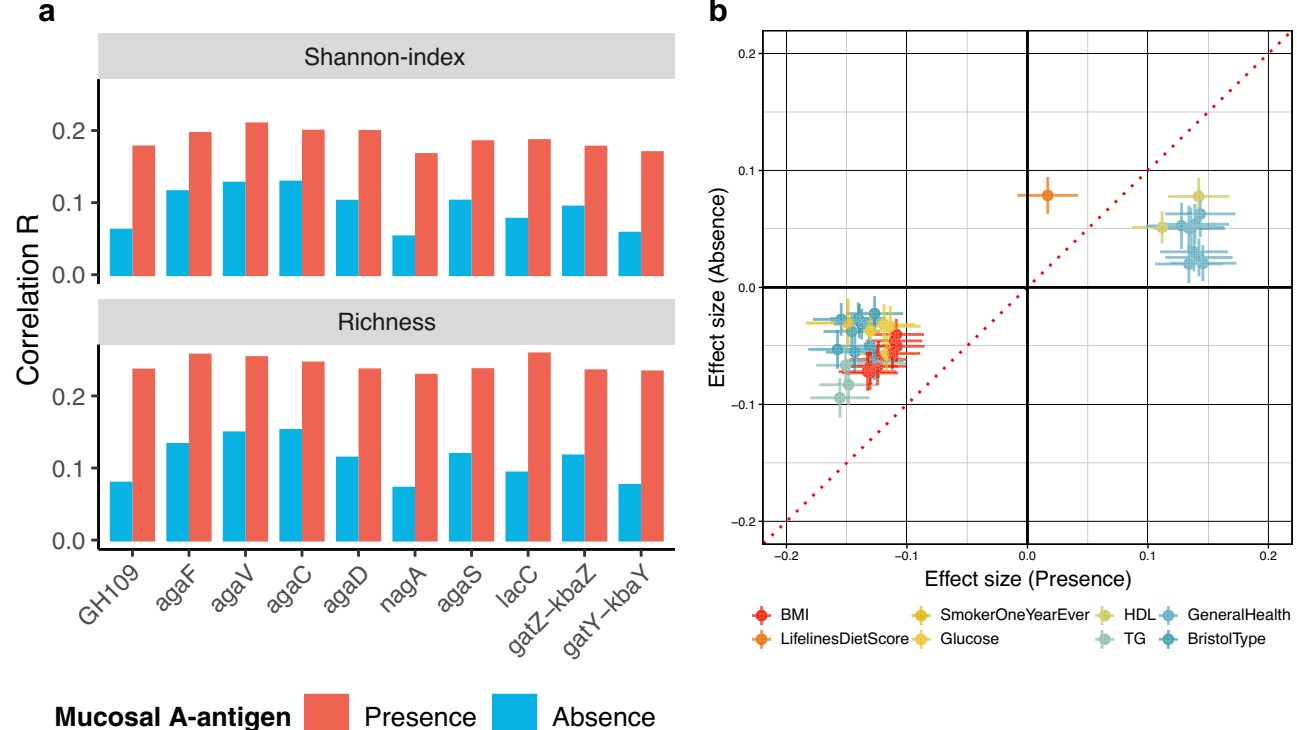

**Extended Data Fig. 10 | Association of GalNAc pathway genes with gut microbiome diversity and host phenotypes. a**, Community-level GalNAc utilization capacity is positively associated with gut microbiome diversity and richness. Bar plot shows the Spearman correlation coefficients (*R*) between the abundance of GalNAc genes in a microbial community with the alpha diversity (Shannon index) (top panel) and richness (bottom panel) of the community for individuals with (red, *n* = 1,868) and without (blue, *n* = 3,866) mucosal A-antigen. Y-axis is the Spearman correlation coefficient. X-axis indicates different GalNAc genes. **b**, Heterogeneity of associations between gut microbial GalNAc utilization genes and human phenotypes in individuals with and without mucosal A-antigen. Scatter plot shows the association effect size (standardized beta-coefficient from linear regression) between GalNAc metabolism gene abundance and host phenotypes in individuals with mucosal A-antigen (x-axis) and those without (y-axis). Error bars indicate the confidence interval of the beta-coefficient estimation. The associations between GalNAc metabolism gene abundance and host phenotypes are significantly higher in individuals with mucosal A-antigen (*n* = 1,868) compared to those without (*n* = 3,866) (Unadjusted $P_{\text{heterogeneity}}$ < 0.05; Cochran's Q test). Dots are colored differently for different phenotypes.

# Reporting Summary

## Statistics

For all statistical analyses, confirm that the following items are present in the figure legend, table legend, main text, or Methods section.

| n/a | Confirmed | |
|---|---|---|
| ☐ | ☒ | The exact sample size (*n*) for each experimental group/condition, given as a discrete number and unit of measurement |
| ☐ | ☒ | A statement on whether measurements were taken from distinct samples or whether the same sample was measured repeatedly |
| ☐ | ☒ | The statistical test(s) used AND whether they are one- or two-sided *Only common tests should be described solely by name; describe more complex techniques in the Methods section.* |
| ☐ | ☒ | A description of all covariates tested |
| ☐ | ☒ | A description of any assumptions or corrections, such as tests of normality and adjustment for multiple comparisons |
| ☐ | ☒ | A full description of the statistical parameters including central tendency (e.g. means) or other basic estimates (e.g. regression coefficient) AND variation (e.g. standard deviation) or associated estimates of uncertainty (e.g. confidence intervals) |
| ☐ | ☒ | For null hypothesis testing, the test statistic (e.g. *F*, *t*, *r*) with confidence intervals, effect sizes, degrees of freedom and *P* value noted *Give P values as exact values whenever suitable.* |
| ☒ | ☐ | For Bayesian analysis, information on the choice of priors and Markov chain Monte Carlo settings |
| ☒ | ☐ | For hierarchical and complex designs, identification of the appropriate level for tests and full reporting of outcomes |
| ☐ | ☒ | Estimates of effect sizes (e.g. Cohen's *d*, Pearson's *r*), indicating how they were calculated |

*Our web collection on statistics for biologists contains articles on many of the points above.*

## Software and code

Policy information about availability of computer code

| Data collection | No specific software was used for data collection. |
|---|---|
| Data analysis | All data analyses were conducted using publicly available tools. For this study the following software was used: R v4.1.0, PLINK (v.alpha 2.1), KneadData v.0.7.4, Bowtie2 v.2.3.4.3,  Trimmomatic v.0.39, Kraken2 v.2.1.2, Bracken v.2.6.2, MetaPhlAn v3,SGV-Finder v.1, GCTA toolbox v.1.94.1, Metal v.2020-05-05, MicrobeAnnotator v2.0.5, Bakta v1.8.1, tblastn v2.5.0+, shortBRED toolkit v.0.9.5, RAxML v8, CSI Phylogeny v1.4, IslandViewer v4. The following R packages were used: lme4qtl v.0.2.2, kinship2 v.1.9.6, vegan v.2.6-2, ape v.5.6.-2, stats v.4.3.0, ggtree v.3.2.1, gggenomes v.0.9.9.9000 The analysis code is available at https://github.com/GRONINGEN-MICROBIOME-CENTRE/SV_GWAS. |

For manuscripts utilizing custom algorithms or software that are central to the research but not yet described in published literature, software must be made available to editors and reviewers. We strongly encourage code deposition in a community repository (e.g. GitHub). See the Nature Portfolio guidelines for submitting code & software for further information.

# Data

Policy information about availability of data

All manuscripts must include a data availability statement. This statement should provide the following information, where applicable:

- Accession codes, unique identifiers, or web links for publicly available datasets
- A description of any restrictions on data availability
- For clinical datasets or third party data, please ensure that the statement adheres to our policy

The profile of SVs of all samples and the full summary statistics of genetic associations with bacterial dSVs and vSVs are available at figshare: https://doi.org/10.25452/figshare.plus.c.6877849. The assembled bacterial genomes from the growth experiment are available at NCBI with accession number PRJNA1024432.

The raw metagenomic sequencing data of all four cohorts are publicly available. Three are deposited at the European Genome–Phenome Archive: Dutch Microbiome Project (accession number EGAS00001005027), Lifelines-DEEP (accession number EGAD00001001991), and 300OB (accession number EGAD00001005083). The 500FG data is available at NCBI SRA under accession number PRJNA319574. The metagenomic data of 300TZFG is available in the NCBI BioProject under accession number PRJNA686265.

To protect participant's privacy and respect the research agreements in the informed consent, genotyping data and participant metadata are not publicly available and cannot be deposited in public repositories. The DMP and LLD data can be accessed by all bona-fide researchers with a scientific proposal by contacting the Lifelines Biobank (instructions at https://www.lifelines.nl/researcher/how-to-apply). Researchers will need to fill in an application form, which will be reviewed within 2 working weeks. If the proposed research complies with Lifelines regulations, e.g., noncommercial use and guarantee of participants' privacy, researchers will then receive a financial offer and a data and material transfer agreement to sign. In general, data will be released within 2 weeks after signing the offer and data and material transfer agreement. The data will be released in a remote system (the Lifelines workspace) running on a high-performance computer cluster to ensure data quality and security. As Lifelines is a non-profit organization dependent on (governmental) subsidies, a fee is required to cover the costs of controlled data access and supporting infrastructure. The fee for data access on the HPC is €3,500 for 1 year and the fee for the Lifelines Workspace environment is €4,500 for 1 year, or less for shorter periods of time. There are no restrictions on the downstream re-use of aggregated, non-identifiable results (as approved by Lifelines), nor are there authorship requirements, but Lifelines does request that it is acknowledged in publications using these data. The data access policy, data access fees and an example Data and Material Transfer Agreement (which includes details on how to acknowledge the use of Lifelines data in publications) are described in detail at https://www.lifelines.nl/researcher/how-to-apply. Note that data access for replication can be arranged via Lifelines. Lifelines will not charge an access fee for controlled access to the full dataset used in the manuscript (including phenotype and sequencing data), for the specific purpose of replication of the results presented in this Article or for further assessment by the reviewers, for a period of three months. Researchers interested in such a replication study or review assessment can contact Lifelines at research@lifelines.nl.

The genotype and metadata of 500FG, 300OB, and 300TZFG cohorts can be requested via the Human Functional Genomics Data Access Committee (Martin Jaeger, e-mail: Martin.Jaeger@radboudumc.nl)]. There are no conditions associated with its use, with the exception of those associated with data that may lead to compromising patient confidentiality, such as raw genomics data. The data are freely available, and no agreement or costs are required. The applicants would receive a response within 4 weeks from application.

Gut microbial SV calling was conducted based on reference microbial genomes from the proGenomes database (http://progenomes1.embl.de/). ShortBRED analysis was performed based on the UniRef90 database (https://ftp.uniprot.org/pub/databases/uniprot/uniref/uniref90/).

# Human research participants

Policy information about studies involving human research participants and Sex and Gender in Research.

| | |
|---|---|
| Reporting on sex and gender | We added sex as covariate in association analyses as the primary goal of the project was to identify associations between host and microbial genetic variants irrespective of sex. |
| Population characteristics | Data from five population-based cohorts were used in this study.<br>DMP cohort is a prospective cohort from the north of the Netherlands that consists of 8,719 individuals. 57.4% of participants are female, the mean age (SD) of participants is 48.42 (14.79) years, mean BMI is 25.56 (4.40).<br>LLD cohort is a prospective cohort from the north of the Netherlands that consists of 1,135 individuals. 58.20% of participants are female, the mean age (SD) of participants is 45.04 (13.60) years and their mean BMI is 25.26 (4.18).<br>The 500FG cohort consists of 534 healthy adult volunteers from the Netherlands. 56.50% of participants are female, the mean age of participants is 27.43 (12.35) years and their mean BMI is 22.70 (2.72).<br>The 300OB is a part of the Functional Genomics project and consists of 302 individuals from the Netherlands with a BMI >27. 44.30% of participants are female, the mean age of participants is 67.07 (5.39) years and their mean BMI is 30.73 (3.48).<br>The 300TZFG cohort consists of 323 individuals from both rural (N = 70, median age 39.6) and urban (N = 253, median age 27.6) areas of Tanzania. A total of 279 individuals with both genotype and metagenomics data are included in the study. |
| Recruitment | DMP and LLD are subsets of the Lifelines cohort, which has been recruited in three stages: recruitment of an index population via general practitioners, subsequent inclusion of their family members, and online self-registration. This cohort is considered representative of adult population of the North of the Netherlands. 500FG and 300OB cohorts are part of the Functional Genomics project. The inclusion of the volunteers took place between 8/2013 until 12/2014 in the Radboud University Medical Center, the Netherlands. 500FG is a population-based cohort, while around half of 300OB participants are clinically diagnosed with metabolic syndrome.<br>The participants of 300TZFG were recruited from the Kilimanjaro region of the Northern Tanzania between March and December 2017, at the Kilimanjaro Christian Medical Center and Lucy Lameck Research Center, in Moshi municipal. Exclusion criteria were pregnancy, a known acute or chronic disease, use of antibiotics or anti-malarials in the previous three months, or receiving treatment for tuberculosis infection in the past year. The information on the study was given through leaflets or announced during the mass gathering. All volunteers were interviewed by a member of the study team using a guided pre-screening questionnaire prior to being invited to the study center. |
| Ethics oversight | The Lifelines study was approved by the ethics committee of the University Medical Center Groningen (METc2007/152). All |

| Ethics oversight | participants signed an informed consent form prior to enrollment. Additional written consents were signed by the DMP participants or legal representatives for children aged under 18 years. The Lifelines-DEEP study was approved by the Institutional Ethics Review Board of the University Medical Center Groningen (ref. M12.113965), the Netherlands. The 300-Obesity study was approved by the IRB CMO Regio Arnhem-Nijmegen (nr. 46846.091.13). The 500FG study was approved by the Ethical Committee of Radboud University Nijmegen (NL42561.091.12, 2012/550). The inclusion of volunteers and experiments were conducted according to the principles expressed in the Declaration of Helsinki. All volunteers gave written informed consent before any material was taken.
The 300TZFG study was approved by the Ethical Committees of the Kilimanjaro Christian Medical University College (CRERC) (no. 936) and the National Institute for Medical Research (NIMR/HQ/R.8a/Vol. IX/2290) in Tanzania. |
|---|---|

Note that full information on the approval of the study protocol must also be provided in the manuscript.

# Field-specific reporting

Please select the one below that is the best fit for your research. If you are not sure, read the appropriate sections before making your selection.

☒ Life sciences ☐ Behavioural & social sciences ☐ Ecological, evolutionary & environmental sciences

For a reference copy of the document with all sections, see nature.com/documents/nr-reporting-summary-flat.pdf

# Life sciences study design

All studies must disclose on these points even when the disclosure is negative.

| Sample size | In order to ensure the analysis power, the study includes as much as subjects as possible from the five cohorts. Thus no sample size calculation was performed. For each cohort we used samples that had both metagenomic and genotype data available: DMP (N = 7,372), LLD (N = 981), 500FG (N = 396), 300OB (N = 266), and 300TZFG (N = 279). |
|---|---|
| Data exclusions | We excluded samples with < 5% of structural variations called. |
| Replication | We used all available samples of four independent Dutch cohorts for meta-analysis. To ensure the replication of the identified associations, we required that the associations were not only significant at Bonferroni-corrected P <0.05 in meta-analysis but also nominally significant (p < 0.05) in at least two cohorts with consistent effect direction. The 300TZFG was used as an extra independent replication cohort in non-European population, with a nominally significant threshold (p < 0.05). |
| Randomization | This is human cohort-based analysis. The sample collection and sequencing were performed in a random order. No extra randomization was done for this study. |
| Blinding | This study is a human cohort based, observational study. Thus no blinding was performed. |

# Reporting for specific materials, systems and methods

We require information from authors about some types of materials, experimental systems and methods used in many studies. Here, indicate whether each material, system or method listed is relevant to your study. If you are not sure if a list item applies to your research, read the appropriate section before selecting a response.

## Materials & experimental systems

| n/a | Involved in the study |
|---|---|
| ☒ | ☐ Antibodies |
| ☒ | ☐ Eukaryotic cell lines |
| ☒ | ☐ Palaeontology and archaeology |
| ☒ | ☐ Animals and other organisms |
| ☒ | ☐ Clinical data |
| ☒ | ☐ Dual use research of concern |

## Methods

| n/a | Involved in the study |
|---|---|
| ☒ | ☐ ChIP-seq |
| ☒ | ☐ Flow cytometry |
| ☒ | ☐ MRI-based neuroimaging |

