## [Peer Review File · Nature]

Manuscript Title: Host genetic regulation of human gut microbial structural variation

Reviewer Comments & Author Rebuttals

Reviewer Reports on the Initial Version:

Referees' comments:

Referee #1 (Remarks to the Author):

This study investigates the important question of whether human genetic variation shapes the strain-level composition of the gut microbiome. Prior studies have tested for associations between genetic variation in humans (and other mammals, eg, mice, baboons) using microbiome datasets resolved only to the species (or higher) taxonomic scale. These prior studies have produced varied results, with only a few loci displaying significant associations with microbiome taxonomic composition across studies (LCT is a notable example). However, because previous methods (e.g., 16S rRNA gene-amplicon sequencing, shotgun metagenomics without assembly) for characterizing the composition of the gut microbiome have not been able to interrogate gut microbial variation at the strain level (ie, the fundamental level of genetic variation in the microbiota), there is undoubtably much still to learn about the interactions between host genetics and gut bacterial symbionts.

Another major limitation of prior studies has been the lack of experimental validation of the inferred interactions between host and bacterial genetic variation. Given the large array of strains within individual bacterial species (and higher taxa), conducting experiments to validate effects of host genetics on microbiome composition at these broad taxonomic scales has been challenging. In contrast, if host-genetic associations with specific bacterial strains (again, the fundamental unit of variation in the microbiota) can be detected, these would afford the opportunity to conduct highly controlled experiments (e.g., by isolating the interactions between individual strain(s) and products/consequences of host genetic variants).

The present study overcomes both major challenges (i.e., lack of strain-level resolution and experimental validation) through a large association study of Dutch individuals and their gut metagenomes followed by in vitro validation of interactions suggested by the associations detected. The main finding is that the ABO genotype of humans differentially affect the abundance of *Faecalibacterium* strains, and that these effects are likely mediated by the influence of ABO genotype on GalNAc levels. A similar association between GalNAc and gut bacteria (*Erysipelotrichaceae*) has been previously reported in pigs (<https://www.nature.com/articles/s41586-022-04769-z>), but the present study is the first (to my knowledge) to demonstrate and experimentally validate this association in humans and the first (to my knowledge) to report host-genetic effects on strains in the microbiota (which represents a major advance).

This study is the most comprehensive investigation of host genetic effects on the gut microbiome

conducted to date (to my knowledge), and it provides an excellent approach that could be employed by future studies in other human (and non-human host) populations. Overall, the manuscript is a major step forward in our understanding (and ability to generate future understanding) of the influence of human genetics on the gut metagenome. This is a timely and important question given the increasing appreciation for the importance of strain level variation in the gut microbiota for human health and the emerging evidence that human populations and gut microbes may have coevolved. I would fully support publication of the manuscript, but have a few critical comments/suggestions:

Heritability analysis: How was the issue of compositionality addressed in the heritability analysis? It has recently been argued (<https://www.biorxiv.org/content/10.1101/2022.04.26.489345v1.abstract>) that failure to address compositionality has been a major weakness of prior heritability studies. In general, the section on heritability seemed underdeveloped relative to the rest of the manuscript. It seems important to more directly compare heritability results/coefficients between strain-level analyses and analyses at higher taxonomic levels.

A strength of this study is the use of structural variants (SVs) to identify strains. The authors were able to catalog, with high confidence, >14,000 SVs in the most abundant 106 gut microbial species. This enabled the authors to conduct association tests with host genetic variation focused on >3,000 SVs that were present in at least 10% of host samples. The prevalence of these strains provided high power for association studies. I think one point that will be important to expand upon in the discussion is that the strain-level variation examined represents only a small fraction of the strain-level variation that exists in the metagenome (e.g., Single nucleotide variants, SNVs, were not tested). There are good reasons for limiting the analysis to SVs (FDR is a major issue), but these could be more thoroughly discussed.

Another strength was that the association study was performed within a single human cohort/population (i.e., the Dutch cohort). This limits the potentially confounding effects of population structure and lifestyle variation. In another sense, this is a limitation of the study (though completely understandable), because it is not clear whether the associations discovered will be applicable to other human populations. Given the results present, I wondered whether it would be possible to directly test for association between *F. rausnitzii* SVs and human genomic variation using available data from other cohorts as well. This may be outside the scope of the study, but validation in other cohorts of the associations detected would greatly expand the potential impact of the study (which will already be exceptionally impactful).

Line 333: “promoting intestinal homeostasis” seems like an overinterpretation of the results. What was the evidence that homeostasis was improved? Increased gut microbiota diversity in itself does not necessarily indicate improved host performance/stability.

Line 353: “Our study demonstrates that associations of host genetics with bacterial SVs 354 can be much stronger than those with bacterial abundances.” This statement seemed poorly supported by the analyses presented. Were direct comparisons made between effects sizes? Some additional context in this discussion would help support this statement.

Line 715: “we shuffled genotype labels 10 times” This seems like potentially too few permutations to give an accurate sense of the FDR. It would be very helpful to present the results (and particularly the variance) from these permutations, as well as additional justification for why more permutations were not performed. It will be important to demonstrate that these 10 permutations were not uncharacteristically favorable to the FDR reported in the paper.

Figure Captions: The captions in all main figures could be expanded slightly so that the figures can be more easily interpreted. What components of the figure represent is not always clearly stated. For example, in Fig. 2 caption there is no mention of the significance threshold; in Fig. 4 the caption does not mention several components of the figures; etc.

Referee #2 (Remarks to the Author):

In this paper, the authors combine fecal microbiome studies from four cohorts collected in the Netherlands (which appear to be a mix of hosts with healthy and unhealthy microbiomes). Around 8000 samples are available with host genotype and microbiome genomes, which is an impressive sample size for a genome-to-genome association study. The authors then associate structural variants called per-species in the microbiome samples (specifically, gene deletions and copy number changes) with common SNP variation in the hosts. There is one convincing association between presence of a gene cluster for GalNAc utilization and hosts with high GalNAc mucus. This association is generally well investigated -- the authors show that an interaction between FUT2 and ABO in humans, and a gene cluster in *F. prausnitzii* are responsible, using more in-depth genomics and growth rate of selected strains on various media. A potentially interesting but less well supported result was that GalNAc-high individuals appeared to have higher microbiome diversity.

I found this to be a broadly well-conducted study, though I suggest a few methodological improvements below, with a convincing finding. (I don't find the other hits convincing, see below, but appropriately the paper only explores the main association). The scale of the study is impressive, but despite this only one convincing association was found – that bacteria which can metabolise a particular sugar are more common in hosts which produce more of this sugar. This is probably partly a feature of the incredibly high multiple testing burden of genome-to-genome studies, but may also be that host genetics does not have a large effect on microbiome diversity at this level. That this is a 'powerful strategy' for improving functional understanding of host genetics on the microbiome doesn't really seem to follow – how many samples would be needed to find more associations, and is this feasible to collect? It is certainly a strength of the association approach that so many potential interactions can be screened at once to lead to such a result, but the overall relevance of this particular result to the field is perhaps limited. (I would also note that similar analyses and results have been found before e.g. <https://www.pnas.org/doi/10.1073/pnas.1305559110>, <https://www.nature.com/articles/ng.3835> and a previous study by some of the same authors <https://pubmed.ncbi.nlm.nih.gov/30918406/>).

Nevertheless, this is a competent genome-to-genome study with a convincing interaction finding. Some more specific comments and suggestions follow:

1) The SV typing needs more detail, I looked at the previous reference but found it insufficient. This is key to the study. Can the authors address specifically: can copy number calls be affected by taxon abundance? Is it possible that deletions are just missed sequence (misassemblies, low DNA, other artefacts). What are the false positive and false negative rates of this calling approach? How are genes or regions for the SVs defined, are these clusters of orthologous genes, or something else? I also don't understand the rationale for looking at only SVs, why weren't SNPs included? As a more minor point, typically this would be called gene variation and copy number variation - structural variation is more usually used when referring to changes in gene order.

2) Rather than calling variants for each species, did the authors consider associating gene clusters via their function irrespective of species, which would likely be more powerful. For example, looking at GalNAc clusters in the entire metagenome of each sample would be an obvious extension. But taking a functional/gene-based association approach from the beginning could increase the number of signals found.

3) The assays of growth presented in figure 4 are broadly convincing and have the advantage of using a variety of strains. However as far as I can tell an effect of strain genetic background has not been ruled out, and an ideal experiment to add would be a knock out of the GalNAc gene cluster vs the wild-type strain, and calculating a competitive index on each of the two media conditions.

4) The result in figure 5 on overall diversity seems too good to be true: double the richness/diversity is observed between GalNAc low/high individuals. Are there other possible host factors that could contribute to this large difference? Any other possible confounders to control for? It would also be interesting to see whether any host associations occur when associated against overall gut metagenome diversity.

Minor points

- Using permutation testing to determine a significance is likely to be inappropriate as the samples are not fully IID. Switching labels within genotype groups (in the metagenome) might be more appropriate, or just a FWER correction. So those results at the borderline of significance are not convincing on their own. QQ plots would also be helpful to assess this.
- The study is referred to as a gut microbiome study, all the samples are fecal. Taxon (and SVs) in fecal samples aren't necessarily representative of the gut as a whole. This should be mentioned somewhere.
- Checking the covariance of the SVs could be useful. If many of them are colinear or closely related, a latent factor analysis of the SVs may be more powerful (for example, PCA, non-negative matrix factorisation or the PEER method. An example of this: <https://www.nature.com/articles/s41586-020-2635-8>).
- Relatedly, population structure of the bacterial metagenomes was not mentioned. Are the SVs sufficiently controlled for this? You could check LD between them.
- Were the results in figure S2 corrected for multiple testing in the same way as the main analysis?
- The github with the code needs more detail in the README to describe what the files included actually do.
- The legend of figure 2 does not contain enough detail to describe the figure. In particular, is this all the associations of the SVs overlaid at each human chromosome position?

- Is *F. prausnitzii* transmitted between people? Mothers and children? In light of this, would there be any selection pressure to either a) lose the gene cluster in question or b) retain the cluster to remain more plastic, in case a GalNAc high host is encountered?
- The panel labels in figure 3 don't all align with the caption.
- Figure 4B -- it is unclear what the relevance of all the different ways that steps can be missing here is. Is it just that the pathway is complete or incomplete?
- The legend of figure 4E-H is poor, and make it hard to follow what the plot shows.
- I am not suggesting this is changed, but in general I am surprised to see that a fixed effects GWAS model with plink was used. A linear mixed model approach would both better control for ancestry and relatedness, and improve power by not removing samples.

Referee #3 (Remarks to the Author):

Review of Zhernakova et al.

This study by Zhernakova et al. reports the results of a large-scale analysis of human genetic polymorphisms that correlate with the presence/absence of gut bacterial structural variants (SVs). The study revealed 18 significant associations and the authors subsequently focused on a connection between the ABO locus, which adds terminal alpha3-linked GalNAc (blood group A) or alpha3-linked Gal (blood group B) to blood group glycans but, importantly, also to secreted mucus glycans. The authors also made a further connection to the function of the *Fut2* locus which is responsible for adding terminal alpha2-linked fucose to mucus glycan chains and this modification is a required precedent for addition of blood groups A/B. Thus, it is intuitive that association between the ABO locus and certain bacterial pathways would only be meaningful if the host has a functional *Fut2* and can actually execute the A/B modification pathway via the ABO locus. A specific connection is established between a locus in certain *Faecalibacterium prausnitzii* (Fp) strains and the presence of blood group A. A second connection is made between blood group A and *Collinsella aerofaciens* (Ca) and the authors provide some support for a functional role by demonstrating that Ca can utilize GalNAc as a nutrient and Fp strains that have a particular locus encoding GalNAc utilization genes (i.e., the identified SV) also mostly can use GalNAc, although one strain that had a locus couldn't and one grew very poorly making this correlation imperfect.

As the authors cite, there have been 3 previous studies connecting variations in the ABO locus and abundance of certain gut bacterial taxa, including one study involving the authors of this paper (all published in *Nature Genetics*). The major advance in this work is to make the connection to the presence of genes in Fp and not just the abundance of this or other organisms. The large human cohort is a strong feature of this study and the sequencing/informatics seem sound, although this is not my area of expertise, which is instead in microbiome function, mucus and glycobiology. In this area, I'm left with the feeling that the connection to GalNAc utilizing bacteria is less complete and this is elaborated on in specific points below. There are many other bacteria from the human gut microbiome that are capable of utilizing GalNAc, so it doesn't seem that a specificity for Fp's ability to utilize this sugar is the only cause of its association with the ABO/*Fut2* loci. On the microbiology side, the imperfect correlation between strains that have the Gal/GalNAc utilization locus would be made stronger if all the strains with the locus grew well on GalNAc or the authors demonstrated that

the strains that have it and still grow poorly truly have inactivated the functions (e.g., due to inversion of the locus). The more exciting path would be to pursue some of the additional human genome polymorphisms that are associated with other bacteria. Although, if this has not been started or revealed mechanistic connections yet could be a long way off.

Specific points:

The introduction or early results might benefit from a more specific description of what is being counted as an SV without going to methods or previous literature. I looked at this in the methods and realize that a pipeline from another group (Zeevi et al) is used but I only had a vague idea of what these “variations” might be until seeing Figure S4, but then was confused as to why the 577-79 segment showed up so strong despite looking very similar among strains in that region on the “right” side of the locus, while the adjacent region to the left is much more striking in how it reflects presence or absence of potentially causal genes and actually looks like a deletion event in some strains (or a gain of function by lateral gene transfer or another mechanism). Duplication/deletion events are discussed heavily in the introduction. Deletions in some lineages could certainly account for loss of function. But duplications of existing genes and subsequent drift to establish new functionality don’t seem like the most common or quickest way to evolve new functions (this is also not my area of expertise). Rather, existing functions might be acquired by lateral gene transfer. This might just need some re-wording to note that the presence/absence of genes are what is being correlated with less description of how they are obtained or loss, unless its more clearly known.

No connection to blood group A cleaving bacteria or enzymes is attempted, and does it necessarily need to be in this study. But it’s worth considering that GalNAc would need to be liberated in this context by a specific microbial enzyme and the producing organism might benefit the most due to attachment to mucus, proximity to released substrate or other “selfish” effects (exceptions also obviously exist in which bacteria release sugars for other). Glycoside hydrolase family 109 (GH109) enzymes have been implicated in cleaving blood group A and were discovered in part in human gut bacteria (*Bacteroides* species, plus others). It would be interesting to know if the identified *F. prausnitzii* and *Collinsella* strains have these enzymes or the activity when assayed biochemically (do they also vary concordantly with the identified GalNAc utilization locus?). If GH109 enzymes are not present, it would suggest that they rely on a different bacterium/enzyme to release GalNAc (or encode members of another family that does this catalysis, which could be tested quite simply using Fp bacterial lysates and commercially available substrates that harbor blood group A, looking for cleavage with a simple assay like thin layer chromatography). There is of course another relevant source of GalNAc in mucin glycans: the core alpha-linked GalNAc that forms the base of each O-linked glycan. Since this linkage needs to be present to extend the mucin glycan chains it must be stoichiometrically more abundant than blood group A but is more difficult to access. Perhaps conditional access to this core GalNAc in pigs vs. humans is part of the basis for discordance between the *fut2* and ABO loci, i.e., in *fut2* hosts there will still be core GalNAc present and perhaps different species are capable of accessing it in pig vs. human. Or, perhaps *Fut2* loss of function polymorphisms don’t naturally exist in pigs?

Fig. 3A: should the label of the gray histogram bars be “Fut2 non-secretors”?

Line 252: What is the meaning of “essential” here? Are these genes essential to the survival of the bacterium and therefore cannot be lost or disrupted (the genetic sense of essential) or are they essential for the pathway in which GalNAC is metabolized. Either way, how is this known from sequence information?

Line 272: The inversion of the locus under consideration in the ATCC strain that grows poorly on GalNAC/Gal is interesting. But, how was it determined that this inversion caused the pathway to be inactive as stated? Some other loci seem to have inversions of the teal colored genes (lacC, KbaZ, NagA) and these seem to tolerate the arrangement change. There is another strain in Fig. 3D/GalNAC that grows noticeably poorly on GalNAC. Can this be explained?

Line 309: The meaning of “biosynthesis cluster” is unclear in this context since the genes that are enumerated are part of a proposed GalNAC degradation pathway. Are they thought to also be involved in synthesizing GalNAC or another molecule?

Lines 313-333: It’s not clear what extra impact classifying individuals as high or low GalNAC secretors is beyond possibly getting some extra statistical power by combining the 3 GalNAC negative groups into one comparison group with a larger subject number? What are the units for the abundance violin plots shown in Fig. S6?

Line 321-323: It’s not intuitive to me how a bacterial intracellular pathway for metabolizing a hexose sugar to central metabolites would “create niches” for other bacteria or associate with diversity. Both of the P values noted here for increased diversity and association with particular genes in people with A high mucus and A- mucus seem pretty impressive. There are a lot of other gut bacteria that also utilize GalNAC when it is presented as a free sugar (for one example, a recent survey of 354 human gut Bacteroidetes strains from 30 species (PMID: 35166563) showed that 347/354 were able to use this sugar as a sole carbon source. Some of these same species and strains are also one that are capable of cleaving it from blood group A glycans. Even while presented last, this section is quite speculative and would need more experimental work.

Line 200: Not clear how the potassium channels would be proposed as “medications”. Mean “targets for potential IBD medications.”?

Line 189: Not clear what the basis for calling something relevant (presumably vs. less relevant) functionality is in this context.

Author Rebuttals to Initial Comments:

Response to Reviewers

Host Genetic Regulation of Human Gut Microbial Structural Variation

Nature 2022-11-17759A

Reviewer #1:

This study investigates the important question of whether human genetic variation shapes the strain-level composition of the gut microbiome. Prior studies have tested for associations between genetic variation in humans (and other mammals, eg, mice, baboons) using microbiome datasets resolved only to the species (or higher) taxonomic scale. These prior studies have produced varied results, with only a few loci displaying significant associations with microbiome taxonomic composition across studies (LCT is a notable example). However, because previous methods (e.g., 16S rRNA gene-amplicon sequencing, shotgun metagenomics without assembly) for characterizing the composition of the gut microbiome have not been able to interrogate gut microbial variation at the strain level (ie, the fundamental level of genetic variation in the microbiota), there is undoubtedly much still to learn about the interactions between host genetics and gut bacterial symbionts.

Another major limitation of prior studies has been the lack of experimental validation of the inferred interactions between host and bacterial genetic variation. Given the large array of strains within individual bacterial species (and higher taxa), conducting experiments to validate effects of host genetics on microbiome composition at these broad taxonomic scales has been challenging. In contrast, if host genetic associations with specific bacterial strains (again, the fundamental unit of variation in the microbiota) can be detected, these would afford the opportunity to conduct highly controlled experiments (e.g., by isolating the interactions between individual strain(s) and products/consequences of host genetic variants).

The present study overcomes both major challenges (i.e., lack of strain-level resolution and experimental validation) through a large association study of Dutch individuals and their gut metagenomes followed by *in vitro* validation of interactions suggested by the associations detected. The main finding is that the ABO genotype of humans differentially affect the abundance of *Faecalibacterium* strains, and that these effects are likely mediated by the influence of ABO genotype on GalNAc levels. A similar association between GalNAc and gut bacteria (*Erysipelotrichaceae*) has been previously reported in pigs (<https://www.nature.com/articles/s41586-022-04769-z>), but the present study is the first (to my knowledge) to demonstrate and experimentally validate this association in humans and the first (to my knowledge) to report host genetic effects on strains in the microbiota (which represents a major advance).

This study is the most comprehensive investigation of host genetic effects on the gut microbiome conducted to date (to my knowledge), and it provides an excellent approach that could be employed by future studies in other human (and non-human host) populations. Overall, the manuscript is a major step forward in our understanding (and ability to generate future understanding) of the influence of human genetics on the gut metagenome. This is a timely and important question given the increasing appreciation for the importance of strain-

level variation in the gut microbiota for human health and the emerging evidence that human populations and gut microbes may have coevolved. I would fully support publication of the manuscript, but have a few critical comments/suggestions:

Question 1.1. Heritability analysis: How was the issue of compositionality addressed in the heritability analysis? It has recently been argued (<https://www.biorxiv.org/content/10.1101/2022.04.26.489345v1.abstract>) that failure to address compositionality has been a major weakness of prior heritability studies. In general, the section on heritability seemed underdeveloped relative to the rest of the manuscript. It seems important to more directly compare heritability results/coefficients between strain-level analyses and analyses at higher taxonomic levels.

Response 1.1: We thank the reviewer for raising this concern. We would first like to clarify that we calculated the heritability of structural variations (SVs) after correcting for species abundance. We did not estimate heritability at the species abundance level. 'Compositionality' refers to data where the values represent the relative proportions or percentages, thus the relative abundance of a species in a microbial community is clearly compositional data. However, our SV data refers to the absence or differential abundance of a specific genomic region in the metagenome. Thus, we do not think compositionality is an issue in our heritability estimation of bacterial SV, as we elaborate in more detail below.

It has been well acknowledged that there are statistical challenges related to microbial compositional data, i.e., microbial relative abundances, as also discussed by Bruijning *et al.*¹. Several data normalization and transformation approaches have been proposed to deal with this issue, such as the widely used centered-log ratio transformation to remove the dependency between the abundances of different taxa. In our original version of the manuscript we did not calculate the heritability of species abundances. In the revised version of the manuscript, we now report the heritability estimates previously calculated for the same dataset by Gacesa *et al.*², which accounted for compositionality using centered-log ratio transformation.

However, the effect of compositionality on SVs remains an open question. Firstly, the deletion SV (dSV) data should not exhibit any compositional effect as the binary dSV values (1 and 0) represent the presence/absence status of genomic segments. There is thus no constraint on the sum of dSV values. Secondly, continuous vSV values represent standardized coverage of genomic segments relative to the mean coverage of the segment containing the SV using the Z-score approach: *Standardized coverage = (raw coverage value - mean coverage per kbp segment) / standard deviation of coverage across the genome.*

Theoretically, there could be a constraint on the total number of reads mapped to a specific genome: an increase in the number of mapped reads in a specific genomic region may cause

¹ Marjolein Bruijning and others, 'Relative Abundance Data Can Misrepresent Heritability of the Microbiome' (bioRxiv, 2022), p. 2022.04.26.489345 <<https://doi.org/10.1101/2022.04.26.489345>>.

² Gacesa, R. *et al.* Environmental factors shaping the gut microbiome in a Dutch population. *Nature* 604, 732–739 (2022).

a decrease in the number of mapped reads in other genomic regions. Thus, the microbial vSV data can theoretically be affected by data compositionality. However, the impact of compositionality on vSV is negligible for three reasons. Firstly, the vSV refers to the deviation of read coverage distribution along the genome. On average, vSV regions only account for 0.3–8.4% of the whole bacterial genome³. Thus, the total sum of reads covering vSV regions is not strongly dependent on the total read counts mapped to the genome. We therefore believe there is no constraint in the total number of vSVs reads. Secondly, dependency of SVs has been accounted for during SV calling, and the correlated genomic regions have been grouped together. For example, *Parabacteroides merdae* ATCC 43184:3581_3582;3584_3586 refers to two linked variable genomic regions 3581–3582kb and 3584–3586kb in the genome of *Parabacteroides merdae* ATCC 43184. Thus, the co-abundance of reported vSVs is rather weak. Thirdly, we corrected for species abundance when estimating the heritability of SVs.

We agree with the reviewer's suggestion to expand the heritability section and have now included a comparison of SV heritability to species abundance heritability for all the SVs that were nominally significantly heritable at $P < 0.05$, which included SVs of 16 species. Out of these, only five species were significantly heritable at the species abundance level. Interestingly, we consistently observed that the heritability of SVs was higher, relatively, than that at the species abundance level. We also noticed that not all species with heritable abundance contained heritable SVs. This comparison is now presented in the Results and in supplementary **Figure S2**. The corresponding text now reads:

Line 179-195:

*"We corrected for species abundance in the heritability estimation of SVs, thus our results are independent of the host genetic effect on species abundance. Next, we compared SV heritability with the heritability of the abundances of the corresponding species previously estimated for the same cohort². For the heritable SVs, the heritability of the abundance of the corresponding species seems to either be lower or not significant (Figure S2), suggesting there is an additional effect of host genetics on microbial SV level. For instance, for the most heritable dSV of *F. prausnitzii* ($h^2 = 0.39$, $P_{DMP} = 8.27 \times 10^{-6}$), the previously reported heritability of *F. prausnitzii* abundance was only 0.13 ($P_{DMP} = 0.02$)². For some species that are heritable on the abundance level, we also detected nominally significant SV heritability, even after correcting for species abundance. For example, for *Akkermansia muciniphila*, with previously reported abundance heritability $h^2 = 0.3$, we found several vSVs showing nominally significant heritability ranging between 0.25 and 0.56 (Figure S2; Table S5). However, this study still lacks sufficient power for heritability calculation and comparison. Accurate heritability estimations of species abundance and microbial genetic variation would require a much larger sample size and careful experimental design (e.g., twin studies)."*

³ David Zeevi and others, 'Structural Variation in the Gut Microbiome Associates with Host Health', *Nature*, 568.7750 (2019), 43–48 <<https://doi.org/10.1038/s41586-019-1065-y>>.

Figure S2. Heritability of gut microbial SVs and corresponding species abundance

Bar height indicates heritability h^2 values for vSVs (blue) and dSVs (cyan) that are nominally significant ($P < 0.05$). Error bars in gray are the 95% confidence intervals ($1.96 \times$ standard error (SE)) of the estimated heritability. Solid red vertical lines indicate the heritability of the corresponding species abundance previously reported in Gacesa et al. Nature (2020), and the red opaque rectangles indicate the corresponding 95% confidence interval. Species names marked with * indicate those with significant abundance heritability at $P < 0.05$ level.

Question 1.2. A strength of this study is the use of structural variants (SVs) to identify strains. The authors were able to catalog, with high confidence, >14,000 SVs in the most abundant 106 gut microbial species. This enabled the authors to conduct association tests with host genetic variation focused on >3,000 SVs that were present in at least 10% of host samples. The prevalence of these strains provided high power for association studies. I think one point that will be important to expand upon in the discussion is that the strain-level variation examined represents only a small fraction of the strain-level variation that exists in the metagenome (e.g., Single nucleotide variants, SNVs, were not tested). There are good reasons for limiting the analysis to SVs (FDR is a major issue), but these could be more thoroughly discussed.

Response 1.2: We thank the reviewer for the suggestion. Indeed, we studied only one type of genetic variation: SVs. Other types of genetic variations that exist in bacterial genomes, especially SNVs, can also provide valuable insight into host–microbiome interaction, and newly developed k-mer-based microbial SNV calling approaches such as GT-Pro⁴ allow for relatively fast metagenotyping of over 100 million SNVs in the gut microbiome. However, the current sample size and multiple testing burden limits analysis power, as such association analysis will need to assess associations across 8 million SNPs in the human genome and over 100 million SNVs in the gut microbiome. Moreover, functional annotation of these SNVs is extremely challenging. We therefore limited our analysis to SVs in the current study. We now acknowledge this in the Discussion:

Line 514-519:

“Second, our study did not include other types of genetic variation, such as single nucleotide variants (SNVs), which have great potential impact on bacterial functionality and host–microbe interaction. However, analyzing genetic associations across the millions of SNVs in the human genome and the hundreds of millions of SNVs in the metagenome would require a much larger sample size. Moreover, functional annotation of SNVs is still challenging.”

Question 1.3. Another strength was that the association study was performed within a single human cohort/population (i.e., the Dutch cohort). This limits the potentially confounding effects of population structure and lifestyle variation. In another sense, this is a limitation of the study (though completely understandable), because it is not clear whether the associations discovered will be applicable to other human populations. Given the results present, I wondered whether it would be possible to directly test for association between *F. prausnitzii* SVs and human genomic variation using available data from other cohorts as well. This may be outside the scope of the study, but validation in other cohorts of the associations detected would greatly expand the potential impact of the study (which will already be exceptionally impactful).

Response 1.3: We thank the reviewer for this suggestion. To replicate the associations in a cohort with different genetic background, diet and environmental exposures, we have now included data from a Tanzanian cohort of 279 individuals. A *F. prausnitzii* SV profile could be generated in 201 of these individuals, and we replicated some of the associations between

⁴ Zhou Jason Shi and others, ‘Fast and Accurate Metagenotyping of the Human Gut Microbiome with GT-Pro’, *Nature Biotechnology*, 40.4 (2022), 507–16 <<https://doi.org/10.1038/s41587-021-01102-3>>.

ABO genotype and *F. prausnitzii* SVs. Interestingly, the frequencies of these SVs are different between the Dutch and Tanzanian cohorts. For example, the top ABO-associated 577–579 dSV in the Dutch cohorts was deleted in 76% of Dutch participants but only 6% of Tanzanian participants. In line with this, we also detected associations specific to the Dutch samples and associations specific to the Tanzanian samples.

We have added this result to the text, and it is also presented in **Figure S4** and **Table S8**:

Line 209-232:

*“We further examined if the associations we identified in the Dutch population could be replicated in non-Europeans with a different genetic background, lifestyle, and environmental exposure profile. To do so, we included an African cohort of 279 individuals from Tanzania, 300TZFG²⁵. SVs of *F. prausnitzii* were detectable in 201 300TZFG individuals at both similar and different frequencies to those seen in the Dutch cohorts (**Figure S4A**), and we detected 12 associations with the ABO locus at a nominally significant level ($P < 0.05$) (**Table S8**). The top-associated SV of *F. prausnitzii* in 300TZFG was dSV 575–577, and its association with rs550057 was significant in both the Dutch meta-analysis ($P = 1.19 \times 10^{-23}$) and 300TZFG ($P = 6.5 \times 10^{-3}$) (**Figure S4B**). For the top-associated *F. prausnitzii* 577–579 dSV in the Dutch cohort, the presence/absence ratio was very different between the Dutch and Tanzanian cohorts: the region was deleted in 76% of DMP participants but only 6% of Tanzanian participants (**Figure S4A**). In line with this, its association with the Dutch top SNP rs635634 was not significant in Tanzanian samples ($P = 0.11$). However, it was nominally associated with another SNP, rs475419 ($P = 0.04$) (**Figure S4C**), and these two SNPs are in weak linkage disequilibrium (LD $r^2_{AFR} = 0.13$, $r^2_{EUR} = 0.19$) in European and African populations but differentially represent blood types in the two populations. For example, rs635634 is in moderate LD with the deletion encoding for O blood type (rs8176719) in northern Europeans ($r^2_{CEU} = 0.41$), but this is not the case in a Kenyan population ($r^2 = 0.11$), where rs475419 instead shows higher LD ($r^2 = 0.43$).*

*In addition to the ABO association, our study also yielded 210 independent suggestive associations (clumping LD $r^2 < 0.1$) at genome-wide significance $P < 5 \times 10^{-8}$ level: 57 associations with dSVs involving 17 species and 153 associations with vSVs involving 30 species (**Table S6** and **S7**).”*

Lines 524-530:

“Fourth, our primary analyses involved only Dutch cohorts, which are very geographically and genetically homogeneous, although we were able to include a Tanzanian replication cohort with a different genetic background, diet, and environmental exposure profile. Future work is needed to assess host genetic and microbial genetic associations in more diverse populations in order to build a better understanding of host–microbiome co-adaptation and co-divergence, as well as to aid in fine-mapping of causal genes.”

Figure S4. Comparison of *F. prausnitzii* dSV associations between the Dutch and Tanzanian cohorts

A. Comparison of the presence/absence ratio of dSVs between the DMP cohort and the Tanzanian cohort. Bar plots show the fraction of samples with (dark blue) or without (cyan) the dSV region. **B.** Association of *F. prausnitzii* dSV 575–577 with rs550057 in both the Dutch and Tanzanian cohorts. Bar plots show the number of samples with (dark blue) and without (cyan) the dSV region per genotype. The p-value indicates the association significance assessed using fastGWA. **C.** Cohort-specific association of *F. prausnitzii* 577–579. The top associated SNP was rs635634 in the DMP cohort (left) and rs475419 in the Tanzanian cohort (right). The linkage disequilibriums (r^2) of the two SNPs are 0.13 and 0.19 in African and European population, respectively, based on the estimate of the 1000 Genomes project. Bar plots show the number of samples with (dark blue) and without (cyan) the dSV region per genotype. The p-value indicates the association significance assessed using fastGWA.

Question 1.4. Line 333: “promoting intestinal homeostasis” seems like an overinterpretation of the results. What was the evidence that homeostasis was improved? Increased gut microbiota diversity in itself does not necessarily indicate improved host performance/stability

Response 1.4: Thank you for pointing this out to us. We have now removed these sentences. In addition, to investigate this issue further, we have now conducted associations between the GalNAc pathway genes and 240 human host environmental exposures and phenotypic factors. Here, we first classified the DMP samples into two groups according to the presence of A-antigen in intestinal mucus as inferred by ABO blood type and *FUT2* status and then conducted an association analysis between the gut microbial GalNAc gene abundances and the 240 human phenotypes using linear regression corrected for age and sex in these two groups separately. At Bonferroni-corrected $P < 0.05$ level, we detected 50 significant associations in the A-antigen presence group and 17 associations in the A-antigen absence group. Strikingly, gut microbial GalNAc gene abundances were significantly associated with blood glucose, Bristol stool type and general health only in the individuals with mucosal A-antigen (**Figure 5D and E**; **Table S18**). In addition, although the abundances of GalNAc genes were significantly associated with BMI, blood triglycerides and high-density lipoprotein in both groups, the effect sizes in the individuals with mucosal A-antigen were significantly higher than in those without mucosal A-antigen. These findings are consistent with the stronger association of GalNAc genes with microbial diversity and richness in individuals with mucosal A-antigens. Altogether, our findings suggest that GalNAc utilization genes have a beneficial impact on human health in individuals with mucosal A-antigens.

We have now updated the Results, and the data is presented in **Figure 5D-E** and **Table S18**.

Line 401-411:

*“Similarly, we associated the abundances of gut microbial GalNAc utilization genes with 240 environmental exposure and health-related parameters in individuals with and without mucosal A-antigen. At Bonferroni-corrected $P < 0.05$ level, we detected 50 significant associations in the A-antigen presence group and 17 associations in the A-antigen absence group. Interestingly, gut microbial GalNAc gene abundances were significantly associated with blood glucose, Bristol stool type, and general health only in individuals with mucosal A-antigen (linear regression, Bonferroni-corrected $P < 0.05$, $P_{\text{heterogeneity}} < 0.05$; **Figure 5D–E and S14**; **Table S18**). Although we observed 11 significant associations between GalNAc genes and blood triglycerides and high-density lipoprotein in both groups, the effect sizes in the individuals with mucosal A-antigen are higher than in those without ($P_{\text{heterogeneity}} < 0.05$; **Figure S14**).”*

Figure 5D–E, The association between GalNAc metabolism gene abundance and host phenotypes in individuals with mucosal A-antigens (D, $N = 1,868$) and without mucosal A-antigens (E, $N = 3,866$). # indicates group-specific significant associations. * indicates significant associations shared by the two groups (Bonferroni-corrected $P < 0.05$). Positive associations are in red. Negative associations are in blue. The color gradients reflect the effect size.

Question 1.5. Line 353: “Our study demonstrates that associations of host genetics with bacterial SVs can be much stronger than those with bacterial abundances.” This statement seemed poorly supported by the analyses presented. Were direct comparisons made between effects sizes? Some additional context in this discussion would help support this statement.

Response: We thank the reviewer for this comment. To follow up this suggestion, we have now compared the heritability of all SVs at $P < 0.05$ levels with the previously estimated heritability of their corresponding species abundance (Gacesa *et al.* 2022, *Nature* 604:732-739). This data is now presented in Figure S2. Although we observe a trend that abundance heritability is weaker or not significant for the species that contained heritable SVs, the current study does not have sufficient power to make a concrete conclusion. We have now removed the statement quoted above from the Discussion. We also now address this in the Results, where we compare the heritability at SV and species abundance level.

Line 181-195:

“Next, we compared SV heritability with the heritability of the abundances of the corresponding species previously estimated for the same cohort Gacesa and others.. For the heritable SVs, the heritability of the abundance of the corresponding species seems to either be lower or not significant (Figure S2), suggesting there is an additional effect of host genetics on microbial SV level. For instance, for the most heritable dSV of *F. prausnitzii* ($h^2 = 0.39$, $P_{DMP} = 8.27 \times 10^{-6}$), the previously reported heritability of *F. prausnitzii* abundance was only 0.13 ($P_{DMP} = 0.02$). For some species that are heritable on the abundance level, we also detected nominally significant SV heritability, even after correcting for species abundance. For example, for *Akkermansia muciniphila*, with previously reported abundance heritability $h^2 = 0.3$, we found several vSVs showing nominally significant heritability ranging between 0.25 and 0.56 (Figure S2; Table S5). However, this study still lacks sufficient power for heritability calculation and comparison. Accurate heritability estimations of species abundance and microbial genetic variation would require a much larger sample size and careful experimental design (e.g., twin studies).

Figure S2. Heritability of gut microbial SVs and corresponding species abundance

Bar height indicates heritability h^2 values for vSVs (blue) and dSVs (cyan) that are nominally significant ($P < 0.05$). Error bars in gray are the 95% confidence intervals ($1.96 \times$ standard error (SE)) of the estimated heritability. Solid red vertical lines indicate the heritability of the corresponding species abundance previously reported in Gacesa et al. Nature (2020), and the red opaque rectangles indicate the corresponding 95% confidence interval. Species names marked with * indicate those with significant abundance heritability at $P < 0.05$ level.

Question 1.6. Line 715: “we shuffled genotype labels 10 times” This seems like potentially too few permutations to give an accurate sense of the FDR. It would be very helpful to present the results (and particularly the variance) from these permutations, as well as additional justification for why more permutations were not performed. It will be important to demonstrate that these 10 permutations were not uncharacteristically favorable to the FDR reported in the paper.

Response 1.6: We thank the reviewer for this comment. We have now decided to apply the stringent Bonferroni correction (5.0×10^{-8} / number of SVs) to control the FDR. This resulted in a p-value cutoff of 2.65×10^{-11} for vSV associations and of 3.00×10^{-11} for dSVs. Using these more stringent cut-offs, only the associations of *ABO* with *F. prausnitzii* SVs remain significant. We now also briefly mention suggestive associations that passed the genome-wide significant level ($P=5.0 \times 10^{-8}$) but did not reach the Bonferroni-corrected threshold. We have updated all the results accordingly:

Line 198-201:

*“Next, we associated the 3,552 SVs with more than 6 million human SNPs, per cohort, followed by a meta-analysis. The genetic associations significant at Bonferroni-corrected $P < 0.05$ level were all associations between the *ABO* locus and SVs of *F. prausnitzii*, including four dSVs and one vSV (Figure 2; Figure S3; Table S6 and S7).”*

Line 229-232:

*“In addition to the *ABO* association, our study also yielded 210 independent suggestive associations (clumping $LD r^2 < 0.1$) at genome-wide significance $P < 5 \times 10^{-8}$ level: 57 associations with dSVs involving 17 species and 153 associations with vSVs involving 30 species (Table S6 and S7).”*

Question 1.7. Figure Captions: The captions in all main figures could be expanded slightly so that the figures can be more easily interpreted. What components of the figure represent is not always clearly stated. For example, in Fig. 2 caption there is no mention of the significance threshold; in Fig. 4 the caption does not mention several components of the figures; etc.

Response 1.7: We thank the reviewer for pointing this out. We have now expanded all the figure legends with more detailed descriptions and explanations.

Reviewer #2:

In this paper, the authors combine fecal microbiome studies from four cohorts collected in the Netherlands (which appear to be a mix of hosts with healthy and unhealthy microbiomes). Around 8000 samples are available with host genotype and microbiome genomes, which is an impressive sample size for a genome-to-genome association study. The authors then associate structural variants called per-species in the microbiome samples (specifically, gene deletions and copy number changes) with common SNP variation in the hosts. There is one convincing association between presence of a gene cluster for GalNAc utilization and hosts with high GalNAc mucus. This association is generally well investigated -- the authors show that an interaction between FUT2 and ABO in humans, and a gene cluster in *F. prausnitzii* are responsible, using more in-depth genomics and growth rate of selected strains on various media. A potentially interesting but less well supported result was that GalNAc-high individuals appeared to have higher microbiome diversity.

Question 2.0. I found this to be a broadly well-conducted study, though I suggest a few methodological improvements below, with a convincing finding. (I don't find the other hits convincing, see below, but appropriately the paper only explores the main association). The scale of the study is impressive, but despite this only one convincing association was found – that bacteria which can metabolise a particular sugar are more common in hosts which produce more of this sugar. This is probably partly a feature of the incredibly high multiple testing burden of genome-to-genome studies, but may also be that host genetics does not have a large effect on microbiome diversity at this level. That this is a 'powerful strategy' for improving functional understanding of host genetics on the microbiome doesn't really seem to follow – how many samples would be needed to find more associations, and is this feasible to collect? It is certainly a strength of the association approach that so many potential interactions can be screened at once to lead to such a result, but the overall relevance of this particular result to the field is perhaps limited. (I would also note that similar analyses and results have been found before e.g. <https://www.pnas.org/doi/10.1073/pnas.1305559110>, <https://www.nature.com/articles/ng.3835> and a previous study by some of the same authors <https://pubmed.ncbi.nlm.nih.gov/30918406/>).

Nevertheless, this is a competent genome-to-genome study with a convincing interaction finding. Some more specific comments and suggestions follow:

Response 2.0: We thank the reviewer for these comments. Despite the large sample size, our study indeed still lacks analysis power. After applying Bonferroni correction, only the associations between the *ABO* genetic variants and SVs of *F. prausnitzii* remained significant, with an additional 210 independent suggestive associations observed at a genome-wide significance level of $P < 5 \times 10^{-8}$. It is also very challenging to estimate how many samples would be needed, and, in general, the genetic impact on the gut microbiome is very small.

Nonetheless, our study makes some critical contributions to the field. Numerous studies have revealed host genetic associations with the relative abundance of species in the human gut microbiome, but functional understanding of those associations is limited. As Reviewer 1 complimented, this is the first host genetic association study on bacterial genetic variation in humans, and it is also the first to experimentally validate the human–microbiome genetic associations identified. Our current study has several general – but important – messages for

the field: i) host–microbe interactions can be seen in genetic associations between the human genome and the gut microbial genomes, ii) the host genetic effect on bacterial genomic variation cannot always be seen at species abundance level, and iii) host genetic association with bacterial structural variation can help to pinpoint bacterial functional and causal genes.

ABO is among the few human loci to be convincingly associated with bacterial relative abundance. Our study helps to zoom in from species abundance to underlying causal pathways. Although a similar observation was previously reported in a mosaic pig population, we reveal differences in genetic associations between humans and pigs. To start, GalNAc-containing species are different in humans and pigs. The GalNAc association is also dependent on *FUT2* genotype in humans, but not in pigs. Moreover, we carried out *in vitro* bacterial culturomics experiments to prove GalNAc utilization by the *F. prausnitzii* strains we identified and conducted additional association analyses to study the relationship between GalNAc-related genes and host health parameters. Our results provide evidence supporting the importance of bacterial GalNAc utilization in human health.

The references mentioned are relevant, but none have conducted comprehensive investigation into the host’s genetic association on gut microbial genetic variation as the present study. For example, Sheppard *et al.*⁵ and Zeevi *et al.*⁶ investigated the association between microbial genetics and host characteristics and adaptation to hosts, while Ansari *et al.*⁷ only explored the links between the human genome and the genetics of the hepatitis C virus. We have now acknowledged these papers in the Introduction.

Based on the above discussion, we have removed the words “powerful” throughout the manuscript but still believe that our study demonstrates the power of genetic variations in moving from species abundance to functionality.

The text now reads:

Line 127-133:

“However, little is known about how human genetics shape the individual’s intestinal environment and exert selective pressure on the genetic landscape of the gut microbiome, with the limited studies performed thus far usually focusing on bacterial or viral isolates^{18,19}. Genetic association between human genetic variants and microbial SVs may thus help us understand the mechanisms that underlie the symbiotic relationship between gut microbes and their human host.”

⁵ Samuel K. Sheppard and others, ‘Genome-Wide Association Study Identifies Vitamin B5 Biosynthesis as a Host Specificity Factor in *Campylobacter*’, *Proceedings of the National Academy of Sciences*, 110.29 (2013), 11923–27 <<https://doi.org/10.1073/pnas.1305559110>>.

⁶ Zeevi and others, ‘Structural Variation in the Gut Microbiome Associates with Host Health’.

⁷ M. Azim Ansari and others, ‘Genome-to-Genome Analysis Highlights the Effect of the Human Innate and Adaptive Immune Systems on the Hepatitis C Virus’, *Nature Genetics*, 49.5 (2017), 666–73 <<https://doi.org/10.1038/ng.3835>>.

Question 2.1. The SV typing needs more detail, I looked at the previous reference but found it insufficient. This is key to the study. Can the authors address specifically: can copy number calls be affected by taxon abundance? Is it possible that deletions are just missed sequence (misassemblies, low DNA, other artefacts). What are the false positive and false negative rates of this calling approach? How are genes or regions for the SVs defined, are these clusters of orthologous genes, or something else?

I also don't understand the rationale for looking at only SVs, why weren't SNPs included?

As a more minor point, typically this would be called gene variation and copy number variation - structural variation is more usually used when referring to changes in gene order.

Response 2.1: We used the tool SGV-Finder to detect gut microbial SVs from metagenomic data. SGV-Finder detects SVs based on read alignment to the reference genomes, and we indeed refer to the original paper⁸ for its technical aspects. Specific details of this method and the code are available at <https://github.com/segalab/SGVFinder>.

In brief, the SV-calling procedure includes two major steps: (1) resolving ambiguous reads with multiple alignments according to mapping quality and genomic coverage using the iterative coverage-based read assignment algorithm (ICRA) and reassigning ambiguous reads to the most likely reference with high accuracy and (2) splitting the reference genomes of each microbial species into genomic bins and examining the coverage of genomic bins across all samples. For determination of dSVs within each species, the genomic bins are classified as deleted or retained in each sample, and those that are deleted in 25–75% of samples are kept in the analysis as raw dSVs. The raw dSVs that are highly correlated in co-occurrence are further merged into larger SV regions and result in the final dSV profile. For the determination of vSVs within each species, the coverage of genomic bins within each sample is standardized using the Z-score approach. Each bin is then assessed across all samples, and those that are highly variable are kept as raw vSVs. The raw vSVs that are highly correlated in standardized coverage are further merged into large SV regions and result in the final vSV profile.

We did observe that 71.8% of vSVs are associated with corresponding species abundance, thus we included the species abundance as a covariate in the GWAS analysis model to remove the confounding effect of species abundance.

We cannot fully exclude the possibility of false discovery of genomic deletions, but several factors that may affect the detection of SVs have been considered in the analysis. First, the developers of SGV-Finder specifically optimized the read mapping procedure to decrease the rate of misalignment of ambiguous reads that can be aligned to the orthologous genomic regions of multiple species using the ICRA algorithm and demonstrated that it effectively improved the accuracy of ambiguous read alignment based on benchmark tests. Second, to avoid bias caused by low amounts of DNA and low species abundance, SGV-Finder first assesses the mean coverage across the genomes for each species and each sample. It then only detects the SVs in microbial species with sufficient coverage in the sample, excluding those with low abundance or coverage from the analysis. Third, the total sequencing depth may also affect the detection of gut microbial SVs, thus we also included the clean read count

⁸ Zeevi and others, 'Structural Variation in the Gut Microbiome Associates with Host Health'.

as a covariate in the GWAS analysis.

We defined the genes that belong to the SV region as follows: the genomic coordinates of SVs are expanded 1kbp to the upstream and downstream flanking regions, and the genes that overlap with the expanded genomic region were considered to be the genes involved in the corresponding SV. Since the ambiguous reads that can be aligned to the orthologous genes of different species have been solved by the ICRA algorithm, the genes that are assigned to the corresponding SVs are species-specific.

Microbial SNVs have great potential to impact bacterial functionality and host-microbe interaction by modifying the gene product. Newly developed k-mer-based approaches such as GT-pro⁹ allow for fast and accurate metagenotyping of over 100 million bacterial SNVs. However, functional annotation of these microbial SNVs remains very challenging. Moreover, the burden of multiple testing is a major issue for genome-to-genome associations. SVs represent long genomic segments that often contain multiple bacterial genes and even complete, very large gene clusters, so microbial SVs can have a big impact on the gene content of bacteria and affect bacterial functionality. Thus, we first prioritized SVs to investigate the genome-genome associations.

We agree with the reviewer that the terminology is confusing. We call all genomic variations we identified "structural variants" (SVs) to be consistent in terminology with the paper of the SGV-Finder authors¹⁰. Copy number variations (CNVs) are usually considered a subtype of SVs and include insertions, deletions and duplications. Zeevi *et al.* explained that they call these genomic variants "structural variants" to specifically differentiate them from CNVs that are often used to describe specific genes present in a variable number of copies in one genome. As we are analyzing metagenomic data, the variation in genomic coverage we observe may result from the presence or absence of the genomic region in multiple strains, which can exhibit varying abundances within a gut microbial community. However, other SV types such as inversions and translocations cannot be detected using SGV-Finder. De novo assembly of large contigs may help in their identification. This was confirmed by our whole-genome sequencing and *in vitro* growth experiment of 12 selected *F. prausnitzii* representative strains, where we detected an inversion in one strain, ATCC27768, in which the GalNAc pathway seems to be deactivated.

We have now added this into the Discussion and acknowledge the limitations of the study.

Line 508-513 (Discussion):

"First, we focused on the common dSVs and vSVs in gut microbial genomes, assessed based on the abundance and distribution of short reads mapped along bacterial genomes. Our study did not capture other types of SVs, such as inversions and translocations, whose comprehensive identification will require whole-genome resequencing and de novo assembly of short or, ideally, long reads. Nonetheless, we could show that genomic inversion could result in dysfunction of the GalNAc pathway."

⁹ Zhou Jason Shi and others, 'Fast and Accurate Metagenotyping of the Human Gut Microbiome with GT-Pro', *Nature Biotechnology*, 40.4 (2022), 507–16 <<https://doi.org/10.1038/s41587-021-01102-3>>.

¹⁰ David Zeevi and others, 'Structural Variation in the Gut Microbiome Associates with Host Health', *Nature*, 568.7750 (2019), 43–48 <<https://doi.org/10.1038/s41586-019-1065-y>>.

Line 514-519 (Discussion):

“Second, our study did not include other types of genetic variation, such as single nucleotide variants (SNVs), which have great potential impact on bacterial functionality and host–microbe interaction. However, analyzing genetic associations across the millions of SNVs in the human genome and the hundreds of millions of SNVs in the metagenome would require a much larger sample size. Moreover, functional annotation of SNVs is still challenging.

We have also included the explanation and description of SV calling from metagenomic data in the Methods section.

Line 153-158 (Results): *“We used SGV-Finder¹⁵ to generate SV profiles. In brief, this method firstly mapped sequencing reads to reference genomes and resolved possible ambiguous read alignments using different algorithms, then the microbial genomes were split into bins and the metagenomic coverage of these bins were compared across samples (see **Methods**). SGV-Finder then identifies bins whose coverage is close to 0 in 25–75% of samples (dSVs) and bins that show variable coverage (vSVs).”*

Line 836-854 (Methods):

“In brief, the SV-calling procedure includes two major steps: (1) resolving ambiguous reads with multiple alignments according to the mapping quality and genomic coverage using the iterative coverage–based read assignment algorithm and reassigning ambiguous reads to the most likely reference with high accuracy and (2) splitting the reference genomes of each microbial species into genomic bins and examining the coverage of genomic bins across all samples. For the determination of dSVs within each species, the genomic bins are classified to the deleted (coverage close to 0) or retained (coverage close to median coverage of the genome) bins in each sample, and those that are deleted in 25–75% of samples are kept in the analysis as raw dSVs. The raw dSVs that are highly correlated in co-occurrence are further merged into larger SV regions to produce the final dSV profile. For the determination of vSVs within each species, the coverage of genomic bins within each sample are standardized using the Z-score approach. Each bin is then assessed across all samples, and those that are highly variable based on a beta-prime distribution are kept as raw vSVs. The raw vSVs that are highly correlated in standardized coverage are further merged into large SV regions to produce the final vSV profile.

To define the genes that belong to the SV region, we expanded the genomic coordinates of SVs 1kbp upstream and downstream, with the genes that overlap with the expanded genomic region considered genes that belong to the corresponding SV.”

Question 2.2. Rather than calling variants for each species, did the authors consider associating gene clusters via their function irrespective of species, which would likely be more powerful. For example, looking at GalNAc clusters in the entire metagenome of each sample would be an obvious extension. But taking a functional/gene-based association approach from the beginning could increase the number of signals found.

Response 2.2: Some relevant work has been done in previous studies by us and by others that linked the gut microbial metabolic pathways to human genetics (e.g. Lopera-Maya *et al*¹¹). However, those analyses were heavily biased toward the bacterial pathways and gene clusters that have been already annotated. But current databases are rather incomplete, and many important pathways are either unannotated or incomplete. As a result, previous studies reported *ABO* association with the lactose and galactose degradation pathway rather than the

¹¹ Esteban A. Lopera-Maya and others, ‘Effect of Host Genetics on the Gut Microbiome in 7,738 Participants of the Dutch Microbiome Project’, *Nature Genetics*, 54.2 (2022), 143–51
<<https://doi.org/10.1038/s41588-021-00992-y>>.

GalNAc pathway. We noticed that the *ABO*-associated SV region not only contains the GalNAc pathway but also some of the genes relevant to galactose metabolism, which is annotated in the database. This demonstrates the novelty of our use of bacterial SVs to pinpoint unknown or poorly annotated functional genes/pathways.

Genetic associations of gene abundances would be an unbiased approach. As demonstrated in our results for genetic associations of GalNAc gene abundances profiled using the ShortBRED tool, their associations are indeed stronger than the association of SVs. It also provides extra evidence to support GalNAc-related genes as the underlying causal genes in the *ABO*–SV association and demonstrates the potential of this approach for pinpointing causal genes. SNV-based association analysis is clearly another way to pinpoint potentially relevant genes, but systematic association of over 300 million bacterial genes¹² and over 100 million SNVs¹³ is not feasible given the computational complexity, multiple testing burden and the sample size of the current study.

Question 2.3. The assays of growth presented in figure 4 are broadly convincing and have the advantage of using a variety of strains. However as far as I can tell an effect of strain genetic background has not been ruled out, and an ideal experiment to add would be a knock out of the GalNAc gene cluster vs the wild-type strain, and calculating a competitive index on each of the two media conditions.

Response 2.3: We thank the reviewer for this suggestion. An experimental knockout of the GalNAc genes is challenging, as there is currently no efficient genome modification strategy for *F. prausnitzii*. *F. prausnitzii* lack plasmid in their cells and seem not to take exogenous DNA fragments. Our attempts to insert a plasmid or linear DNA failed, making CRISPR-Cas modification currently impossible. Gene modification might be possible via phages or transposons, but the experiment would take a long time as we would need to design and validate phage- or transposon-based techniques for an efficient knockout.

Thus, instead of directly knocking out GalNAc pathways, we performed additional bioinformatics and gene expression analysis to prove that the GalNAc utilization is independent of the genetic background.

Firstly, the phylogenetic tree already showed that the presence of SV 577–579 is independent of the phylogenetic relationship of the strains (**Figure 3C**).

¹² Baoli Zhu, Xin Wang, and Lanjuan Li, 'Human Gut Microbiome: The Second Genome of Human Body', *Protein & Cell*, 1.8 (2010), 718–25 <<https://doi.org/10.1007/s13238-010-0093-z>>.

¹³ Zhou Jason Shi and others, 'Fast and Accurate Metagenotyping of the Human Gut Microbiome with GT-Pro', *Nature Biotechnology*, 40.4 (2022), 507–16 <<https://doi.org/10.1038/s41587-021-01102-3>>.

Figure 3C: Phylogenetic tree of the strains we used in vitro culture experiments and the organization of genes involved in GalNAc utilization in the SV region. The x-axis indicates the base pair position starting from the flanking gene *dinB*. The different colored lines indicate the location of same genes in different strains. Names of strains with the SV region are in red. Names of strains without the region are in black.

Secondly, to further validate the function of the GalNAc utilization cluster, we tested the expression of the GalNAc utilization genes of two closely related strains, HTF-495 (with GalNAc pathway) and HTF-441 (without GalNAc pathway), in glucose and GalNAc media. None of the GalNAc-related gene transcripts were detected in HTF-441, not even in glucose medium, confirming the absence of these genes. An expression-induction experiment using GalNAc showed that the expression of GalNAc genes was significantly up-regulated in the GalNAc medium. The *agaC* gene responsible for GalNAc uptake showed the largest response: a 63.5-fold increase in its expression in GalNAc medium compared to that in glucose medium. This suggests that *agaC* might be a key gene for GalNAc uptake.

Thirdly, we also performed expression-induction for the ATCC27768 strain where the SV region was reversed and found that this strain cannot use GalNAc as the carbon source. Its genetic background was similar to HTF-495. Interestingly, we observed that the expression of GalNAc-uptake genes in ATCC27768 (*agaC*, *agaD* and *agaV*) were not induced by GalNAc (Figure 3F), although the fold-change of *agaF* was similar between ATCC27768 and HTF-495.

Figure 3F. Fold change of gene expression upon GalNAc induction compared to glucose induction. The y-axis shows the fold change of the *agaC*, *agaD*, *agaV*, and *agaF* genes in HTF495 and ATCC27768 upon GalNAc induction relative to glucose induction. Each dot indicates one replicate, and the bar indicates the standard error of three replicates.

Altogether, we are confident that the growth of GalNAc-containing species in GalNAc medium was not influenced by their genetic background but is rather dependent on the presence and activation of GalNAc genes.

Question 2.4. The result in figure 5 on overall diversity seems too good to be true: double the richness/diversity is observed between GalNAc low/high individuals. Are there other possible host factors that could contribute to this large difference? Any other possible confounders to control for? It would also be interesting to see whether any host associations occur when associated against overall gut metagenome diversity.

Response 2.4: We apologize for this confusion. The figure compared the correlation coefficients between GalNAc genes and bacterial richness and diversity in individuals with or without mucosal A-antigens. In other words, the correlation coefficient is nearly doubled, but not the richness and diversity themselves. The richness was actually marginally significantly different between individuals with and without mucosal A-antigens (Wilcoxon rank-sum test, $P = 0.013$) and diversity was not significantly different between two groups (Wilcoxon rank-sum test, $P = 0.335$). To avoid this confusion, we have now visualized this result (**Figure 5C**) using a scatter plot where each dot represents an association strength (correlation coefficient R-value) between a GalNAc gene and gut microbiome diversity or richness in individuals with (x-axis) and without mucosal A-antigen (y-axis). If the dots fall away from the diagonal line, it suggests a difference in the correlation coefficients between the two groups. We also assessed this difference using a heterogeneity test and observed that the correlations are stronger in the individuals with mucosal A-antigen compared to those without ($P_{\text{heterogeneity}} < 0.05$).

Figure 5C. Comparison of the correlations of GalNAc metabolism gene abundance with gut microbiome alpha diversity and richness between two groups: individuals with mucosal A-antigen (x-axis in red) and those without (y-axis in blue). Each dot represents a Spearman correlation coefficient between a GalNAc gene with the Shannon index (green) or richness (blue). The error bar indicates the confidence interval of the correlation R estimation.

Minor points

Question 2.5. Using permutation testing to determine a significance is likely to be inappropriate as the samples are not fully IID. Switching labels within genotype groups (in the metagenome) might be more appropriate, or just a FWER correction. So those results at the borderline of significance are not convincing on their own. QQ plots would also be helpful to assess this.

Response 2.5 We thank the reviewer for this suggestion. We have now used FWER (Bonferroni correction) (5×10^{-08} / number of SVs) as the significance threshold. This resulted in a $P = 2.65 \times 10^{-11}$ cut-off for vSV associations and a $P = 3.00 \times 10^{-11}$ cut-off for dSVs. Indeed, using these more stringent cut-offs, only the associations between ABO and *F. prausnitzii* SVs remain significant. We also have added the QQ plots for the significant associations to the Supplementary figures (**Figure S3**).

Figure S3. Q-Q plots of the associations between human SNPs and *F. prausnitzii* SVs

X-axis is the expected log-transformed p-value. Y-axis is the observed log-transformed p-value. Each dot represents an association p-value. The diagonal is represented by the solid red line. The lambda (λ) value represents the genomic inflation factor.

Question 2.6. The study is referred to as a gut microbiome study, all the samples are fecal. Taxon (and SVs) in fecal samples aren't necessarily representative of the gut as a whole. This should be mentioned somewhere

Response 2.6: We agree with the reviewer on this comment and have now discussed this limitation in the Discussion:

Line 519-524 (Discussion):

“The third limitation of the current study is related to the use of fecal microbiota data to represent the gut microbiome. It is important to note that the microbiome is not entirely the same across the different intestinal compartments, and further investigation into the microbiome of different gastrointestinal tract segments and mucosal layers would provide a more comprehensive landscape of host–microbe genetic crosstalk⁴⁸.”

Question 2.7. Checking the covariance of the SVs could be useful. If many of them are colinear or closely related, a latent factor analysis of the SVs may be more powerful (for example, PCA, non-negative matrix factorisation or the PEER method. An example of this: <https://www.nature.com/articles/s41586-020-2635-8>)

Response 2.7: During SV detection, SVs that are closely related are merged into one. This is the default setting of SGV-Finder and is reflected in the ID of SVs. For example, a dSV called “*Bifidobacterium longum*:367_368 and 7 segments” covers 8 genomic regions in

Bifidobacterium longum that are closely related. Thus, after this step, the SV profiles no longer present many closely related SVs.

While we agree that factor analysis is useful for identifying genetic loci associated with multiple closely related phenotypes, it would not report the association with individual phenotypes (SV regions in our case). In this study we wanted to detect specific bacterial genomic regions that may harbor functional genes involved in the host–microbiome interaction. Therefore, we needed the association results for specific SVs. Moreover, bacterial SV data is very sparse, and factor analysis is not very well suited to such sparse matrices. Thus, with respect, we believe that latent factor analysis is not very suitable for this study.

Question 2.8. Relatedly, population structure of the bacterial metagenomes was not mentioned. Are the SVs sufficiently controlled for this? You could check LD between them.

Response 2.8: Thank you for this suggestion. We have now focused on *F. prausnitzii* as it is the only one that passes the Bonferroni correction. We first checked its population genetic structure by running principal coordinate analysis on its SV profile, and the top two PCs only explained 5.6% and 4.4% variation, respectively. There is no clear sample clustering based on the SV profile. However, we did detect significant associations between *ABO*-associated SVs and the top PCs ($P < 0.05$). This result is now presented in **Figure S6**.

Figure S6. Population genetic structure of F. prausnitzii and its association with ABO-associated SVs. The x-axis and y-axis show the top two PCs. Each dot represents a sample. In each panel figure, dots are colored differently based on the presence/absence status of dSVs or the level of vSVs following the color key shown in the figure.

We further adjusted the population structure of *F. prausnitzii* by adding the top five PCs as covariates but found that the impact of population structure on the *ABO*–SV association is

limited (**Table S11**). Now we have added this finding to the Results:

Line 248-253 (Results):

*“Although the ABO-associated SVs were correlated with the population genetic structure of *F. prausnitzii* (Table S10; Figure S6), *F. prausnitzii* associations with the ABO locus remained significant even after correcting for the first five principal components of the population structure of *F. prausnitzii* ($P_{DMP} = 3.91 \times 10^{-33}$ before population structure adjustment, $P_{DMP} = 2.39 \times 10^{-32}$ after population structure adjustment; Table S11).”*

Table S11: Comparison of ABO - *F. prausnitzii* SV association before and after population structure correction.

SV_id	SV_name	SNP_rsId	SNP_position	Effect_allele	Other_allele	N_DMP	No_population_structure_correction		With_population_structure_correction	
							Beta_DMP	pvalue_DMP	Beta_DMP	pvalue_DMP
F.prausnitzii	Faecalibacterium cf. prausnitzii KLE1255:577_579	rs635634	9:136155000	T	C	4318	0.8392	3.9064E-33	0.835388	2.39E-32
F.prausnitzii	Faecalibacterium cf. prausnitzii KLE1255:1154_1155	rs550057	9:136146597	T	C	4318	0.5973	1.8319E-25	0.618177	1.15E-23
F.prausnitzii	Faecalibacterium cf. prausnitzii KLE1255:575_577	rs550057	9:136146597	T	C	4318	0.5589	3.0965E-22	0.579756	4.32E-22
F.prausnitzii	Faecalibacterium cf. prausnitzii KLE1255:1154_1155	rs41302667	9:136330428	A	G	4318	0.5845	4.0144E-10	0.622975	7.37E-10
F.prausnitzii	Faecalibacterium cf. prausnitzii KLE1255:2910_2911	rs550057	9:136146597	T	C	4318	0.3544	2.8020E-10	0.339163	1.05E-08
F.prausnitzii	Faecalibacterium cf. prausnitzii KLE1255:885_887	rs07666	9:136149399	A	G	4317	0.2249	8.5000E-16	0.225595	1.76E-21

Question 2.9. Were the results in figure S2 corrected for multiple testing in the same way as the main analysis?

Response 2.9: Figure S2 (now Figure S5) reports the original statistical p-value before adjusting for multiple testing. Here we zoom into the ABO–SV associations to investigate the association with ABO-determined blood type and the effect of FUT2 secretor status in different cohorts. Thus, no multiple testing is needed. In the revised manuscript, we now use the significance cutoff determined by the Bonferroni approach (5×10^{-08} / number of SVs) for genome-wide association analysis. This resulted in a $P = 2.65 \times 10^{-11}$ cutoff for vSV associations and a $P = 3.00 \times 10^{-11}$ cutoff for dSVs.

Question 2.10. The github with the code needs more detail in the README to describe what the files included actually do.

Response 2.10: We thank the reviewer for the suggestion. We have added a README file and more description in the code, please see: https://github.com/GRONINGEN-MICROBIOME-CENTRE/SV_GWAS.

Question 2.11. The legend of figure 2 does not contain enough detail to describe the figure. In particular, is this all the associations of the SVs overlaid at each human chromosome position?

Response 2.11: Yes, these are all associations of all SVs. We have now expanded the legend of **Figure 2**, which now reads:

Figure 2. Manhattan plot of genome-wide associations of human SNPs and gut microbial SVs. The upper and lower panels show GWAS results for dSVs and vSVs, respectively. The x-axis shows the genomic position of human chromosomes (chromosomes 1–22) for both the upper and lower panel. The y-axes in both panels show statistical significance as $-\log_{10} P$ value. Red horizontal lines indicate the study-wide significance cutoffs determined using the Bonferroni method: 3.00×10^{-11} for dSV and 2.65×10^{-11} for vSV associations. Significantly associated loci are highlighted in yellow, labeled with the nearby human gene and the corresponding species name.

Question 2.12. Is *F. prausnitzii* transmitted between people? Mothers and children? In light of this, would there be any selection pressure to either a) lose the gene cluster in question or b) retain the cluster to remain more plastic, in case a GalNAc high host is encountered?

Response 2.12: We thank the reviewer for bringing up this interesting question. It has been previously shown by Valles-Colomer *et al.* that *F. prausnitzii* can be transmitted from mothers to infants and within one household¹⁴. As our DMP cohort includes individuals from 2,756 families, we also assessed whether individuals living in the same household were more likely to share this SV region, independent of their blood group. Indeed, we do find that individuals in our cohort are more likely to have the *F. prausnitzii* SV 577–579 if they co-house with people with this SV (Figure S17), which suggests strain-sharing via co-housing.

¹⁴ Mireia Valles-Colomer and others, 'The Person-to-Person Transmission Landscape of the Gut and Oral Microbiomes', *Nature*, 614.7946 (2023), 125–35 <<https://doi.org/10.1038/s41586-022-05620-1>>.

Figure S17. Fraction of samples with or without SV 577–579 dependent on household members

Each bar represents the fraction of samples with the GalNAc-containing 577–579 dSV region in individuals with/without A-antigen and with/without household members with the dSV.

In addition, we checked for any evidence of horizontal gene transfer that may drive the gain/loss of the GalNAc utilization gene cluster. Our hypothesis was that if GalNAc genes are transferred between different strains, their genetic distance should be smaller than the genetic distance of the strains based on species marker genes. We thus constructed a phylogenetic tree based only on the GalNAc metabolism genes and found that it is distinct from the phylogenetic tree constructed based on species marker genes (Figure S15A and B). Moreover, the phylogenetic distance is smaller in GalNAc metabolism genes compared to phylogenetic distance of marker genes (Figure S15C), which supports the mobilizability of GalNAc utilization gene cluster.

Figure S15. Different phylogenetic distances between *F. prausnitzii* strains based on marker genes and GalNAc genes

A, Phylogenetic trees based on the phylogenetic distance of marker genes (left) and GalNAc genes (right). B, Scatter plot of between-strain genetic distance based on marker genes (x-axis) and GalNAc genes (y-axis). Each dot represents pairwise phylogenetic distance between two strains. C, Comparison of between-strain genetic distance based on marker genes and GalNAc genes. Each box plot represents the distribution of pairwise phylogenetic distances of strains based either on GalNAc genes or on species marker genes. The center line of the box is the median, the box hinges are the lower and upper quartiles of the distribution, and each dot represents a pair of two strains.

Based on this observation, we also predicted whether the mobilizable genomic islands exist in the GalNAc gene region of *F. prausnitzii*. For six out of seven strains, a least one genomic island was predicted to be in or near GalNAc genes by at least one of the two prediction methods we used (Figure S16).

Figure S16. Prediction of mobilizable genomic islands in the GalNAc region of *F. prausnitzii*

Mobilizable genomic islands (GIs) predicted for the seven GalNAc-containing *F. prausnitzii* strains. For ATCC27768, the analysis was done for both original and reserved directions. Only the region containing GalNAc and its surrounding region are shown, along with their genomic base pair positions. The GalNAc genes are labeled and colored differently based on the transcription directions: green indicates forward transcription and red indicates backward transcription. To reduce false predictions, all genome contigs were checked by aligning them against a reference genome *F. prausnitzii* strain Indica, and the regions in HTF-383 that cannot be mapped to the reference genome are colored in gray. GI regions predicted by IslandPath-DIMOB are indicated in blue. GI regions predicted by SIGI-HMM are indicated in orange.

Regarding the loss or gain of GalNAc-containing region, we compared the presence/absence status of 577–579 dSV region in 119 individuals for whom *F. prausnitzii* SV profiles were measured at two time points 4-years apart. The presence/absence status of the dSV region did not change for 85 (71.4%) individuals, while 23 individuals gained the region and 11 individuals lost the region (McNemar's one-sided test, $P = 0.029$, **Figure S18**).

Figure 18. Gain or loss of the *F. prausnitzii* dSV 577–579 region after 4 years

Alluvial plot shows the number of individuals with and without the 577–579 region in *F. prausnitzii* at two time-points 4-years apart. Number of individuals with the same presence or absence status for this region after 4 years are marked in gray. Number of individuals that gained the region are marked in red, and the number of who lost the region are marked in blue.

Altogether, our findings suggest that the GalNAc utilization gene cluster region is potentially mobilizable and may be transmitted between bacteria and host. Moreover, individuals seem more likely to gain the region than to lose the region over time.

We now have added the description of the result of the strain transmission and mobilizability analysis to the text:

Line 415-435 (Results):

“Evidence for bacteria-to-bacteria and human-to-human transmission of GalNAc genes

*We further wondered whether the GalNAc-containing SV region could be transmitted between bacteria or between hosts. We first noticed that the SV region is not lineage-specific and can be seen in all major phylogenetic branches of *F. prausnitzii* in a tree constructed based on conservative marker genes (see Methods) (Figure 3C). This was further confirmed by a clear discrepancy between the phylogenetic distances of GalNAc utilization genes compared to those of species marker genes (Figure S15A and B; phylogenetic distance = 0.73): the divergence of the GalNAc utilization genes was significantly smaller than the divergence of *F. prausnitzii* marker genes (Figure S15C). Moreover, we identified genomic islands in the regions flanking the GalNAc utilization gene cluster (Figure S16). All these findings support the potential mobilizability of the GalNAc pathway.*

*As recent evidence supports transmission of *F. prausnitzii* between hosts³¹, we also assessed whether individuals living in the same household with individuals with the SV were also more likely to have this SV, independent of their blood group. Our analysis identified a significant effect of household SV presence, and this effect was independent of the host’s genetic background (odds ratio = 2.29, $P = 3.19 \times 10^{-7}$) and additive to the genetic background effect (odds ratio = 3.27, $P < 2 \times 10^{-16}$; Figure S17). We also compared the *F. prausnitzii* SV profiles in 119 individuals at two timepoints 4-years apart and found that the presence/absence status of the dSV region did not change for 85 (71.4%) individuals, while 23 individuals gained the region and 11 individuals lost the region (McNemar's one-sided test $P=0.029$, Figure S18).”*

Question 2.13. The panel labels in figure 3 don't all align with the caption.

Response 2.13: We apologize for the mismatched panel labels. We have now corrected the labels.

Question 2.14. Figure 4B -- it is unclear what the relevance of all the different ways that steps can be missing here is. Is it just that the pathway is complete or incomplete?

Response 2.14: Here we describe the number of strains per species with the completed GalNAc pathway. We classified the GalNAc pathway with an initial cleavage step of GalNAc from A-antigen by one gene (*GH109*) that encodes a glycoside hydrolase and five key metabolic steps of downstream GalNAc utilization involving nine genes. Missing genes encoding any of these steps may affect GalNAc utilization. But the genes involved in the GalNAc pathway could be also involved in other metabolic pathways, e.g., *lacC* is also involved in galactose metabolism. We thus focused on the species with the complete GalNAc pathway. We have rephrased the legend of **Figure 4B** to make this clear.

Question 2.15. The legend of figure 4E-H is poor, and make it hard to follow what the plot shows.

Response 2.15: We thank the reviewer for pointing this out. We have now expanded the legend to make the figure self-explanatory and easy to follow. The legend now reads:

Figure 4. GalNAc utilization capacity of strains of *F. prausnitzii* and other ABO-associated species. **A**, ABO-associated taxa reported in human cohorts. **B**, Completeness of the GalNAc pathway in four gut microbial species associated with human ABO blood type: *F. prausnitzii*, *Collinsella aerofaciens*, *Faecalicatena lactaris*, and *Bifidobacterium bifidum*. The upper bar plot shows the number of strains (intersection size), and the combination of black dots in each vertical column underneath represents the presence of the genes in the corresponding step. The bar plot on the left represents the number of strains (set size) containing the genes of each corresponding GalNAc metabolism step. **C–D**, Proportion of strains with a complete GalNAc pathway in *F. prausnitzii* (**C**) and *C. aerofaciens* (**D**). **E–H**, Growth curves of *C. aerofaciens* strains containing GalNAc pathway genes on media supplemented with different sugars. The x-axis indicates the hours after culturing in the medium. The y-axis indicates cell density measured as OD_{600} value. Points on the growth curves represent means \pm SD (standard deviation) of three replicates. Yeast Casitone Fatty Acids (YCFA) medium and YCFA-glucose (Glc) medium were used as negative and positive controls, respectively. GalNAc and Galactose (Gal) were supplied in YCFA medium to test if *C. aerofaciens* can grow with the monosaccharides released from A-antigen and B-antigen as their sole carbohydrate sources.

Question 2.16. I am not suggesting this is changed, but in general I am surprised to see that a fixed effects GWAS model with plink was used. A linear mixed model approach would both better control for ancestry and relatedness, and improve power by not removing samples.

Response 2.16: We thank the reviewer for this comment. We have now updated the GWAS results using fastGWA, which is based on linear mixed models¹⁵, and the results are very comparable. Only the *ABO*-*F. prausnitzii* SV associations passed the Bonferroni correction.

¹⁵ Longda Jiang, Zhili Zheng, Ting Qi, and others, 'A Resource-Efficient Tool for Mixed Model Association Analysis of Large-Scale Data', *Nature Genetics*, 51.12 (2019), 1749–55 <<https://doi.org/10.1038/s41588-019-0530-8>>; Longda Jiang, Zhili Zheng, Hailing Fang, and others, 'A Generalized Linear Mixed Model Association Tool for Biobank-Scale Data', *Nature Genetics*, 53.11 (2021), 1616–21 <<https://doi.org/10.1038/s41588-021-00954-4>>.

Reviewer #3:

This study by Zhernakova *et al.* reports the results of a large-scale analysis of human genetic polymorphisms that correlate with the presence/absence of gut bacterial structural variants (SVs). The study revealed 18 significant associations and the authors subsequently focused on a connection between the ABO locus, which adds terminal alpha3-linked GalNAc (blood group A) or alpha3-linked Gal (blood group B) to blood group glycans but, importantly, also to secreted mucus glycans. The authors also made a further connection to the function of the Fut2 locus which is responsible for adding terminal alpha2-linked fucose to mucus glycan chains and this modification is a required precedent for addition of blood groups A/B. Thus, it is intuitive that association between the ABO locus and certain bacterial pathways would only be meaningful if the host has a functional Fut2 and can actually execute the A/B modification pathway via the ABO locus. A specific connection is established between a locus in certain *Faecalibacterium prausnitzii* (Fp) strains and the presence of blood group A. A second connection is made between blood group A and *Collinsella aerofaciens* (Ca) and the authors provide some support for a functional role by demonstrating that Ca can utilize GalNAc as a nutrient and Fp strains that have a particular locus encoding GalNAc utilization genes (i.e., the identified SV) also mostly can use GalNAc, although one strain that had a locus couldn't and one grew very poorly making this correlation imperfect.

As the authors cite, there have been 3 previous studies connecting variations in the ABO locus and abundance of certain gut bacterial taxa, including one study involving the authors of this paper (all published in Nature Genetics). The major advance in this work is to make the connection to the presence of genes in Fp and not just the abundance of this or other organisms. The large human cohort is a strong feature of this study and the sequencing/informatics seem sound, although this is not my area of expertise, which is instead in microbiome function, mucus and glycobiology. In this area, I'm left with the feeling that the connection to GalNAc utilizing bacteria is less complete and this is elaborated on in specific points below. There are many other bacteria from the human gut microbiome that are capable of utilizing GalNAc, so it doesn't seem that a specificity for Fp's ability to utilize this sugar is the only cause of its association with the ABO/Fut2 loci. On the microbiology side, the imperfect correlation between strains that have the Gal/GalNAc utilization locus would be made stronger if all the strains with the locus grew well on GalNAc or the authors demonstrated that the strains that have it and still grow poorly truly have inactivated the functions (e.g., due to inversion of the locus). The more exciting path would be to pursue some of the additional human genome polymorphisms that are associated with other bacteria. Although, if this has not been started or revealed mechanistic connections yet could be a long way off.

Response: We thank the reviewer for these comments. Indeed, *F. prausnitzii* is not the only GalNAc utilizer in the gut. Previously, three large-scale human cohort studies revealed ABO association with the abundance of three other species: *Collinsella aerofaciens*, *Faecalicatena lactaris* and *Bifidobacterium bifidum*. By zooming in on their genomes, we also detected GalNAc genes in their genomes. However, the GalNAc pathway in those species does not localize in a SV region, thus we do not observed ABO–SV associations for these species. In

contrast, the GalNAc pathway genes in *F. prausnitzii* fall in the SV region and thus we observed a strong association between *ABO* genetic variants and GalNAc-containing SVs.

Following the reviewer's suggestion, we have also tested expression of GalNAc genes in ATCC27768, in which the region is inverted. We also agree that it is interesting to pursue additional polymorphisms associated with other bacteria. But, as the reviewer has noted, other signals are less convincing, and only the *ABO* association passed the Bonferroni-corrected $P < 0.05$ level. Thus, we still focus on the *ABO* locus in this study but have now conducted extra experiments. Please see the details in our responses to specific points below.

Specific points:

Question 3.1. The introduction or early results might benefit from a more specific description of what is being counted as an SV without going to methods or previous literature. I looked at this in the methods and realize that a pipeline from another group (Zeevi *et al*) is used but I only had a vague idea of what these "variations" might be until seeing Figure S4, but then was confused as to why the 577-79 segment showed up so strong despite looking very similar among strains in that region on the "right" side of the locus, while the adjacent region to the left is much more striking in how it reflects presence or absence of potentially causal genes and actually looks like a deletion event in some strains (or a gain of function by lateral gene transfer or another mechanism). Duplication/deletion events are discussed heavily in the introduction. Deletions in some lineages could certainly account for loss of function. But duplications of existing genes and subsequent drift to establish new functionality don't seem like the most common or quickest way to evolve new functions (this is also not my area of expertise). Rather, existing functions might be acquired by lateral gene transfer. This might just need some re-wording to note that the presence/absence of genes are what is being correlated with less description of how they are obtained or loss, unless its more clearly known.

Response 3.1:

We appreciate the feedback provided by the reviewer. We have added a brief description of SV calling at the beginning of the Results section to improve the clarify of the study and the interpretation of SVs.

Line 153-158 (Results):

*"We used SGV-Finder¹⁵ to generate SV profiles. In brief, this method firstly mapped sequencing reads to reference genomes and resolved possible ambiguous read alignments using different algorithms, then the microbial genomes were split into bins and the metagenomic coverage of these bins were compared across samples (see **Methods**). SGV-Finder then identifies bins whose coverage is close to 0 in 25–75% of samples (dSVs) and bins that show variable coverage (vSVs)."*

Secondly, as the reviewer has pointed out, our comparison of the whole genomic sequences of 12 *F. prausnitzii* isolates revealed that the right regions of the 577-579kb region exhibit high similarity among all strains. However, the presence or absence of genes in the left region corresponds to the presence or absence of the 577-579kb region. Based on this observation, we consider the 577-579 SV region as a representative proxy for the entire region, and the entire deletion region extends to approximately ~23kb. Within this extended SV region, we

have identified the presence of the GalNAc pathway.

This region can either be gained or deleted in certain strains. In light of this, we hypothesize that this region might be a mobile element. To investigate this hypothesis, we conducted several analyses. Firstly, previous studies by Valles-Colomer et al. have demonstrated the transmission of *F. prausnitzii* from mothers to infants and within households. Since our DMP cohort comprises individuals from 2,756 families, we examined whether individuals residing in the same household were more likely to share this SV region, irrespective of their blood group. Our findings support this notion, as individuals co-habiting with people possessing this SV (Figure S17) exhibit a higher likelihood of harboring the *F. prausnitzii* SV 577_579, suggesting strain sharing through co-housing.

Figure S17. Fraction of samples with or without SV 577-579 dependent on the household members.

Each bar represents the fraction of samples with the GalNAc-containing 577-579 dSV region in individuals with/without A-antigen and with/without household members with the dSV.

In addition, we checked if we see any evidence of horizontal gene transfer which may drive the gain/loss of the GalNAc utilization gene cluster. Our hypothesis was that if GalNAc genes are transferred between different strains, their genetic distance should be smaller than the genetic distance of the strains based on species marker genes. We thus constructed the phylogenetic tree based on the GalNAc metabolism genes only and found that it is distinct from the phylogenetic tree constructed based on species marker genes (Figure S15A and B). Moreover, the phylogenetic distance is smaller in GalNAc metabolism genes compared to phylogenetic distance of marker genes (Figure S15C), which supports the mobilizability of GalNAc utilization gene cluster.

Figure S15. Different phylogenetic distance between *F. prausnitzii* strains based on marker genes and GalNAc genes. **A**, Phylogenetic tree based on phylogenetic distance of marker genes (left) and GalNAc genes (right). **B**, Scatter plot of between-strain genetic distance based on marker genes (x-axis) and GalNAc genes (y-axis). Each dot represents pairwise phylogenetic distance between two strains. **C**, Comparison of between-strain genetic distance based on marker genes and GalNAc genes. Each boxplot represents the distribution of pairwise phylogenetic distances of strains either based on GalNAc genes or species marker genes: the center line represents the median; the box hinges represent the lower and upper quartiles of the distribution; and each dot represents a pair of two strains.

Based on this observation, we also predicted whether the mobilizable genomic islands exist in the GalNAc gene region of *F. prausnitzii*. For all seven strains, a least one genomic island was predicted in or near GalNAc genes by at least one method (**Figure S16**).

Figure S16. Prediction of mobilizable genomic islands in the GalNAc region of *F. prausnitzii*

Mobilizable genomic islands (GIs) predicted for the seven GalNAc-containing *F. prausnitzii* strains. For ATCC27768, the analysis was done for both original and reserved directions. Only the region containing GalNAc and its surrounding region are shown, along with their genomic base pair positions. The GalNAc genes are labeled and colored differently based on the transcription directions: green indicates forward transcription and red indicates backward transcription. To reduce false predictions, all genome contigs were checked by aligning them against a reference genome *F. prausnitzii* strain Indica, and the regions in HTF-383 that cannot be mapped to the reference genome are colored in gray. GI regions predicted by IslandPath-DIMOB are indicated in blue. GI regions predicted by SIGI-HMM are indicated in orange.

Regarding the loss or gain of GalNAc-containing region, we compared the presence/absence status of 577-579 dSV region in 119 individuals for which *F. prausnitzii* SV profiles were measured at two time points at 4-year part. The presence/absence status of dSV region did not change for 85 (71.4%) individuals, while 23 individuals gained the region, and 11 individuals lost the region (McNemar's one-sided test $P=0.029$, **Figure S18**).

Figure 18: Gain or loss of *F. prausnitzii* dSV 577-579 region in four years.

The bar plot shows the number of individuals with the 577-579 region in *F. prausnitzii* at two time points at 4-year part. The fractions of individuals with the same presence/absence status of this region are marked in gray. The fraction of individuals gained the region marked in red, and the fraction of individuals lost the region marked in blue.

Altogether, our findings suggest that the GalNAc utilization gene cluster region is potentially mobilizable and may be transmitted between bacteria and host. Moreover, individuals seem more likely to gain the region than to lose the region over time.

We have added the result description for the strain transmission and mobilizability analysis to the text:

Line 415-435 (Result):

*“We further wondered whether the GalNAc-containing SV region could be transmitted between bacteria or between hosts. We first noticed that the SV region is not lineage-specific and can be seen in all major phylogenetic branches of *F. prausnitzii* that were constructed based on conservative marker genes (see **Methods**) (Figure 3C). This was further confirmed by a clear discrepancy between the phylogenetic distances of GalNAc utilization genes as compared to those of species marker genes (Figure S15A and B; phylogenetic distance = 0.73): the divergence of the GalNAc utilization genes was significantly smaller than the divergence of *F. prausnitzii* marker genes (Figure S15C). Moreover, we identified genomic islands at or near the GalNAc utilization gene cluster (Figure S16). All these findings supported the potential mobilizability of the GalNAc pathway.*

*As recent evidence has supported transmission of *F. prausnitzii* between hosts Valles-Colomer and others., we also assessed whether individuals living in the same household as individuals with the SV were more likely to have this SV, independently of their A-blood group. Our analysis identified a significant effect of household SV presence, and this effect was independent of the genetic background of the hosts (odds ratio=2.29, $P=3.19 \times 10^{-7}$). Moreover, this effect was seen to be additive to the genetic background effect (odds ratio=3.27, $P < 2 \times 10^{-16}$ (Figure S17). Furthermore, we compared *F. prausnitzii* SV profiles of 119 individuals measured at two time points 4-year apart. The presence/absence status of dSV region did not change for 85 (71.4%) individuals, while 23 individuals gained the region, and 11 individuals lost the region (McNemar's one-sided test $P=0.029$, **Figure S18**).”*

Question 3.2. No connection to blood group A cleaving bacteria or enzymes is attempted, and does it necessarily need to be in this study. But it's worth considering that GalNAc would need to be liberated in this context by a specific microbial enzyme and the producing organism might benefit the most due to attachment to mucus, proximity to released substrate or other "selfish" effects (exceptions also obviously exist in which bacteria release sugars for other). Glycoside hydrolase family 109 (GH109) enzymes have been implicated in cleaving blood group A and were discovered in part in human gut bacteria (*Bacteroides* species, plus others). It would be interesting to know if the identified *F. prausnitzii* and *Collinsella* strains have these enzymes or the activity when assayed biochemically (do they also vary concordantly with the identified GalNAc utilization locus?). If GH109 enzymes are not present, it would suggest that they rely on a different bacterium/enzyme to release GalNAc (or encode members of another family that does this catalysis, which could be tested quite simply using *Fp* bacterial lysates and commercially available substrates that harbor blood group A, looking for cleavage with a simple assay like thin layer chromatography). There is of course another relevant source of GalNAc in mucin glycans: the core alpha-linked GalNAc that forms the base of each O-linked glycan. Since this linkage needs to be present to extend the mucin glycan chains it must be stoichiometrically more abundant than blood group A but is more difficult to access. Perhaps conditional access to this core GalNAc in pigs vs. humans is part of the basis for discordance between the *fut2* and *ABO* loci, i.e., in *fut2* hosts there will still be core GalNAc present and perhaps different species are capable of accessing it in pig vs. human. Or, perhaps *Fut2* loss of function polymorphisms don't naturally exist in pigs?

Response 3.2: We have now searched for the presence of glycoside hydrolase family 109 (GH109) in the 12 *F. prausnitzii* and two *C. aerofaciens* strains for which we had whole-genome sequences in this study. We first obtained 2,113 GH109 protein sequences from CAZy (http://www.cazy.org/GH109_characterized.html) and then conducted a homolog search for GH109 genes in the genomes of *F. prausnitzii* and *C. aerofaciens* strains using *tblastn* (version 2.5.0+). Indeed, all seven *F. prausnitzii* and two *C. aerofaciens* strains with the GalNAc pathway also have *GH109* in their genomes, and the gene is located in the SV region. Out of the five *F. prausnitzii* strains without GalNAc pathways, *GH109* was found in only one (HTF-441). This suggests that cleavage of free mucosal A-antigens to release GalNAc by GH109 is part of the GalNAc pathway encoded by the SV region.

We have now updated the description of the GalNAc pathway to include a step 0 for the cleavage of A-antigen. Figure 3 has also been updated.

Lines 289-292:

"Specifically, the region contains one gene, *GH109*, that encodes a glycoside hydrolase that can cleave GalNAc from A-antigen, as well as nine genes involved in five key metabolic steps of downstream GalNAc utilization (see **Figure 3B** and **Table S14** for gene description). "

Figure 3. GalNAc utilization underlies the ABO association with *F. prausnitzii* SVs. A, Association between ABO blood type and the presence or absence of genomic segment 577–579 kbp (SV 577–579)

of *F. prausnitzii* is dependent on FUT secretor status. Individuals were grouped into different groups based on blood types (A/AB vs B/O blood types) and FUT2 secretor status (secretors vs non-secretors). The y-axis refers to the fraction of individuals with the 577–579 region in the DMP dataset. Association p-values are reported based on linear mixed models. **B**, Scheme of the GalNAc pathway identified in the associated SV region, which is divided into the initial cleavage from A-antigen (step 0) and five key steps of GalNAc utilization (steps 1–5). In Step 0, the gene GH109 encoding an α -N-acetylgalactosaminidase carries out hydrolysis for the cleavage of A-antigen and the release of GalNAc. In Step 1. GalNAc uptake and trans-membrane transport, four genes – *agaF*, *agaV*, *agaC* and *agaD* – are homologous to the four necessary subunits of the GalNAc PTS system II complex protein (Enzyme II^{aga}). Enzyme II^{aga} takes up exogenous GalNAc and releases the phosphate ester N-acetyl-D-galactosamine 6-phosphate (GalNAc6P) into the cell cytoplasm in preparation for metabolism. In Step 2. Hydrolysis of GalNAc6P, the *nagA* gene encodes N-acetyl-glucosamine-6-phosphate deacetylase, which catalyzes the hydrolysis of the N-acetyl group of GalNAc6P to yield D-galactosamine 6-phosphate (GalN6P) and acetate. In Step 3. Isomerization-deamination of GalN6P, the gene *agaS* encodes the D-galactosamine-6-phosphate deaminase that catalyzes the isomerization-deamination of GalN6P to form D-tagatose 6-phosphate (T6P) and ammonium ion. In Step 4. Phosphorylation of T6P, the gene *lacC* (also known as *pfkB* or *fruK*) encodes the T6P kinase that responds to the formation of tagatose 1,6-bisphosphate (TBP) from T6P with energy exchange. This gene is also involved in the catalytic activity from fructose 6-phosphate (F6P) to fructose-1,6-bisphosphate (FBP). In Step 5. Synthesis of final products, the catalytic subunits of the tagatose-1,6-bisphosphate aldolase (TBPA), *gatY-kbaY* or *gatZ-kbaZ*, can synthesize D-glyceraldehyde 3-phosphate (DHAP) and glyceralone phosphate (GAP) from TBP. **C**, Phylogenetic tree of the strains we used for in vitro culture experiments and the organization of genes involved in GalNAc utilization in the SV region. The x-axis indicates the base pair position starting from the flanking gene *dinB*. The different colored lines indicate the location of same genes in different strains. Names of strains with the SV region are in red. Names of strains without the region are in black. **D–E**, Growth curve of *F. prausnitzii* strains with or without SV 577–579 on media supplemented with GalNAc (D) and Galactose (E). The x-axis refers to the time points in hours. The y-axis refers to cell density as measured by OD₆₀₀ value. Points with bars on the growth curves represent means \pm SD (standard deviation) of three replicates. **F**, Fold change of gene expression upon GalNAc induction compared to glucose induction. The y-axis shows the fold change of the *agaC*, *agaD*, *agaV*, and *agaF* genes in HTF495 and ATCC27768 upon GalNAc induction relative to glucose induction. Each dot indicates one replicate, and the bar indicates the standard error of three replicates.

In addition, we acknowledge the reviewer's comment that mucus contains numerous GalNAc-peptide O-linked sugars. However, it is important to note that normal alpha-1,3-specific hydrolase enzymes, such as GH109, cannot cleave peptide-O-glycan bonds. There are other alpha-linked GalNAc residues in the mucus, e.g., GalNAc-alpha1,3-GalNAc in core-5 and GalNAc-alpha,6-GalNAc in core-7 O-glycans. However, core 5-8 O-glycans have a very limited occurrence¹⁶. Therefore, while these O-linked glycans are highly abundant, they may not serve as a source of free GalNAc for bacteria.

Supporting this notion, a study by Yang et al. (Nature 2022) measured the concentration of

¹⁶ Brockhausen I, Schachter H, Stanley P. O-GalNAc Glycans. In: Varki A, Cummings RD, Esko JD, et al., editors. Essentials of Glycobiology. 2nd edition. Cold Spring Harbor (NY): Cold Spring Harbor Laboratory Press; 2009. Chapter 9. Available from: <https://www.ncbi.nlm.nih.gov/books/NBK1896/>

free GalNAc in the pig cecum and found that animals with the AA blood type have at least twice the amount of free GalNAc compared to animals with the OO blood type. This finding suggests that the concentration of free GalNAc in pigs is also determined by the A blood type. Alpha-linked GalNAc present in mucus glycans may not be readily accessible to bacteria. We believe that a similar situation exists in humans as well.

However, it is worth mentioning that genetic variation in the FUT2 gene in pigs is poorly characterized. A literature search on PubMed using the keywords "FUT2 pigs" (search date: 5 July 2023) yielded 12 studies, but only two of them investigated FUT2 genetic variation. One study examined the association between FUT2 genotype and production and reproductive traits in pigs¹⁷, while another older study demonstrated the similarity between porcine FUT2 and human FUT2¹⁸ but did not identify any mutations in the porcine FUT2 gene. Additionally, studies on fucosyltransferase gene evolution in primates have shown the absence of FUT2 null alleles¹⁹.

Therefore, it is indeed possible that a loss-of-function allele of FUT2 does not exist in pigs.

Question 3.3. Fig. 3A: should the label of the gray histogram bars be “Fut2 non-secretors”?

Response 3.3: Thank you for pointing this out. We have corrected the label of the gray histogram bars in Figure 3A.

Question 3.4. Line 252: What is the meaning of “essential” here? Are these genes essential to the survival of the bacterium and therefore cannot be lost or disrupted (the genetic sense of essential) or are they essential for the pathway in which GalNAc is metabolized. Either way, how is this known from sequence information?

Response 3.4: We apologize for the confusion. These genes are involved in GalNAc utilization, but we do not know how essential each is for bacterial growth and GalNAc utilization. We have now rephrased the sentence.

Line 289-292:

“Specifically, the region contains one gene, GH109, that encodes a glycoside hydrolase that can cleave GalNAc from A-antigen, as well as nine genes involved in five key metabolic steps of downstream GalNAc utilization (see Figure 3B and Table S14 for gene description).”

Moreover, we performed a GalNAc expression-induction experiment for the HTF-495 and ATCC27768 strains in relation to glucose medium. Compared to glucose induction, GalNAc

¹⁷ Wang H, Wu S, Wu J, Sun S, Wu S, Bao W. Association analysis of the SNP (rs345476947) in the FUT2 gene with the production and reproductive traits in pigs. *Genes Genomics*. 2018 Feb;40(2):199-206. doi: 10.1007/s13258-017-0623-7. Epub 2017 Oct 20.

¹⁸ Meijerink E, Fries R, Vögeli P, Masabanda J, Wigger G, Stricker C, Neuenschwander S, Bertschinger HU, Stranzinger G. Two alpha(1,2) fucosyltransferase genes on porcine chromosome 6q11 are closely linked to the blood group inhibitor (S) and Escherichia coli F18 receptor (ECF18R) loci. *Mamm Genome*. 1997 Oct;8(10):736-41. doi: 10.1007/s003359900556. PMID: 9321466.

¹⁹ Pol-André Apoil and others, Evolution of α 2-Fucosyltransferase Genes in Primates: Relation Between an Intronic *Alu-Y* Element and Red Cell Expression of ABH Antigens, *Molecular Biology and Evolution*, Volume 17, Issue 3, March 2000, Pages 337–351

induction resulted in a dramatic increase in GalNAc uptake genes in HTF-495, namely *agaC*, *agaD*, *agaV*, and *agaF*. (Figure S11). This may suggest the importance of GalNAc uptake genes in GalNAc utilization.

Figure S11. Comparison of GalNAc-induced expression fold changes in GalNAc genes between HTF-495 and ATCC27768

Bar plots represent the average fold changes of gene expression induced by GalNAc compared to glucose induction. Each dot represents a sample, and the bar indicates the standard deviation of three replicates. The light blue and dark blue bars represent the expression fold-change of glucose induction and GalNAc induction in relation to glucose induction, respectively. Thus the fold-change in glucose was set to 1. * indicates a difference significant at $P < 0.05$ level (t-test two-sided). ns – not significant.

Question 3.5. Line 272: The inversion of the locus under consideration in the ATCC strain that grows poorly on GalNAc/Gal is interesting. But, how was it determined that this inversion caused the pathway to be inactive as stated? Some other loci seem to have inversions of the teal colored genes (*lacC*, *KbaZ*, *NagA*) and these seem to tolerate the arrangement change. There is another strain in Fig. 3D/GalNAc that grows noticeably poorly on GalNAc. Can this be explained?

Response 3.5: As ATCC27768 grow poorly in the medium with GalNAc as the sole carbon source, we decided to conduct a GalNAc induction experiment by first pre-culturing the ATCC strain in glucose-containing YCFAG medium and then splitting the pre-culture into GalNAc-constraining YCFA-GalNAc medium or YCFAG medium. For comparison, we also selected two closely related strains, HTF-495 and HTF-411, as positive and negative controls, respectively. HTF-495 contains the GalNAc pathway and can grow well in the GalNAc-only medium, whereas HTF-411 does not contain the GalNAc pathway and grows poorly in the GalNAc-only medium. qPCR was then conducted to assess the expression of GalNAc genes and potential regulators (*ptsH*, *rhaR* and *immR*). Interestingly, our results show that expression of the GalNAc-uptake genes *agaV*, *agaC* and *agaD* of ATCC27768 was only marginally induced by GalNAc compared to glucose induction, while expression of these genes was dramatically induced in HTF-495. We also observed that other GalNAc genes in ATCC27768 were equally induced in ATCC27768 and HTF-495. We thus conclude that the genomic inversion in ATCC27768 mainly affects expression of GalNAc-uptake genes.

We have now added this new result to the manuscript.

Line 321-342 (Results):

“Genomic inversion affects expression of GalNAc uptake genes

*ATCC27768 was the only strain that harbors the GalNAc pathway, which showed poor growth in the GalNAc medium. However, the GalNAc region is reversed in ATCC27768 (Figure 3C), and this genomic inversion may result in dysfunction of this pathway. As ATCC27768 grew poorly in GalNAc medium, we carried out a GalNAc induction experiment to investigate the transcription of GalNAc genes and potential regulators (*ptsH*, *rhaR*, and *immR*) in this region. ATCC27768 was first pre-cultured in a glucose medium, and the pre-cultured bacteria was split and transferred to either glucose or GalNAc medium (see Methods). We then compared the expression fold change in GalNAc medium to that in glucose medium. The positive control was the close relative strain HTF-495, which can grow in GalNAc medium. The negative control was HTF-441, which lacks the GalNAc utilization gene cluster (Figure S11).*

*Gene expression of GalNAc genes was not detected in HTF-441, confirming their absence (data not shown). Interestingly, upon GalNAc induction, the expression of three GalNAc uptake genes, *agaC*, *agaD*, and *agaV*, was only marginally increased in ATCC27768, whereas these genes showed a dramatic increase in HTF-495. For instance, GalNAc induction resulted in a 63.5-fold increase in *agaC* expression in HTF-495 compared to glucose induction, but only in a 3-fold change in ATCC27768 (Figure 3F). However, we also noticed that the expression of other GalNAc genes showed similar fold changes in ATCC27768 and HTF-495 (Figure S11). This suggests that genomic inversion of ATCC27768 only affects the expression of GalNAc uptake genes and not GalNAc metabolism genes.”*

Figure 3F, Fold change of gene expression upon GalNAc induction compared to glucose induction. The y-axis shows the fold change of the *agaC*, *agaD*, *agaV*, and *agaF* genes in HTF495 and ATCC27768 upon GalNAc induction relative to glucose induction. Each dot indicates one replicate, and the bar indicates the standard error of three replicates.

Figure S11. Comparison of GalNAc-induced expression fold changes in GalNAc genes between HTF-495 and ATCC27768

Bar plots represent the average fold changes of gene expression induced by GalNAc compared to glucose induction. Each dot represents a sample, and the bar indicates the standard deviation of three replicates. The light blue and dark blue bars represent the expression fold-change of glucose induction and GalNAc induction in relation to glucose induction, respectively. Thus the fold-change in glucose was set to 1. * indicates a difference significant at $P < 0.05$ level (t-test two-sided). ns – not significant.

The reviewer also pointed out that HTF-383 seems to grow poorly in GalNAc medium. We therefore repeated the experiment and extended the culturing time to 40 hours. HTF-383's growth was indeed slower relative to that of other strains, but its OD₆₀₀ value reached over 0.7 at 40 hours, which is comparable to that of the other strains. Thus we conclude that this strain can also grow well in the GalNAc medium, just more slowly.

The growth curve of HTF-383 for a longer period time is now shown in **Figure S9**:

Figure S9. Growth curve of *F. prausnitzii* strain HTF-383 in GalNAc medium over a longer growth time (40 hours)

X-axis refers to culturing time (in hours). Y-axis refers to the cell density measured as OD₆₀₀ value. Red points with bar on the growth curve represent means \pm SD (standard deviation) of three replicates.

Question 3.6. Line 309: The meaning of “biosynthesis cluster” is unclear in this context since the genes that are enumerated are part of a proposed GalNAc degradation pathway. Are they thought to also be involved in synthesizing GalNAc or another molecule?

Response 3.6: Thank you for pointing this out. We have rephrased the sentence, which now reads:

Line 371-373:

“Here we observed a strong inter-correlation between the GalNAc utilization genes, confirming that these genes are likely to be present as a gene cluster and to function collaboratively.”

Question 3.7. Lines 313-333: It's not clear what extra impact classifying individuals as high or low GalNAc secretors is beyond possibly getting some extra statistical power by combining the 3 GalNAc negative groups into one comparison group with a larger subject number? What are the units for the abundance violin plots shown in Fig. S6?

Response 3.7: We apologize for the lack of clarification. The presence of mucosal A-antigen depends on the genotype of *ABO* and *FUT2*. *ABO*-coded A-blood type determines whether an individual has A-antigen in their blood or not. *FUT2* genotype determines whether A-antigen can be secreted to the body fluid and intestinal mucus. Therefore, only *FUT2* secretors with A-blood type can have mucosal A-antigen, while the other three groups do not. We hypothesized that the abundance of GalNAc genes would be more relevant to human health in individuals with mucosal A-antigens than to those without. Thus we would like to compare individuals with or without mucosal A-antigens, making it logical to pool the other three groups together. Of course, we also had a greater analysis power by pooling three groups of individuals without

mucosal A-antigens together.

We have now clarified this in the text.

Line 378-382:

"We further hypothesized that the abundance of GalNAc genes might be more relevant for human health in individuals with mucosal A-antigens than for those without. To check this, we characterized individuals in our cohorts as having either genetically determined presence or absence of A-antigen in intestinal mucus, based on their ABO and FUT2 genotypes."

Figure S6 is now **Figure S12** in the revised manuscript. We have also now added unit information to the figure. The abundance level of GalNAc genes generated by ShortBRED refers to the number of reads mapped to GalNAc genes per kilobase per million total mapped reads (RPKM). The unit of $\ln(\text{RPKM})$ is normalized gene abundance using $\ln(\text{RPKM}+1)$ transformation, with 1 being added to the original RPKM value to deal with zero values.

Question 3.8. Line 321-323: It's not intuitive to me how a bacterial intracellular pathway for metabolizing a hexose sugar to central metabolites would "create niches" for other bacteria or associate with diversity. Both of the P values noted here for increased diversity and association with particular genes in people with A high mucus and A- mucus seem pretty impressive. There are a lot of other gut bacteria that also utilize GalNAC when it is presented as a free sugar (for one example, a recent survey of 354 human gut Bacteroidetes strains from 30 species (PMID: 35166563) showed that 347/354 were able to use this sugar as a sole carbon source. Some of these same species and strains are also one that are capable of cleaving it from blood group A glycans. Even while presented last, this section is quite speculative and would need more experimental work.

Response 3.8: We thank the reviewer for pointing this out. Indeed, there is no evidence to directly support that GalNAc utilization can provide an environmental niche for other species. There could be more GalNAc utilizers in individuals with mucosal A-antigens, hence we observed higher correlation between GalNAc genes and the population diversity and richness in these individuals. We have now removed this speculative sentence. The text now reads:

Line 389-395:

*"As many gut microbes can have the GalNAc pathway, we further hypothesized that the presence of mucosal A-antigen can provide an extra energy source to promote the growth of GalNAc utilizers. In agreement with this, we found that the abundances of GalNAc genes were positively associated with microbial richness and diversity and that these associations were stronger in individuals with mucosal A-antigen ($P_{\text{heterogeneity}} < 0.05$, $I^2 > 0.7$; **Figure 5C and S13; Table S17**)."*

Moreover, we added extra analyses to assess the association between GalNAc genes and host health parameters in individuals with and without mucosal A-antigens. Interestingly, we also detect stronger associations between GalNAc genes and host health in individuals with mucosal A-antigens than in those without. Our data suggest that the abundance of GalNAc genes is more relevant for the health for individuals with mucosal A-antigens.

The new results have now been added to the manuscript and are presented in **Figure 5D-E**.

Line 401-411:

“Similarly, we associated the abundances of gut microbial GalNAc utilization genes with 240 environmental exposure and health-related parameters in individuals with and without mucosal A-antigen. At Bonferroni-corrected $P < 0.05$ level, we detected 50 significant associations in the A-antigen presence group and 17 associations in the A-antigen absence group. Interestingly, gut microbial GalNAc gene abundances were significantly associated with blood glucose, Bristol stool type, and general health only in individuals with mucosal A-antigen (linear regression, Bonferroni-corrected $P < 0.05$, $P_{\text{heterogeneity}} < 0.05$; **Figure 5D–E** and **S14**; **Table S18**). Although we observed 11 significant associations between GalNAc genes and blood triglycerides and high-density lipoprotein in both groups, the effect sizes in the individuals with mucosal A-antigen are higher than in those without ($P_{\text{heterogeneity}} < 0.05$; **Figure S14**).”

Figure 5D–E, D–E, The association between GalNAc metabolism gene abundance and host phenotypes in individuals with mucosal A-antigens (**D**, $N = 1,868$) and without mucosal A-antigens (**E**, $N = 3,866$). # indicates group-specific significant associations. * indicates significant associations shared by the two groups (Bonferroni-corrected $P < 0.05$). Positive associations are in red. Negative associations are in blue. The color gradients reflect the effect size.

Question 3.9. Line 200: Not clear how the potassium channels would be proposed as “medications”. Mean “targets for potential IBD medications.”?

Response 3.9: We apologize for the confusion. We have updated the results section and have removed this sentence.

Question 3.10. Line 189: Not clear what the basis for calling something relevant (presumably vs. less relevant) functionality is in this context.

Response 3.10: We thank the reviewer for this comment. We should have rephrased the paragraph. This sentence has now been removed.

Reviewer Reports on the First Revision:

Referees' comments:

Referee #1 (Remarks to the Author):

The authors have addressed all of my comments. The new analyses and figures lend further support for the conclusions presented in the initial submission.

Referee #2 (Remarks to the Author):

This is a very thorough revision and I thank the authors for addressing all of my comments. The reported association remains robust, and other elements of the paper are improved. I have no further comments or suggestions.

Referee #3 (Remarks to the Author):

The authors have responded admirably to all of the critiques previously raised, especially by providing additional analysis of GH109 enzyme content and gene expression in GalNAc utilizing vs. non-utilizing strains. I have no further criticisms.

Author Rebuttals to First Revision:

Response to Reviewers

Referees' comments:

Referee #1 (Remarks to the Author):

The authors have addressed all of my comments. The new analyses and figures lend further support for the conclusions presented in the initial submission.

Referee #2 (Remarks to the Author):

This is a very thorough revision and I thank the authors for addressing all of my comments. The reported association remains robust, and other elements of the paper are improved. I have no further comments or suggestions.

Referee #3 (Remarks to the Author):

The authors have responded admirably to all of the critiques previously raised, especially by providing additional analysis of GH109 enzyme content and gene expression in GalNAc utilizing vs. non-utilizing strains. I have no further criticisms.

Reply: Thank three reviewers for their positive feedback.